# Efficient recovery and recycling/upcycling of precious metals using hydrazide-functionalized star-shaped polymers

Seung Su Shin [1,4], Youngkyun Jung [2,4], Sungkwon Jeon [1], Sung-Joon Park[1], Su-Jin Yoon[2], Kyung-Won Jung[2], Jae-Woo Choi[2,3] ✉ & Jung-Hyun Lee [1] ✉

There is a growing demand for adsorption technologies for recovering and recycling precious metals (PMs) in various industries. Unfortunately, amine-functionalized polymers widely used as metal adsorbents are ineffective at recovering PMs owing to their unsatisfactory PM adsorption performance. Herein, a star-shaped, hydrazide-functionalized polymer (S-PAcH) is proposed as a readily recoverable standalone adsorbent with high PM adsorption performance. The compact chain structure of S-PAcH containing numerous hydrazide groups with strong reducibility promotes PM adsorption by enhancing PM reduction while forming large, collectable precipitates. Compared with previously reported PM adsorbents, commercial amine polymers, and reducing agents, S-PAcH exhibited significantly higher adsorption capacity, selectivity, and kinetics toward three PMs (gold, palladium, and platinum) with model, simulated, and real-world feed solutions. The superior PM recovery performance of S-PAcH was attributed to its strong reduction capability combined with its chemisorption mechanism. Moreover, PM-adsorbed S-PAcH could be refined into high-purity PMs via calcination, directly utilized (upcycled) as catalysts for dye reduction, or regenerated for reuse, demonstrating its high practical feasibility. Our proposed PM adsorbents would have a tremendous impact on various industrial sectors from the perspectives of environmental protection and sustainable development.

Precious metals (PMs), such as gold (Au), palladium (Pd), and platinum (Pt), are invaluable in various energy- and environment-related applications owing to their exceptional physicochemical properties. These applications include batteries, electronics, fuel cells, catalysts, and several processes within the petroleum industry[1–5]. Because of the scarcity of these PMs, however, the demand for suitable technologies to recover them from natural water or secondary industrial resources such as electronic waste (e-waste) and spent catalysts has grown steadily[6,7].

Hydrometallurgical processes, including membrane filtration[8], adsorption[9], electrochemical treatment[10], and precipitation[11,12], have been employed for PM recovery. Adsorption is a highly effective method for PM recovery because of its simplicity, safety, and excellent recovery efficiency even at low PM concentrations[13,14]. Various PM adsorbents have been developed using organic and/or inorganic materials, including carbon- and silica-based materials, metal oxides, metal–organic frameworks, polymers, and biomass[15]. In particular, polymer-based adsorbents have been extensively employed owing to

[1]Department of Chemical and Biological Engineering, Korea University, Seoul 02841, Republic of Korea. [2]Center for Water Cycle Research, Korea Institute of Science and Technology, Seoul 02792, Republic of Korea. [3]Division of Energy & Environment Technology, KIST School, Korea National University of Science and Technology, Seoul 02792, Republic of Korea. [4]These authors contributed equally: Seung Su Shin, Youngkyun Jung. ✉e-mail: plead36@kist.re.kr; leejhyyy@korea.ac.kr

their high adsorption capacity, scalability, and chemical tunability[16]. Commercially available amine-functionalized polymers, such as branched poly(ethyleneimine) (bPEI) and poly(allylamine hydrochloride) (PAAm), are commonly used to fabricate PM adsorbents because of their high electrostatic affinity to PM ions[17–20]. However, these polymers cannot be used as standalone adsorbents because of their small particle size, which hinders their collection. Consequently, they must be chemically anchored onto porous supports (e.g., fibers, beads, and sheets) with a large surface area[21–23]. Unfortunately, the use of supports with a high weight fraction inevitably reduces the weight-based PM adsorption capacity of the adsorbent and complicates its fabrication[24]. Additionally, the relatively slow PM adsorption kinetics exhibited by commercial amine polymer-based adsorbents can hinder high-throughput PM recovery[25]. These factors underscore the necessity of developing advanced polymer adsorbents that can rapidly recover PMs with high capacity and selectivity.

In this study, we synthesize a star-shaped, hydrazide-functionalized polymer (poly(acryloyl hydrazide), S-PAcH)[26] and demonstrate that it can be used as a standalone adsorbent to achieve highly efficient, selective, and rapid PM recovery. Combined with its electrostatic chemisorption ability, the highly reducible hydrazide groups of S-PAcH can effectively reduce PM ions to metal nanoparticles (NPs)[27–29], thereby enhancing its adsorption capability and selectivity toward PMs. The reduction-mediated formation of PM NPs can simultaneously induce the intra/intermolecular chain fusion of S-PAcH to produce large and stable precipitates that can be collected easily. In particular, the star-shaped architecture of S-PAcH, which features densely packed PAcH linear arms, can provide a high density of hydrazide groups, further improving its PM adsorption performance while promoting its intra/intermolecular-fusion-induced precipitation.

The PM adsorption capacity and selectivity of S-PAcH were characterized using model and simulated feed solutions and compared with those of its linear counterpart and commercial amine polymers (i.e., bPEI and PAAm) (Supplementary Table 1). Based on these experiments, we identified the effects of the architecture and chemistry of the standalone polymer on PM adsorption performance, which have not yet been investigated. Furthermore, the structural and physicochemical properties of S-PAcH before and after PM adsorption were comprehensively analyzed to identify its PM adsorption mechanism that can be utilized for designing standalone adsorbents. We also demonstrate that PM-adsorbed S-PAcH (PM/S-PAcH) precipitates can be refined into high-purity PMs, utilized directly as catalysts for dye reduction, or regenerated for reuse. Finally, we highlight the high commercial viability of S-PAcH by demonstrating its higher PM recovery performance and recoverability compared with commercial amine polymers and reducing agents using real-world leachate feed solutions.

## Results

### Physicochemical and PM adsorption properties of S-PAcH
S-PAcH was synthesized by growing ~19 poly(methyl acrylate) (PMAc) linear arms on a β-cyclodextrin (CDx) core via atom transfer radical polymerization (S-PMAc), followed by amination with hydrazine (Fig. 1a and Supplementary Figs. 1–3). S-PAcHs with short and long PAcH arms, denoted by S-PAcH(s) and S-PAcH(L), respectively, and the linear counterpart of S-PAcH(s), denoted by L-PAcH, were synthesized (Supplementary Figs. 3 and 4 and Supplementary Table 1). S-PAcH possessed a spherical morphology with a diameter of approximately 9 nm, exhibiting excellent solubility in water (i.e., the solubility limit of ~10 wt.%) (Fig. 1b and Supplementary Fig. 5). Under acidic conditions, the PAcH-series polymers (PAcHs) exhibited positive charges that decreased with increasing pH because of the reduced protonation of their carbonyl and amine groups (Fig. 1c)[29,30]. Beyond the isoelectric point (~5.5), the charges of PAcHs transitioned to negative. Moreover,

compared with bPEI and PAAm, PAcHs had lower positive charges (at pH <6) because of their electronegative carbonyl groups[31].

The addition of S-PAcH to PM-ion-containing aqueous solutions at pH 2 resulted in rapid precipitation (Fig. 1d and Supplementary Fig. 6). The precipitates could be readily collected by simple sedimentation, centrifugation, or membrane filtration (Fig. 1d, magnified image)[32]. Notably, S-PAcH induced more prominent and faster precipitation than L-PAcH (Supplementary Fig. 6), demonstrating the higher precipitation propensity of the star-shaped polymer. By contrast, precipitation was not observed when commercial amine polymers were added to the PM-ion-containing solutions (Supplementary Fig. 6). The observed precipitation, particularly with S-PAcH, is likely owing to the formation of PM NPs via reduction; this reduction process is facilitated by the hydrazide groups of the adsorbents, which have potent reducing capabilities[26–29]. Compared with isolated amines, the hydrazide, consisting of two adjacent amines bonded to a carbonyl group, exhibits a higher propensity for electron donation (i.e., reduction) because of the efficient resonance stabilization of the oxidized hydrazine amine by the adjacent amine group[33]. PM reduction by PAcHs was evidenced by the observations of the characteristic color (i.e., red to purple) and ultraviolet-visible (UV-vis) peak of Au NP when PAcHs were added to Au aqueous solutions (Supplementary Figs. 6 and 7).

The precipitation behavior of the polymers was characterized by monitoring the changes in their hydrodynamic diameters ($H_R$) with increasing contact time in an Au aqueous solution (Fig. 1e). The $H_R$ of most of the polymers increased rapidly and plateaued within 30 min. Specifically, bPEI and PAAm achieved $H_R$ values at the nanoscale level because their PM adsorption mechanisms mainly rely on electrostatic and chelation interactions[34], which are ineffective at inducing precipitation. By contrast, the $H_R$ values of PAcHs reached the microscale level, enabling their facile collection. The growth of precipitates was observed to be larger and faster for S-PAcH when compared with L-PAcH, which is consistent with our visual observations (Supplementary Fig. 6). A similar precipitation trend was observed when the adsorbents were added to Pd and Pt aqueous solutions (Supplementary Fig. 8). PAcHs reduced PM ions to NPs, which likely acted as strong binders to induce the intra/intermolecular fusion of PAcH chains, consequently leading to the formation of large aggregates[27]. Compared with the linear structure of L-PAcH, the star-shaped structure of S-PAcH, characterized by multiple PAcH linear arms radially confined to a core, results in more compact chains with a higher local density of hydrazide groups by restricting chain entanglement[35,36]. Therefore, S-PAcH is more conducive to inducing intra/intermolecular fusion compared with L-PAcH because it can enable multiple chain contacts and provide a greater number of reducing sites[37], thereby facilitating precipitation.

The mechanical integrity of the PAcH-induced precipitates was characterized by monitoring the $H_R$ of the Au/PAcH precipitates with increasing solution rotation speed (Fig. 1f). Whereas L-PAcH showed a substantial decrease in $H_R$ with increasing rotation speed, S-PAcHs maintained their high $H_R$. This indicates that the star-shaped structure of S-PAcH is also beneficial for forming strong precipitates by reinforcing the intra/intermolecular fusion mechanism[38], thus enabling their reliable recovery even under realistic high-shear conditions. In addition, the high stability of the Pd/S-PAcH and Pt/S-PAcH precipitates was confirmed (Supplementary Fig. 9).

### PM adsorption mechanism of S-PAcH
We tested our hypothesis regarding the PM adsorption mechanism of S-PAcH under acidic conditions by characterizing the PM/S-PAcH precipitates formed at pH 2 using various analytical tools. Transmission electron microscopy (TEM) identified the crystal lattice structure of the PM NP corresponding to each PM/S-PAcH (Figs. 2a–c and Supplementary Fig. 10). X-ray diffraction (XRD) analysis detected the characteristic diffraction peaks of the reduced PM polycrystal for each PM/S-PAcH (Fig. 2d)[39]; no crystalline XRD peaks were observed for

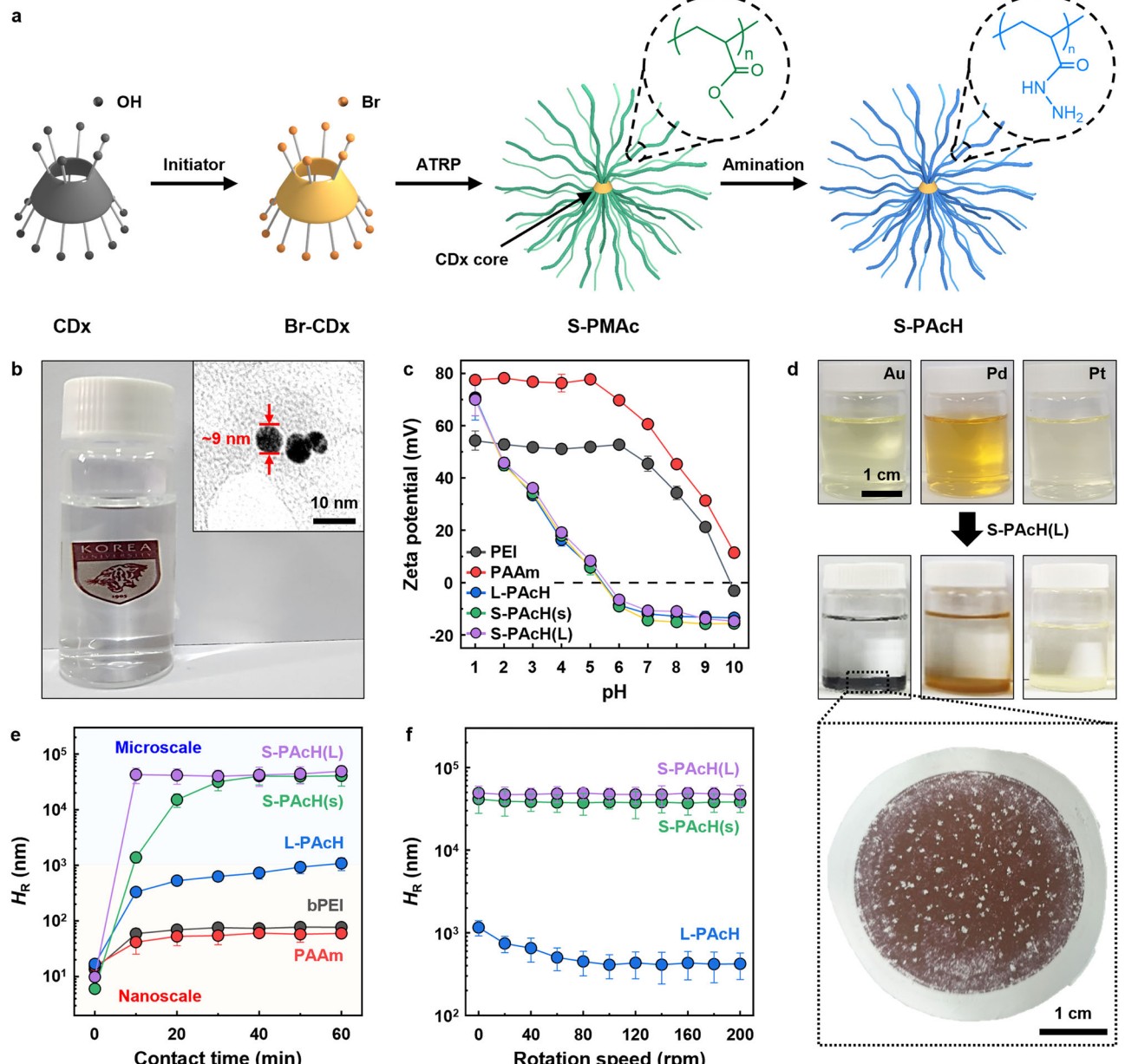

**Fig. 1 | Physicochemical properties and PM adsorption behavior of the polymers. a** Schematic of the synthesis process of S-PAcH via atom transfer radical polymerization (ATRP) and amination. **b** Photograph of a S-PAcH(L) aqueous solution (Inset: TEM image of S-PAcH(L)). **c** Zeta potentials of commercial amine (bPEI and PAAm) and PAcH-series polymers as a function of solution pH. **d** Photographs of three PM (Au, Pd, and Pt, 200 mg L$^{-1}$) aqueous solutions (pH = 2) before (top) and after (bottom) the addition of S-PAcH(L) (0.2 g L$^{-1}$). The magnified photograph shows the Au/S-PAcH(L) precipitates collected by filtration with a polysulfone ultrafiltration membrane. **e** Hydrodynamic diameter ($H_R$) of the polymers after their addition (0.2 g L$^{-1}$) to a Au (200 mg L$^{-1}$) aqueous solution (pH = 2) as a function of contact time. **f** $H_R$ of the Au/PAcH precipitates as a function of rotation speed. The precipitates were formed by allowing contact between the PAcH-series polymers (0.2 g L$^{-1}$) and the Au (200 mg L$^{-1}$) aqueous solution (pH = 2) for 24 h. Error bars in (**b**, **e**, **f**) represent standard deviations determined from three replicates.

PM/bPEI and PM/PAAm (Supplementary Fig. 11). Moreover, X-ray photoelectron spectroscopy (XPS) revealed that the PM/S-PAcH precipitates exhibited two deconvoluted PM peaks corresponding to ionic and reduced PM (PM(0)) metal states with a high fraction of the metal state (80–89%) (Fig. 2e, Supplementary Fig. 12 and Supplementary Table 2); this is in contrast to PM/bPEI and PM/PAAm, which showed only ionic PM peak (Supplementary Figs. 13 and 14). All characterization results supported the reduction of PM ions to PM NPs by S-PAcH. PM/S-PAcH precipitates formed with different S-PAcH concentrations exhibited the nearly identical fraction of the PM metal state (Supplementary Fig. 15 and Supplementary Table 3), indicating that the degree

of PM reduction is determined by the inherent reduction capability of the hydrazide group of S-PAcH. The N1s XPS peak of PM/S-PAcH was broader (the width at zero point of ∼5.8) than that of the pristine S-PAcH (the width at zero point of ∼4.5) and deconvoluted into three peaks at 399.7 (–NO$_2$), 400.5 (N–metal–N), and 401.9 (protonated amine) eV[20], which were absent for S-PAcH (Fig. 2f and Supplementary Figs. 2 and 16). Deconvolution of the C1s peak revealed two peaks at 284.8 (C–C) and 288.0 (O=C–N) eV for both S-PAcH and PM/S-PAcH (Fig. 2g and Supplementary Fig. 17)[28]. These results suggest that protonated amines, –NO$_2$ groups, and N–metal–N chelation bonding are formed while carbonyl oxygen atoms remaining unprotonated in

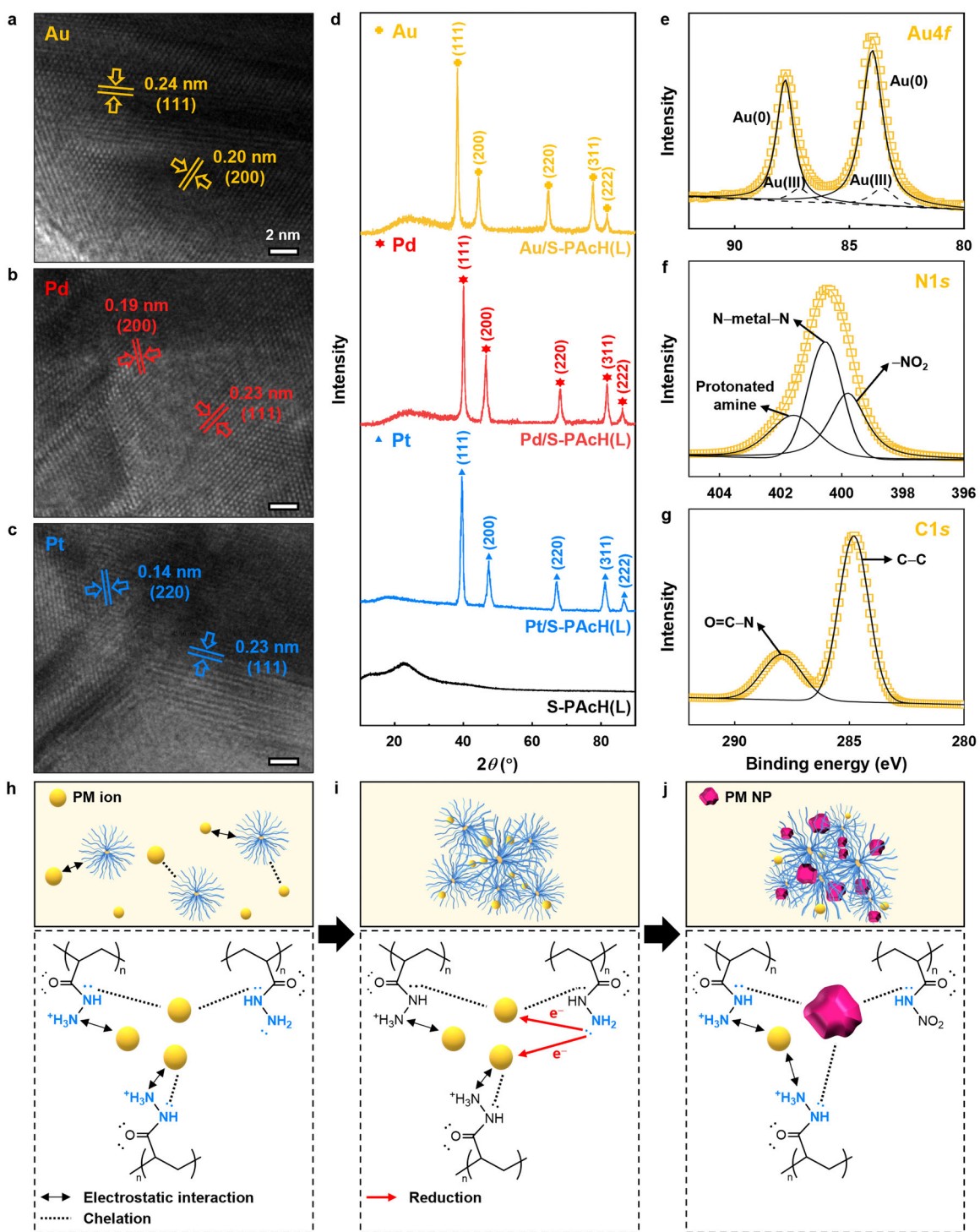

**Fig. 2 | Characterization of the PM/S-PAcH(L) precipitates and proposed PM adsorption mechanism of S-PAcH. a–c** TEM images of the PM/S-PAcH(L) precipitates: (**a**) Au, (**b**) Pd, and (**c**) Pt. **d** XRD patterns of the pristine S-PAcH(L) and PM/S-PAcH(L) precipitates. **e–g** Deconvolution of the high-resolution (**e**) Au4*f*, (**f**) N1*s*, and (**g**) C1*s* XPS peaks of the Au/S-PAcH(L) precipitate. The precipitates were formed by allowing contact between S-PAcH(L) (0.2 g L⁻¹) and PM (200 mg L⁻¹) aqueous solutions (pH = 2) for 3 h. **h–j** Proposed PM adsorption mechanism of S-PAcH.

PM/S-PAcH after PM adsorption. The formation of −NO₂ groups in S-PAcH after PM adsorption was further confirmed by FT-IR analysis where PM/S-PAcH exhibited the peak at 1596 cm⁻¹ (N=O stretching, −NO₂)[40], which was absent for S-PAcH (Supplementary Fig. 18).

Given the results above, the PM adsorption mechanism of S-PAcH at low pH (i.e., pH 2 where PM adsorption tests were performed) can be depicted as illustrated in Fig. 2h–j. PM ions would exist as deprotonated anionic species (i.e., $AuCl_4^-$, $PdCl_4^{2-}$, and $PtCl_6^{2-}$) at low pH owing to the strong acidity of their precursors[41]. The primary amines (−NH₂) of S-PAcH are protonated preferentially over its secondary amines (−NH−) under acidic conditions owing to their higher basicity (i.e., electron donating nature)[33]. Furthermore, hydrazide −NH₂ of

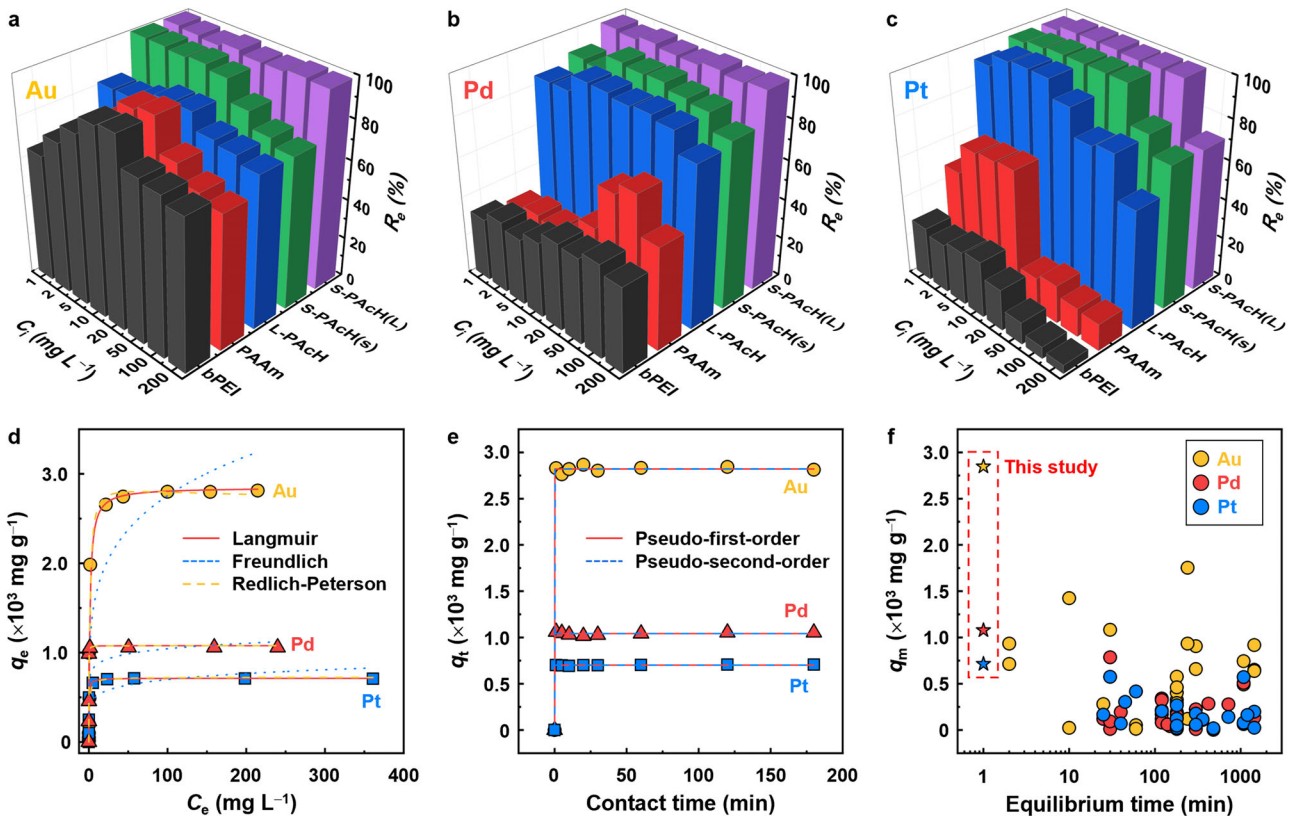

**Fig. 3 | PM recovery performance of the polymers and adsorption mechanism analysis.** Recovery efficiency ($R_e$) of commercial amine (bPEI and PAAm) and PAcH-series polymers for (**a**) Au, (**b**) Pd, and (**c**) Pt as a function of initial PM ion concentrations ($C_i$) (polymer concentration = 0.2 g L⁻¹, solution pH = 2, contact time = 3 h). **d** Corresponding adsorption isotherms of S-PAcH(L) for three PMs (Au, Pd,

and Pt) and their fits to three isotherm models. **e** Adsorption kinetics of S-PAcH(L) for three PMs and their fits to two kinetics models (S-PAcH(L) concentration = 0.2 g L⁻¹, $C_i$ = 200 mg L⁻¹, solution pH = 2). **f** Comparison of the PM adsorption performance of S-PAcH(L) with those of other reported PM adsorbents.

S-PAcH would presumably be protonated to a less extent than conventional −NH₂ owing to its lower basicity (i.e., higher p$K_b$)[42]. This was evidenced by the fact that the positive zeta potential of S-PAcH noticeably increased while those of conventional amine polymers were nearly unchanged when pH decreased from 2 to 1 (Fig. 1c). Hence, it can be reasonably postulated that S-PAcH at low pH (i.e., pH 2) contains both protonated (−NH₃⁺) and unprotonated (−NH₂) primary amines (i.e., protonated and unprotonated hydrazides). Under this circumstance, the protonated −NH₃⁺ of S-PAcH would adsorbs anionic PM species via long-range electrostatic interactions[30,43], followed by ion-exchange and chelation mainly with the unshared electron-bearing nitrogen atoms of its unprotonated −NH− (Fig. 2h)[14,44]. Meanwhile, S-PAcH molecules coagulate owing to their screened electrostatic charges. Subsequently, the unprotonated −NH₂ of S-PAcH then reduce the adsorbed PM ions to NPs while being converted into −NO₂, as given by −NH₂ + 2H₂O (solvent water) → −NO₂ + 6H⁺ + 6e⁻ (Fig. 2i and j)[28]. Because one −NH₂ group of S-PAcH provides six electrons during its oxidation to −NO₂, it can reduce multiple PM ions (i.e., 2 for AuCl₄⁻, 3 for PdCl₄²⁻, and 1.5 for PtCl₆²⁻) to PM NPs[28] (Fig. 2i and Supplementary Note 1). Continuous PM reduction leads to NP growth and induces intra/intermolecular chain fusion through chelation (N−metal−N) between the NPs and unshared electron-bearing nitrogen atoms of −NH− in neighboring PAcH chains, leading to the rapid formation of large and robust precipitates (Fig. 2j). A small fraction of the adsorbed PM species exists as an ionic state in PM/S-PAcH via ion electrostatic and chelation interaction (Fig. 2j), as evidenced by the ionic PM and N1s (corresponding to the protonated amine) XPS peaks detected for PM/S-PAcH.

## PM adsorption performance of S-PAcH

Similar to other studies, PM adsorption tests were performed at pH 2 because the optimal PM recovery efficiency ($R_e$) was yielded at pH 2 for all the investigated adsorbents, and typical PM leaching effluents are strongly acidic (pH 0–2) (Supplementary Figs. 19–21 and Supplementary Note 2)[43,45,46]. Commercial amine polymers achieved the maximum $R_e$ for all PMs at a certain initial PM ion concentration ($C_i$) (Fig. 3a–c) because their interaction probability with PM ions becomes low at low $C_i$ while their adsorption sites are saturated at high $C_i$[47]. By contrast, S-PAcH, in particular S-PAcH(L), exhibited very high $R_e$ (~100%) for all PMs even at low $C_i$ (<50 mg L⁻¹), in which microscale precipitation was not induced (Fig. 3a–c and Supplementary Figs. 6 and 22), demonstrating its superior PM recovery performance. S-PAcH(L) also maintained its very high $R_e$ (>99%) for all PMs even at 1 M hydrochloric acid (HCl) (corresponding to pH ~0) (Supplementary Fig. 23 and Supplementary Note 2). Commercial amine polymers exhibited good Au adsorption ability but were ineffective at adsorbing Pd and Pt. The amine polymers adsorb metal species mainly via electrostatic interactions and subsequent ion-exchange with protonated amines and chelation with unprotonated amines[48]. Their low $R_e$ for Pd and Pt can be explained by the fact that the ion-exchange process is more favorable for monovalent Au ions than for divalent Pd and Pt ions[49]. Furthermore, because Pt ions, which exhibit a larger ionic radius (i.e., lower charge density) than Pd ions, are less effectively adsorbed owing to weaker electrostatic interactions, the amine polymers exhibited a lower $R_e$ for Pt than for Pd[50].

L-PAcH exhibited a higher $R_e$ than bPEI and PAAm, particularly for Pd and Pt. Although L-PAcH had a lower positive charge than bPEI and

PAAm, its stronger reducing ability, which is imparted by hydrazide groups, predominantly favored PM adsorption over electrostatic and chelation interactions[48], resulting in its substantially higher $R_e$ even for Pd and Pt. Notably, S-PAcH(s) exhibited a noticeably higher $R_e$ than L-PAcH, thereby demonstrating the advantages of a star-shaped polymer architecture in PM adsorption; a star-shaped structure with compact arm chains provides a higher local density of collaborative adsorptive sites than a linear structure[35]. S-PAcH(L) with longer arms had a higher $R_e$ than S-PAcH(s), presumably because longer arm chains can increase the free volume through which PM species readily permeate[35].

The adsorption isotherm data of S-PAcH(L) for the three PMs fit the Langmuir and Redlich−Peterson models ($\alpha$ value = 1.0, equivalent to the Langmuir model) ($R^2 > 0.80$) better than the Freundlich model (Fig. 3d and Supplementary Table 4). This result indicates that PM species are adsorbed on S-PAcH(L) primarily via homogeneous monolayer formation[51], which is consistent with our proposed mechanism that homogeneous-monolayered, chemisorbed PM ions[48] are subsequently reduced to PM NPs. The maximum adsorption capacities ($q_m$) of S-PAcH(L) for Au, Pd, and Pt were determined to be 2847, 1078, and 714 mg g$^{-1}$, respectively, from the model fitting. These values are qualitatively consistent with the results of the thermogravimetric analysis (TGA) (Supplementary Fig. 24). The higher $q_m$ of S-PAcH(L) for Au than for Pd and Pt can be explained by the higher reduction potential (i.e., higher tendency to undergo reduction)[47] and ion-exchange ability[49] of Au ion species ($AuCl_4^-$). S-PAcH(L) also exhibited rapid PM adsorption, achieving equilibrium adsorption for all three PMs within 1 min, thereby enabling high-throughput recovery. Unfortunately, we were unable to determine an appropriate kinetics model for this adsorbent because of its high adsorption rate and short equilibrium time (i.e., the time when adsorption capacity reaches 98% of the equilibrium value). The adsorption kinetic data fit both the pseudo-first-order and pseudo-second-order models well ($R^2 \approx 1.00$) (Fig. 3e and Supplementary Table 5). Nevertheless, we speculate that the rate-limiting step of PM adsorption by S-PAcH(L) is chemisorption, which can dictate the reduction of adsorbed PM ions[52]. Compared with other reported PM adsorbents, S-PAcH(L) exhibited significantly higher $q_m$ values and shorter adsorption equilibrium times for all three PMs (Fig. 3f and Supplementary Table 6). S-PAcH(L) was also more effective at recovering PMs than conventional reducing agents such as hydrazine and sodium borohydride (NaBH$_4$) (Supplementary Figs. 25−27 and Supplementary Note 3). This result highlights the beneficial feature of S-PAcH(L) with both adsorbent and reductant functions, which synergistically improves PM recovery performance above that achievable by amine polymers with an adsorption function only or reducing agents with a reduction function only; the unprecedentedly high-capacity and rapid PM adsorption of S-PAcH(L) can be attributed to its high reduction capability combined with its effective adsorption mechanism via strong electrostatic and chelation interactions, endowed by its numerous hydrazide groups that are effectively packed in a star-shaped configuration.

## Practical applications of S-PAcH

Selective PM adsorption is of critical importance for the practical application of adsorbents because real PM-containing feeds include other metal ions. A computer central processing unit (CPU) is a representative type of e-waste that contains considerable amounts of Au along with Cu and Ni ions[53]. In the case of a simulated CPU leachate, S-PAcH(L) achieved ~100% $R_e$ for Au but negligible $R_e$ values for Cu (~1.6%) and Ni (~1.2%), thereby demonstrating its remarkably high adsorption capacity and selectivity toward Au; indeed, the $R_e$ and selectivity toward Au achieved by this adsorbent substantially exceeded those of bPEI and PAAm (Fig. 4a). Spent catalysts, which are typically supported by γ-alumina (Al$_2$O$_3$) substrates, are regarded as secondary resources for Pd and Pt[53]. For the simulated leachates of spent alumina-supported catalysts, S-PAcH(L) preferentially adsorbed

both Pd and Pt (i.e., ~100% $R_e$) over Al (i.e., <3% $R_e$), thus confirming its significantly higher $R_e$ and selectivity toward Pd and Pt compared with those of bPEI and PAAm (Fig. 4b and c). Furthermore, S-PAcH(L) selectively and completely recovered trace amounts of Au, Pd, or Pt (i.e., ~100% $R_e$) from simulated groundwater containing abundant coexisting Na, K, Mg, and Ca metal ions (i.e., <2.1% $R_e$), which was not achieved by bPEI and PAAm (Fig. 4d and Supplementary Fig. 28). Unlike anionic PM ion species, these other coexisting metal ions are cationic, and thus, can be electrostatically repelled by positively charged amine and hydrazide polymers. Although S-PAcH(L) with a lower positive charge was expected to adsorb PM ions less selectively than bPEI and PAAm, its strong PM reduction ability likely significantly enhanced its adsorption capacity and selectivity toward PMs with relatively higher reduction potentials[27]. Compared with reducing agents that can reduce coexisting cations as well as PM ions[54], S-PAcH(L) also exhibited significantly higher selectivity toward PMs (Supplementary Fig. 29 and Supplementary Note 4), further highlighting the benefit of its both adsorbent and reductant functions in selective PM recovery. Moreover, considering the difficulty in recovering small molecular-sized reducing agents[54], we believe that our PAcH with higher PM adsorption performance and recoverability would be more cost-effective at recovering PMs compared with reducing agents (Supplementary Fig. 30, Supplementary Tables 7 and 8, and Supplementary Note 5). S-PAcH(L) also maintained its high $R_e$ for PMs even for simulated feed solutions containing model organic pollutants that can possibly impair the PM adsorption performance of adsorbents by forming adsorbent−pollutant complexes (Supplementary Fig. 31 and Supplementary Note 6). This result demonstrates the high adsorption selectivity of S-PAcH(L) toward PMs over organic pollutants, which can be attributed to its rapid PM adsorption capability combined with its poor affinity with relatively hydrophobic organic pollutants[55]. In combination with its excellent PM adsorption performance, the remarkable ability of S-PAcH to form large and strong precipitates that can be readily collected by membrane filtration enables the selective and efficient recovery of PMs from complex feeds containing coexisting metal ions and organic compounds (Fig. 4e).

To demonstrate the practical feasibility of employing S-PAcH in PM recovery, we calcined the collected PM/S-PAcH(L) precipitates in air at 600 °C and subsequently treated them with a HCl (37%) solution followed by heat (80 °C) (Fig. 4f). The obtained materials were found to be solid PM particles with a purity of 99.9%, corresponding to 24 Karat, which can be reused as raw materials for various applications (referred to as a recycling process). Because S-PAcH can adsorb PMs in a reduced NP form, the PM/S-PAcH precipitates can also be directly utilized as a catalyst in chemical reactions (referred to as a value-added upcycling process)[56]. To verify this inference, we evaluated the catalytic activity of PM/S-PAcH for the reduction of 4-nitrophenol (4-NP) and methyl orange (MO), which are representative organic dye contaminants commonly found in chemical industry wastewater[57]. All reaction tests were performed with 4-NP or MO-containing feed solutions in the presence of NaBH$_4$ as a reducing agent (Fig. 4f). When PM/S-PAcH(L) was added to the dye-containing feed solutions, the solutions lost their characteristic color and became transparent (Fig. 4f), indicating complete dye reduction. Consistently, the addition of PM/S-PAcH(L) to the dye solutions resulted in the complete disappearance of the characteristic UV-vis peaks of 4-NP and MO[58], which was not observed when S-PAcH(L) was added to the dye solutions (Fig. 4g and Supplementary Fig. 32). Turnover number (TON) and turnover frequency (TOF) are critical performance metrics for evaluating catalytic activity. PM/S-PAcH(L) exhibited comparable (Au/S-PAcH(L)) and/or even higher (Pd and Pt/S-PAcH(L)) TON and TOF values compared with other reported catalysts (Supplementary Fig. 33 and Supplementary Tables 9 and 10). This result confirms the excellent catalytic activity of PM/S-PAcH in the reduction reaction and

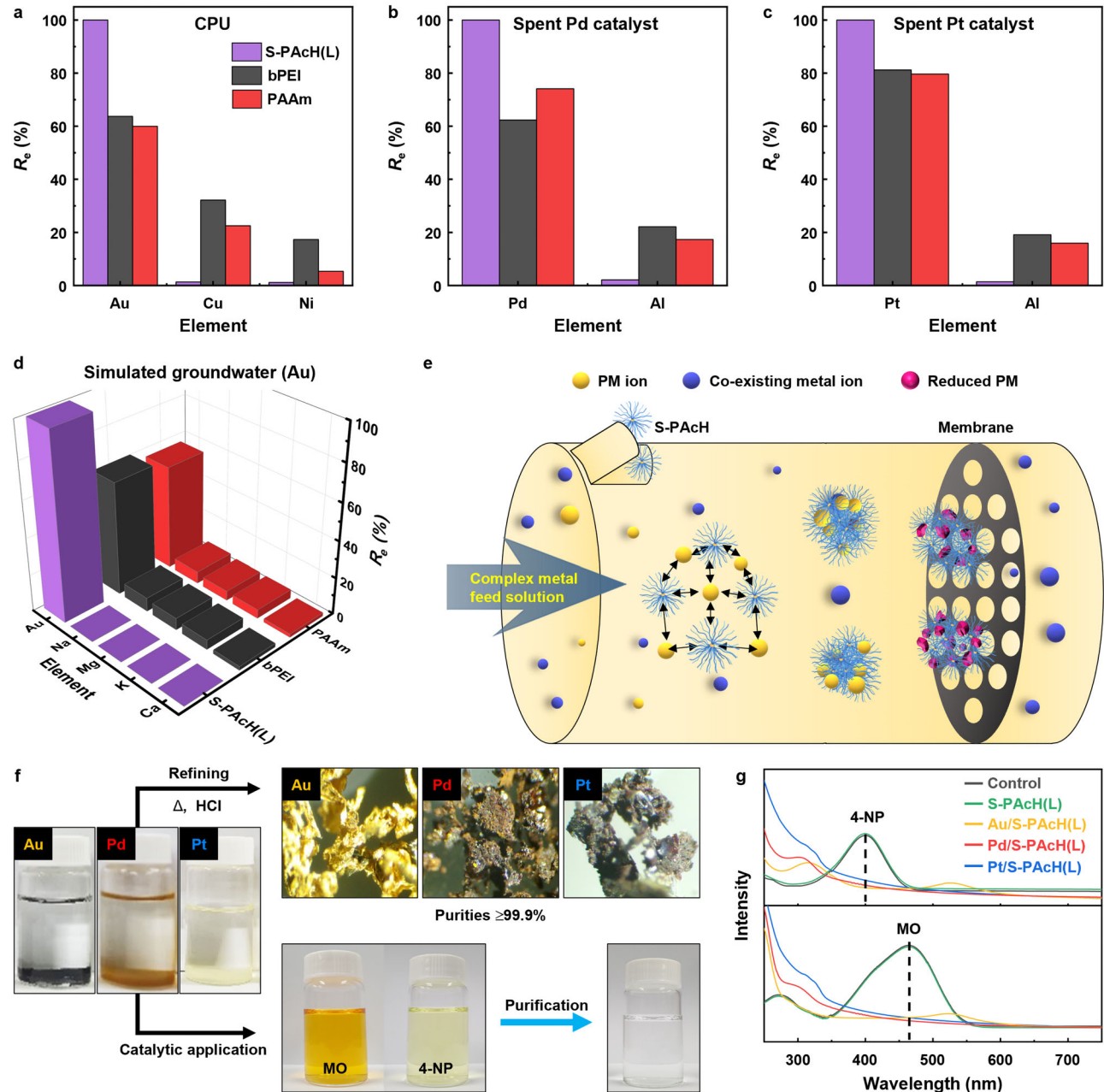

**Fig. 4 | Investigation of PM recovery with simulated feed solutions and demonstration of the practical feasibility. a–d** Recovery efficiency ($R_e$) of the polymers (S-PAcH(L), bPEI, and PAAm) with simulated leachate and groundwater feed solutions: (**a**) CPU leachate, (**b**) spent Pd catalyst leachate, (**c**) spent Pt catalyst leachate, and (**d**) groundwater (Au) (polymer concentration = 0.2 g L$^{-1}$, solution pH = 2, contact time = 3 h). **e** Schematic of the process of selective PM recovery from a complex metal feed solution using S-PAcH. **f** Refinement of the PM/S-PAcH(L) precipitates into pure PMs (top) and their direct catalytic application to dye (MO and 4-NP) reduction (bottom). **g** UV-vis spectra of organic dye (MO and 4-NP) solutions containing NaBH$_4$ before (control) and after the addition of the S-PAcH(L) or PM/S-PAcH(L) precipitates.

demonstrates its potential utility in various environmental and energy-related applications, including petroleum cracking, carbon dioxide reduction, water remediation, and hydrogen energy production[5].

An electro-sorption process, in which PM ions are recovered via reduction on the electrode surface under electric potential, has also been employed for selective PM recovery[10]. Unfortunately, the electro-sorption process exhibits a trade-off between PM selectivity and recovery rate depending on the electric potential strength[59]. In contrast, combined with its high reduction capability, the strong electrostatic repulsion of S-PAcH toward coexisting metal cations enables highly selective and rapid PM recovery, overcoming the trade-off of the electro-sorption process. Recently, an innovative precipitation

method for Au recovery using a simple tertiary diamide compound as a highly Au-selective and recyclable precipitant has been proposed by other researchers[12]. This unique strategy induces precipitation by forming a supramolecule between the proton-chelated structure and Au ions via chemical interactions, while our approach induces precipitation by forming Au NPs via reduction. Although the diamide precipitant selectively recovers Au from acidic solutions, its recovery performance could be significantly affected by its dissolution process owing to its limited solubility in water, unlike our highly water-soluble S-PAcH adsorbent. Moreover, compared with the diamide precipitant displaying low Pd and Pt uptake, our S-PAcH exhibits excellent adsorption capacity and selectivity toward Pd and Pt, indicating its

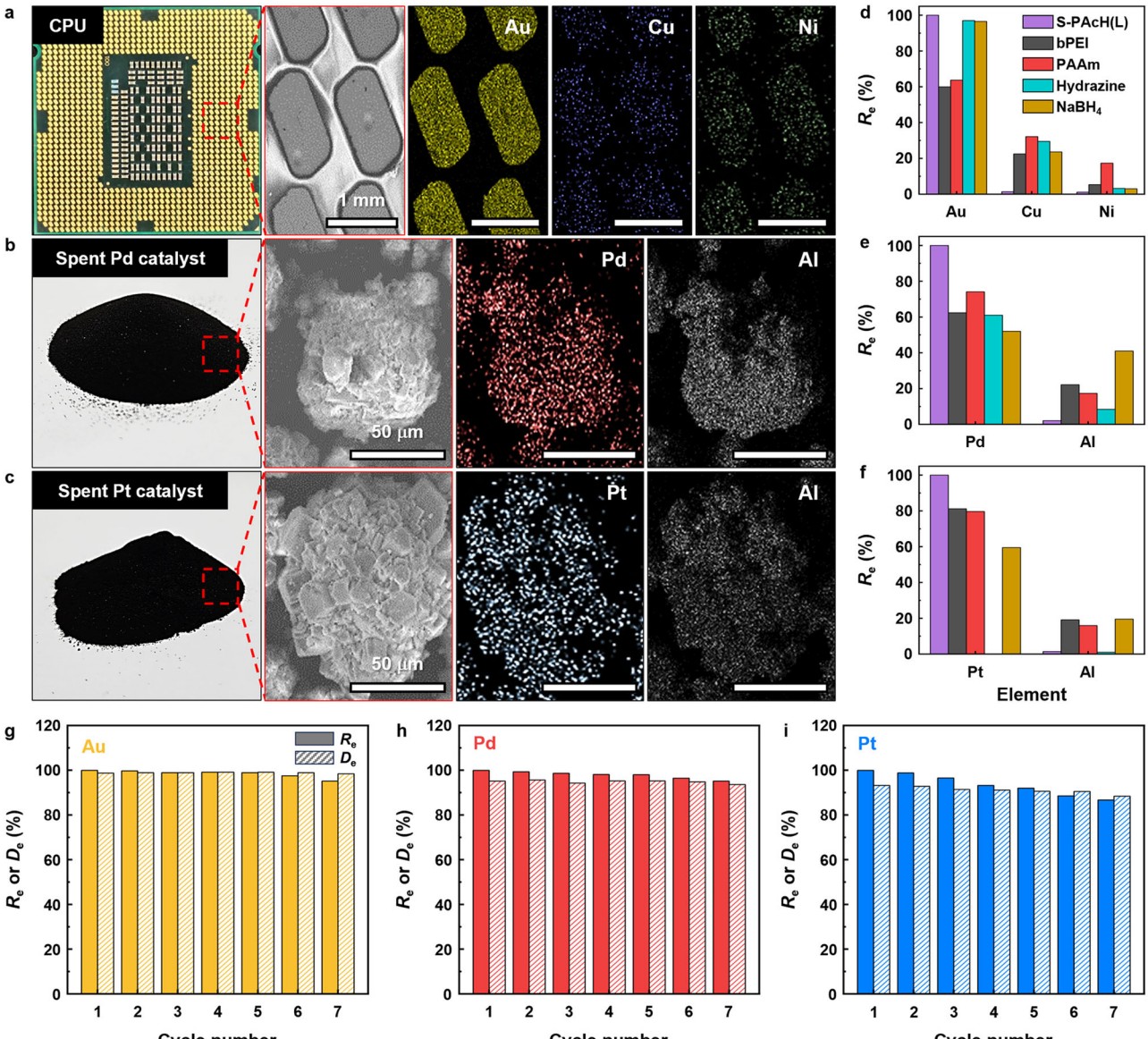

**Fig. 5 | Investigation of PM recovery with real-world feed solutions and evaluation of adsorbent reusability. a–c** Photographs, SEM, and SEM-EDX images of real-world samples: (**a**) CPU, (**b**) spent Pd catalyst, and (**c**) spent Pt catalyst. **d,e**, Recovery efficiency ($R_e$) of the polymers (S-PAcH(L), bPEI, and PAAm) and reducing agents (hydrazine and NaBH$_4$) with real-world leachate feed solutions: (**d**) CPU leachate, (**e**) spent Pd catalyst leachate, and (**f**) spent Pt catalyst leachate (polymer and reducing agent concentration = 0.2 g L$^{-1}$, solution pH = 2, contact time = 3 h). **g–i**, $R_e$ and desorption ($D_e$) efficiency of S-PAcH(L) with real-world leachate solutions as a function of the number of adsorption–desorption cycles: (**g**) CPU leachate (Au), (**h**) spent Pd catalyst leachate, and (**i**) spent Pt catalyst leachate.

versatile use for PM recovery. Although more toxic agents (thiourea and Fe$^{3+}$) are needed for regenerating S-PAcH compared with that (deionized (DI) water) needed for recovering the diamide precipitant, S-PAcH can be sustainably reused via a well-established regeneration method, as will be demonstrated below. S-PAcH can also be upcycled for value-added usages (e.g., catalysts) because it can adsorb PMs as a reduced metal NP form, which is not feasible for the diamide precipitant that adsorbs PMs as an ionic form only.

We further assessed the feasibility of S-PAcH for its use in PM recovery from the leachates of real-world CPU and spent Pd/Pt catalysts (Figs. 5a–c and Supplementary Fig. 34). S-PAcH(L) completely recovered PMs (i.e., ~100% $R_e$) from all the real-word leachate solutions without absorbing coexisting metal ions (Fig. 5d–f) and organic pollutants (Supplementary Fig. 35). Compared with S-PAcH(L), commercial amine polymers and reducing agents exhibited lower $R_e$ for

PMs and higher $R_e$ for coexisting metal ions, and thus, displaying significantly lower selectivity toward PMs (Fig. 5d–f). This result highlights the practically feasible, excellent PM recovery performance and selectivity of S-PAcH(L), which are attributable to its high reduction capability combined with its electrostatic and chelation interaction-mediated adsorption mechanism. Furthermore, S-PAcH(L) was reusable by desorbing PM from PM/S-PAcH(L) following a well-established regeneration protocol using PM desorption agents (thiourea, iron chloride (FeCl$_3$), and HCl)[60]. The $R_e$ value of S-PAcH(L) very slightly decreased with increasing the number of adsorption–desorption cycles (Fig. 5g–i); S-PAcH(L) underwent ~5% (for Au and Pd) and ~14% (for Pt) reductions in its $R_e$ after seven adsorption–desorption cycles, corresponding to ~0.7% (for Au and Pd) and ~2% (for Pt) reductions in $R_e$ per adsorption–desorption cycle. Compared with other reported PM adsorbents, S-PAcH(L) exhibited a relatively lower reduction in $R_e$

per adsorption–desorption cycle (Supplementary Table 11), confirming its excellent reusability. After PM desorption, desorbed PM ions can coexist with desorption agents, which need to be removed via additional separation processes to obtain high-purity PMs. $Fe^{3+}$ ions can be readily removed by adjusting the solution pH to 3–4, where Fe ions can be preferentially precipitated over PM ions (Supplementary Figs. 19 and 36 and Supplementary Note 2)[61]. Thiourea and HCl can also be removed by thermal treatment because they are completely vaporized (thiourea) and decomposed (HCl) at 300 °C[62]. We believe that PM recovery by our S-PAcH is cost-effective owing to the excellent PM adsorption capacity and selectivity of S-PAcH combined with its facile collection. The cost-effectiveness of S-PAcH can also be optimized by calcinating or regenerating it depending on the PM concentration (PM adsorption capacity) in feed solutions (Supplementary Table 12 and Supplementary Note 7).

## Discussion

In this study, a star-shaped, hydrazide-functionalized polymer (S-PAcH) was synthesized as a standalone PM adsorbent. The synthesized S-PAcH possessed densely packed arm chains with a high density of strongly reducible hydrazide groups. Compared with commercial amine polymers, reducing agents, and other laboratory-made PM adsorbents, S-PAcH exhibited significantly higher adsorption capacity and selectivity toward PMs with rapid adsorption kinetics. The superior PM recovery performance of S-PAcH can be attributed to its strong reduction ability combined with its effective chemisorption mechanism. Moreover, its star-shaped structure promoted intra/intermolecular chain fusion by enhancing PM reduction and facilitating multiple chain contacts, leading to the formation of large, mechanically strong precipitates that could be readily collected by membrane filtration. The collected PM/S-PAcH precipitates could be converted into high-purity PMs by calcination, directly used as catalysts for dye reduction, or regenerated for reuse. We successfully demonstrated that the star-shaped polymers can be used as high-capacity and selective PM adsorbents in a standalone manner by identifying their underlying PM adsorption mechanism. Our findings suggest that the rational tailoring of both the chemistry and physical architecture of polymers could enable the fabrication of high-performance adsorbents that can recover valuable resources or remove hazardous species in various applications, including battery and e-waste recycling, waste catalyst recovery, and wastewater treatment.

## Methods

### Materials

CDx (97.0%), methyl acrylate (99%), copper(I) bromide (99.9%), $N,N,N',N'',N''$-pentamethyldiethylenetriamine (99%), bromoisobutyryl bromide, ethyl $\alpha$-bromoisobutyrate (98%), 1-methyl-2-pyrrolidone (99.0%), tetra-$n$-butyl ammonium bromide (≥98.0%), hydrazine hydrate (50–60%), sodium carbonate (≥99.0%), copper nitrate trihydrate ($Cu(NO_3)_2 \cdot 3H_2O$, ≥99.0%), aluminum nitrate nonahydrate ($Al(NO_3)_3 \cdot 9H_2O$, ≥99.9%), nickel nitrate hexahydrate ($Ni(NO_3)_2 \cdot 6H_2O$, ≥99.9%), sodium nitrate ($NaNO_3$, ≥99.0%), magnesium nitrate hexahydrate ($Mg(NO_3)_2 \cdot 6H_2O$, ≥99.0%), potassium nitrate ($KNO_3$, ≥99.0%), calcium nitrate tetrahydrate ($Ca(NO_3)_2 \cdot 4H_2O$, ≥99.0%), nitric acid (70%), HCl (37%), aluminum oxide (alumina), $FeCl_3$ (≥99.9%), thiourea (≥99.0%), polychlorinated biphenyls (PCB No. 52), polybrominated diphenyl ethers (4,4′,6,6′-tetrabromo-2,2′-biphenol), polyaromatic hydrocarbons (Benzo[a]pyrene), and phthalate esters (dibutyl phthalate) were obtained from Sigma-Aldrich (USA). HCl (1 N) standard solution, sodium hydroxide (NaOH, 1 N) standard solution, dichloromethane (99.8%), methanol (99.9%), tetrahydrofuran (99.5%), and $NaBH_4$ (98%) were purchased from Daejung Chemical (South Korea). PAAm (molecular weight ($M_w$) = 150 kg mol$^{-1}$), bPEI ($M_w$ = 70 kg mol$^{-1}$), silica gel (silica), 4-NP (99%), and MO were procured from Alfa Aesar (USA). Natural organic matter (2R101N), fulvic acid (3S101F), and humic acid (3S101H) were purchased from International Humic Substances Society (USA). Furthermore, Au, Pd, and Pt (1000 mg L$^{-1}$) standard solutions were acquired from Kanto Chemical Co. (Japan). DI water was prepared using a Milli-Q purification system (Millipore, USA). Polysulfone (PSF) ultrafiltration membranes (M-M2540PS20, molecular weight cut-off = 20 kg mol$^{-1}$) and cellulose filter paper (JIS P 3801, pore size = 1 μm) were obtained from Applied Membranes Inc. (USA) and Advantec (Japan), respectively.

### Characterization

The chemical structures of the synthesized polymers were identified using proton nuclear magnetic resonance ($^1$H NMR, JNM-ECZ500R, JEOL, Japan) and Fourier-transform infrared (FT-IR, Spectrum Two spectrometer, PerkinElmer, USA) spectroscopy. The surface zeta potentials of the polymers were measured at different pH values (2–10) using a zeta potential analyzer (ELSZ-2000, Otsuka Electronics, Japan). The $H_R$ of the polymers was analyzed using dynamic light scattering (DLS, ELSZ-2000, Otsuka Electronics, Japan). The chemical structures of the polymers before and after PM adsorption were characterized using XPS (PHI 5000 VersaProbe, Ulvac-PHI, Japan) equipped with a monochromatic Al Kα X-ray source. The morphology of S-PAcH(L) before and after PM adsorption was examined using a high-resolution TEM (Tecnai F20, FEI, USA) operated at 200 kV. XRD (Rigaku Dmax 2500, Rigaku, Japan) was performed to analyze the crystallographic structures of the polymers before and after PM adsorption. An UV-vis spectrometer (Cary 5000, Agilent Technologies, USA) was employed to verify the formation of Au NPs in Au aqueous solutions to which S-PAcH had been added. The thermal degradation behavior of S-PAcH(L) and PM/S-PAcH(L) was analyzed using TGA (TGA Q500, TA Instrument, USA) by heating them from 25 to 800 °C under a $N_2$ environment at a ramping rate of 10 °C min$^{-1}$.

### Adsorption tests

PM stock solutions at predetermined PM ion concentrations were prepared by diluting the respective PM (1000 mg L$^{-1}$) standard solutions with DI water while adjusting their pH to 1–10 using 1 N HCl and NaOH aqueous solutions. Polymer adsorbents (10 mg) were added to the PM solutions (50 mL) and stirred at 200 rpm for 3 h. The mixture was then filtered through a PSF ultrafiltration membrane, and the supernatant was collected. The PM concentrations of the solutions obtained before ($C_i$) and after ($C_e$) the addition of the adsorbents were determined using an inductively coupled plasma optical emission spectrometer (ICP-OES, ICAP 7200, Thermo Scientific, USA). The adsorption capacity at equilibrium ($q_e$, mg g$^{-1}$), which represents the adsorbate mass (mg) per unit adsorbent mass (g), of the adsorbent was calculated using

$$q_e = \frac{(C_i - C_e) \times V}{M} \tag{1}$$

where $V$ is the solution volume and $M$ is the adsorbent mass.

The $R_e$ (%) was calculated using the following equation:

$$R_e = \frac{C_i - C_e}{C_i} \times 100 \tag{2}$$

At least three replicates were performed, and the results were averaged.

### Precipitation kinetics and stability analysis

The $H_R$ of the polymers was characterized to qualitatively analyze their adsorption degree and associated precipitation behavior[63].

Polymer adsorbents (10 mg) were added to the PM (200 mg L$^{-1}$) solutions (50 mL) at pH 2 and then stirred at 200 rpm. After a specific time (0–60 min), the $H_R$ of the precipitates in the solution was analyzed using DLS. To evaluate the mechanical stability of the precipitates, we added PAcH-series polymers (10 mg) to the PM (200 mg L$^{-1}$) solutions (50 mL) at pH 2 and stirred the mixtures at 200 rpm for 24 h, which induced precipitation. After static storage for 3 h, the mixture was stirred at a rotation speed of 0–200 rpm for 1 h, and the $H_R$ of the precipitates in the solution was measured using DLS.

### Adsorption selectivity analysis

The simulated leachates of a decommissioned computer central processing unit (CPU) (containing Au) and spent catalysts (containing Pd and Pt) were prepared by following a previously reported protocol[48]. Because CPU leachates mainly contain Cu (299 mg L$^{-1}$), Ni (17 mg L$^{-1}$), and Au (17 mg L$^{-1}$) ions, the simulated CPU leachate solution was prepared using a Au standard solution, Cu(NO$_3$)$_2$·3H$_2$O, and Ni(NO$_3$)$_2$·6H$_2$O. The simulated leachate of the spent alumina-supported Pd catalyst, which contained Pd (369.6 mg L$^{-1}$) and Al (330.4 mg L$^{-1}$) ions, was prepared using a Pd standard solution and Al(NO$_3$)$_3$·9H$_2$O. The simulated leachate of the spent Pt catalyst, which contained Pt (316.5 mg L$^{-1}$) and Al (375.1 mg L$^{-1}$) ions, was prepared using a Pt standard solution and Al(NO$_3$)$_3$·9H$_2$O. Simulated groundwater was also prepared by following a previously reported protocol[7]. Considering that groundwater typically contains Na, K, Mg, and Ca ions, the simulated solution, which contained each PM (1 mg L$^{-1}$) and these four metal (1000 mg L$^{-1}$ each) ions, was prepared using the respective PM standard solution, NaNO$_3$, KNO$_3$, Mg(NO$_3$)$_2$·6H$_2$O, and Ca(NO$_3$)$_2$·4H$_2$O. S-PAcH(L) and reducing agents (0.2 g L$^{-1}$) was added to the simulated solution at pH 2, stirred at 200 rpm for 3 h, and filtered through a PSF membrane. The metal ion concentrations of the permeate solution were measured using an ICP-OES.

### Refinement of PM/S-PAcH(L)

S-PAcH(L) (10 mg) was added to the PM (200 mg L$^{-1}$) solutions (50 mL) at pH 2 and then stirred at 200 rpm for 30 min. The precipitates formed in the solution were collected by PSF membrane filtration and vacuum-dried at room temperature for 24 h. The collected PM/S-PAcH(L) precipitates (50 mg) were loaded into a furnace and heated to 600 °C under an air-purged environment at a ramping rate of 30 °C min$^{-1}$. The temperature was maintained at 600 °C for 3 h to ensure calcination and then lowered to 25 °C. The calcined powder was transferred to a vial, to which a HCl (37%) aqueous solution (10 mL) was added, and the supernatant containing carbon ash was removed. The obtained solution was heated to 80 °C and stirred at 200 rpm for 24 h to enable the complete evaporation of the HCl solution. The resulting particles were washed three times with DI water to obtain the PM particles. The collected PM particles were dissolved in aqua regia, which is composed of a mixture of HCl (37%) and nitric acid (70%) at a volume ratio of 3:1, and their concentrations in the solution ($C$) were analyzed using an ICP-OES to determine their purity using the following equation:

$$\text{Purity (\%)} = \frac{m}{V_{\text{regia}} \times C} \times 100 \tag{3}$$

where $m$ is the mass of the PM particles and $V_{\text{regia}}$ is the volume of the aqua regia.

### Catalytic activity of PM/S-PAcH(L)

S-PAcH(L) (10 mg) was added to the PM (200 mg L$^{-1}$) solutions (50 mL) at pH 2 and then stirred at 200 rpm for 30 min. The precipitates formed in the solution were collected by PSF membrane filtration, vacuum-dried at room temperature, and then redispersed in DI water to obtain a PM/S-PAcH(L) (0.2 g L$^{-1}$)-containing solution. This solution (1.5 mL) was added to a dye (4-NP or MO, 0.02 mM)/NaBH$_4$ (2 mM)

aqueous solution (1.5 mL). Dye reduction was then monitored using an UV-vis spectrophotometer.

### PM recovery from the leachates of real-world samples

CPU (Au, Intel, USA) was obtained from an end-of-life computer, and spent catalysts (Pd and Pt, Sigma Aldrich, USA) were obtained after their use in hydrogenation reactions. Real-world leachate feed solutions were prepared by following a previously reported protocol[28]. Each real-world sample (20 g) was immersed in aqua regia (500 mL) for 3 d. The mixture was filtered through a cellulose filter paper to remove undissolved solids and further diluted to 1 L with DI water while adjusting pH to 2 using 1 N NaOH aqueous solution. Polymer adsorbents or reducing agents (10 mg) were added to each leachate solution (50 mL) and stirred at 200 rpm for 3 h. The mixture was then filtered through a PSF ultrafiltration membrane, and the supernatant was collected. The metal ion concentrations of the solutions obtained before and after the addition of the adsorbents and reducing agents were measured using an ICP-OES. To identify the effect of coexisting organic pollutants on PM recovery by PAcH(L), the total organic carbon (TOC) concentrations of the solutions obtained before and after the addition of PAcH(L) were measured using a TOC analyzer (TOC-L, Shimadzu, Japan). The TOC concentrations in the real-world leachate solutions were 1.4–2.1 mg L$^{-1}$, and metal composition can be found in Supplementary Fig. 34.

### Regeneration of PM/S-PAcH(L)

S-PAcH(L) (10 mg) was added to each of the above prepared real-word CPU and spent catalyst leachate solutions (50 mL) and stirred at 200 rpm for 3 h. The mixture was then filtered through a PSF ultrafiltration membrane, and the supernatant was collected. The PM ion concentrations of the solutions obtained before ($C_L$) and after the addition of S-PAcH(L) were measured using an ICP-OES to calculate $R_e$. The filtrated PM/S-PAcH(L) was put into the aqueous solution containing desorption agents (thiourea (1 M), FeCl$_3$ (1 M), and HCl (1 M)), and sonicated for 30 min to desorb PM species from PM/S-PAcH(L). The mixture was filtered through a PSF ultrafiltration membrane. Small thiourea (molecular weight = 76.12 g mol$^{-1}$) and ionic species (i.e., FeCl$_3$, HCl, and PM ions) permeated through the membrane, whereas large S-PAcH(L) (molecular weight = 318 kg mol$^{-1}$) was screened, enabling complete collection of S-PAcH(L). The PM concentration of the permeate solution ($C_D$) was measured using an ICP-OES to calculate the PM desorption efficiency ($D_e$) as given by:

$$D_e = \frac{C_D}{C_L \times R_e} \times 100 \tag{4}$$

The filtrated S-PAcH(L) was further washed with methanol to completely remove loosely bound thiourea and ionic species. Because S-PAcH(L) is marginally soluble in methanol to form clusters while thiourea and ionic species are highly soluble in methanol, the filtrated S-PAcH(L) was immersed in a methanol bath with stirring for 1 h followed by filtration through a cellulose filter paper. The complete removal of desorption agents and PM ions from S-PAcH(L) was confirmed by the XPS spectra of S-PAcH(L) before and after the regeneration process (Supplementary Fig. 37). The collected S-PAcH(L) was freeze-dried and then reused to repeat the above adsorption–desorption process seven times.

## Data availability

The data that support the findings of this study are available within the paper and its Supplementary Information/Source data file. Additional data are available from the corresponding author upon request. Source data are provided with this paper or obtained from Figshare repository at https://doi.org/10.6084/m9.figshare.25106891. Source data are provided with this paper.

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

## Acknowledgements

This research was supported by the National Research Foundation of Korea (NRF) grant funded by the Korean government (2023R1A2C2002913 (J.-H.L) and 2020M3H4A3106366 (J.-W.C)).

## Author contributions

J.-H.L. conceived the idea. J.-H.L. and J.-W.C. supervised the study and experiments. S.S.S. and Y.J. conducted the polymer synthesis, characterization, adsorption tests, and visualization. S.S.S., Y.J., S.J., S.-J.P., S.-J.Y, and K.-W.J. analyzed the experimental results. S.S.S. and Y.J. wrote the original manuscript. J.-H.L. and J.-W.C. reviewed and revised the manuscript. All the authors discussed the results and provided comments.

## Competing interests

The authors declare no competing interests.
