## [Peer Review File · Nature Communications]

REVIEWER COMMENTS

Reviewer #1 (Remarks to the Author):

This manuscript presents an interesting contribution in the field of 'Adsorptive Recovery of the Precious Metals (PMs)', with a high potential of making a significant impact. Particularly, the rapid precipitation of PM/S-PAcH, facilitating facile filtration-based removal, is a significant breakthrough. Having said that I would like the authors to address the below-mentioned concerns/comments.

1. The chemistry employed in this work has already been reported by Zhang et. al., (Mater. Chem. A, 2018, 6, 10217; cited in the manuscript as well as Supp Info). However, the smart design of the star-shaped water-soluble adsorbent presents the key in this current work to achieve unprecedented adsorption capacity and easy removal. Nevertheless, the unique attribute also makes the adsorbent limited as a single-use material in metal recovery applications. This is evident from the upcycling experiment, wherein the authors need to calcine the PM/S-PAcH assembly to obtain the pure metal. How will this be sustainable? The authors need to discuss this part explicitly. Moreover, as this aspect forms the key strong point of the work, it would be good if the authors present a cost analysis and show that the PM recovery through this approach would indeed be economically viable and potentially sustainable.

2. Line 84: Please specify the solubility limit.

3. Line 140: What is the proportion of the ionic-to-reduced metal species?

4. Line 168: The authors state that the PM species are unlikely to interact with the adsorbent at low initial concentration. This may not be correct, as the adsorption here is majorly assessed from the precipitation phenomenon, which occurs only above a certain threshold concentration, for obvious reasons.

5. Line 190: The authors state that the adsorption can be associated with the homogeneous monolayer formation. While at a later part in the same paragraph, they state that the adsorption is very rapid and hence reliable kinetics could not be followed. Nevertheless, the authors speculate that the initial chemisorption would be the rate-determining step. While this is agreeable, the correlation between this kinetics and nanoparticle growth is not clearly linked. A 3-D nanoparticle growth is obviously not a monolayer, but a reductive multilayer adsorption process should have taken place. Another important

associated question is the electron-transfer mechanism, leading to reductive adsorption. The authors state that the hydrazine moieties in the neighbouring PAcH chains reduce the PMs through electron transfer. If this is the case, would the degree of metal ion reduction depend on the amount (molarity) of hydrazine groups?

5. Line 241: Here, the reduction of 4-NP would be more appropriate than degradation, as no mineralization is happening here. Similar is the case with methyl orange.

6. Lines 453 & 458: 'R' to be subscript.

Reviewer #2 (Remarks to the Author):

This manuscript reports a method of Precious Metals recovery by Hydrazide-Functionalized Star-Shaped Polymers. Various characterization techniques were employed and lots of experiments had been done to verify the effectiveness of S-PAcH for the adsorption of PMs. Although some interesting results were present, I still believed that this study fails to provide enough new insight on the underlying working mechanism. Hence, this manuscript is not appropriate for Nature Communications.

Specific comments:

- 1, The explanation of the reaction mechanism is uncertain, the new insights need to be illustrated and the novelty need to be justified as well.
- 2, What is the reference for HR with adsorption capacity?
- 3, The redox reaction of precious metals requires overcoming the potential complexation with organic pollutants, but this lacks experimental evidence.
- 4, The adsorbent is incinerated to obtain high-purity precious metals, and the author should conduct economic evaluations, which are necessary for the sustainability aspect.
- 5, The lack of appropriate discussion that frames insights to existing concepts of PMs electro-sorption.
- 6, In the context of the title's reference to "upcycling," it is incumbent upon us to establish a clear and academically rigorous definition of this process as it pertains to the recovery of precious metals. Subsequently, we must formulate a robust methodology to substantiate and validate this definition. How do we define this process concerning the recovery of precious metals?
- 7, The highlights and innovation of this work are not enough in the whole manuscript.
- 8, In the context of experimental data pertaining to precious metal recovery, the inclusion of empirical findings from real-world samples would enhance the persuasiveness and credibility of the results.

- 9, Turnover frequency (TOF) and turnover number (TON) are critical performance metrics for evaluating precious metal catalysts. Have TON and TOF been calculated for the catalyst used for organic molecule removal? Do they have obvious advantages over other reported catalysts?
- 10, The TOC (Table of Content) graph is necessary to help readers better understand the highlights of the work if the journal requires it.
- 11, The authors have mentioned that “In this study, we develop, for the first time, a star-shaped, hydrazide-functionalized polymer (poly(acryloyl hydrazide), S-PACH).....” in terms of the star-shaped, hydrazide-functionalized materials, what is different between this work and doi.org/10.1016/j.cej.2022.137883 ? Please confirm whether it is the first time synthesis?
- 12, The level of rigor in academic paper writing currently falls short of the desired standards.
- 13, In metal reduction processes, such as the direct reduction of Au from trivalent to zero valence, without transitioning to a monovalent state, have the authors paid attention to the intermediate state?
- 14, In real waste solutions containing precious metals, various other ions are present, and the pH levels can vary significantly, including the presence of strong acids and bases. Have the authors taken these practical situations into consideration?

Reviewer #3 (Remarks to the Author):

In their work, the Choi and Lee research groups introduce two star-shaped polymers designed for the efficient recovery of precious metals, including gold, palladium, and platinum. These polymers are synthesized via ATRP polymerization using functionalized beta-cyclodextrin molecular skeletons. The incorporation of hydrazide groups onto the polymer chains grants them excellent reduction capabilities, a crucial factor in precipitating precious metals through reduction reactions. The authors evaluated the performance of their star-shaped polymers against other amino-based polymers and observed a significant improvement in adsorption capacity, selectivity, and kinetics for precious metal precipitation. While the reported findings are promising, the reviewer recommends against publication of this manuscript in Nature Communications due to the following concerns:

1. Mischaracterization of Polymer Function: The manuscript describes the star-shaped polymers as adsorbents, but they appear to function primarily as reducing agents. Comparing their efficacy with that of amino-based polymers, which lack reduction capabilities, is an inappropriate comparison. It would be more appropriate to compare these polymers with established reducing agents like hydrazine, particularly in terms of cost-effectiveness and efficiency.

2. Lack of Evidence for Polymer Recovery and Reusability: They claim that these polymers are recoverable adsorbents for precious metals is not substantiated by experimental data. The manuscript does not provide evidence to support the polymers' recovery or reusability, which is a crucial aspect of their proposed application.

3. Unclear Advantages Over Traditional Reductants: The manuscript does not convincingly demonstrate the advantages of using these star polymers over simpler, more conventional molecular reductants like hydrazine or NaBH₄. This comparison is vital for establishing the novelty and practical utility of the polymers.

4. Comparison with Other Precipitation Methods: The selective precipitation of anionic precious metal ions using small hydrophobic anionic species has been documented (Nat. Commu. 2021, 12, 6258). The manuscript fails to adequately compare the proposed polymers with these established methods, particularly in terms of efficiency and reusability.

5. Ineffectiveness at Realistic Metal Concentrations: The manuscript uses unrealistically high concentrations of metals (200 ppm) for the recovery simulations, which do not reflect real-world conditions found in CPU leachates (~20 ppm). Figure S5 suggests that these polymers are ineffective at lower, more realistic concentrations, raising significant concerns about their practical application.

Given these critical issues, I believe that the manuscript does not meet the high standards of scientific rigor and innovation required for publication in Nature Communications. Therefore, I must recommend the rejection of this manuscript.

Responses to the Reviewers' Comments

Reviewer #1

General comment: This manuscript presents an interesting contribution in the field of 'Adsorptive Recovery of the Precious Metals (PMs)', with a high potential of making a significant impact. Particularly, the rapid precipitation of PM/S-PAcH, facilitating facile filtration-based removal, is a significant breakthrough. Having said that I would like the authors to address the below-mentioned concerns/comments.

Response to General Comment: We greatly appreciate the reviewer for many valuable and insightful comments, which helped us significantly improve the scientific and practical impacts of our manuscript. We carefully addressed all the issues raised by the reviewer. Please refer to our responses to the reviewer comments as follows.

Comment 1-1: The chemistry employed in this work has already been reported by Zhang et al., (Mater. Chem. A, 2018, 6, 10217; cited in the manuscript as well as Supp Info). However, the smart design of the star-shaped water-soluble adsorbent presents the key in this current work to achieve unprecedented adsorption capacity and easy removal. Nevertheless, the unique attribute also makes the adsorbent limited as a single-use material in metal recovery applications. This is evident from the upcycling experiment, wherein the authors need to calcine the PM/S-PAcH assembly to obtain the pure metal. How will this be sustainable? The authors need to discuss this part explicitly. Moreover, as this aspect forms the key strong point of the work, it would be good if the authors present a cost analysis and show that the PM recovery through this approach would indeed be economically viable and potentially sustainable.

Response to Comment 1-1: We appreciate the reviewer's very insightful comment and useful suggestion. We totally agree with the reviewer that the sustainable and economical use of adsorbents is critical for their practical application. Let us respond to the reviewer's comments point by point as follows.

(1) Sustainable use (reusability) of our adsorbent

As mentioned in the manuscript, our proposed star-shaped hydrazide-functionalized polymer (S-PAcH) adsorbs PMs via reduction combined with electrostatic and chelation interactions. Careful reanalysis of its PM adsorption mechanism allowed us to reveal that the primary amine ($-NH_2$) groups of S-PAcH are oxidized to $-NO_2$ groups when they reduce PM ions to PM NPs, as appended at the end of our response to this comment.

To regenerate PM-adsorbed adsorbents by desorbing PMs, agents with high affinity for PMs, such as thiourea, cyanide, and thiocyanate, are commonly used [Ruan et al., *J. Phys. Chem. C* 121 (2017) 25882]. In our study, we used thiourea and Fe^{3+} to regenerate S-PAcH through PM desorption. Thiourea with high affinity for PMs facilitates the desorption of PM species (especially PM ions) from PM-adsorbed S-PAcH (PM/S-PAcH), while Fe^{3+} with high oxidation ability oxidizes adsorbed PM NPs to PM ions, thus assisting PM desorption [Zhou et al., *Environ. Sci. Technol.* 57 (2023) 3334]. Meanwhile, the reduced form of Fe^{3+} (Fe^{2+}) can reduce the $-NO_2$ groups (oxidized amine groups) of S-PAcH to primary amine ($-NH_2$) groups, thus regenerating S-PAcH [Hofstetter et al., *Environ. Sci. Technol.* 40 (2006) 235].

The reusability of our S-PAcH was assessed with real feed solutions, which are the leachates of real-world CPU and spent Pd and Pt catalyst samples. Experiments were performed as

follows. CPU (Au, Intel) was obtained from an end-of-life computer, and spent catalysts (Pd and Pt, Sigma Aldrich) were obtained after their use in hydrogenation reactions. Real-world leachate feed solutions were prepared by following a previously reported protocol [Zhang et al., *Sep. Purif. Technol.* 292 (2022) 121021]. Each real-world sample (20 g) was immersed in aqua regia (500 mL) for 3 d. The mixture was filtered through a cellulose filter paper (JIS P 3801, pore size = 1 μm , Advantec) to remove undissolved solids and further diluted to 1 L with DI water while adjusting pH to 2 using 1 N NaOH aqueous solution. The PM ion concentrations of the obtained leachate solutions were measured using ICP-OES. PM ion concentrations in the real-world leachates were 9.6 (Au), 10.8 (Pd), and 11.1 (Pt) mg L^{-1} , respectively. Next, S-PACH(L) (10 mg) was added to each leachate solution (50 mL) and stirred at 200 rpm for 3 h. The mixture was then filtered through a PSF ultrafiltration membrane, and the supernatant was collected. The PM ion concentration of the permeate solution was measured using an ICP-OES to calculate the PM recovery efficiency (R_e). The filtrated PM/S-PACH(L) was put into the aqueous solution containing thiourea (1M), FeCl_3 (1M), and HCl (1M) and sonicated for 30 min to desorb PM species from PM/S-PACH(L). The mixture was filtered through a PSF ultrafiltration membrane, and the PM concentration of the permeate solution (C_D) was measured using an ICP-OES to calculate the PM desorption efficiency (D_e), as given by:

$$D_e = \frac{C_D}{C_L \times R_e} \times 100 \quad (\text{Eq. R1})$$

where C_L is the PM ion concentration of the real-world leachate. The filtrated S-PACH(L) was freeze-dried and then used to repeat the above adsorption–desorption process seven times.

Fig. R1 shows the R_e and D_e values as a function of the number of adsorption–desorption cycles. The R_e value of S-PACH(L) very slightly decreased with increasing the number of adsorption–desorption cycles; S-PACH(L) underwent ~5% (for Au and Pd) and ~14% (for Pt) reductions in its R_e after seven adsorption–desorption cycles, corresponding to ~0.7% (for Au and Pd) and ~2% (for Pt) reductions in R_e per adsorption–desorption cycle. **Compared with other reported PM adsorbents, S-PACH(L) exhibited a lower reduction in R_e per adsorption–desorption cycle (Table R1), confirming its excellent reusability (sustainable use).**

Fig R1. Recovery (R_e) and desorption (D_e) efficiency of S-PACH(L) with real-world leachate solutions as a function of the number of adsorption–desorption cycles: (a) CPU leachate (Au), (b) spent Pd catalyst leachate, and (c) spent Pt catalyst leachate (S-PACH(L) concentration = 0.2 g L^{-1} , solution pH = 2, and adsorption time = 3 h).

Table R1. Reductions in R_e per adsorption–desorption cycle of reported PM adsorbents.

Adsorbent	PM	Reduction in R_e per adsorption–desorption cycle	Ref.
ADH@BC hybrid membrane ^a	Au	0.4	[1]
Poly-Cys-g-PDA@GPUF ^b	Au	2.5	[2]
2-Mercaptobenzothiazole-impregnated amine-functionalized resin	Pd	4.8	[3]
AHPP-MOF ^c	Pd	2.1	[4]
MNP-G3 ^d	Pd	1.7	[5]
Poly(allylamine hydrochloride)-modified E. coli	Pt	4.4	[6]
	Au	0.7	
S-PAcH(L)	Pd	0.7	This study
	Pt	2	

^aAdipic dihydrazide-grafted bacterial cellulose hybrid membrane. ^bCysteine polymer brush-grafted polydopamine-modified graphene-based polyurethane foam. ^c4-amino-3-hydroxybenzoic acid-modified Zr-based metal-organic framework. ^dMagnetic nanoparticle modified by third-generation dendrimer. ([1] Zhang et al., *Sep. Purif. Technol.* 292 (2022) 121021; [2] Xue et al., *React. Funct. Polym.* 136 (2019) 138; [3] Sharma and Rajesh, *Chem. Eng. J.* 283 (2016) 999; [4] Tang et al., *Chem. Eng. J.* 407 (2021) 127223; [5] Yen et al., *J. Hazard. Mater.* 322 (2017) 215; [6] Mao et al., *Water Res.* 44 (2010) 5919)

[Reanalyzed PM adsorption mechanism of S-PAcH]

To identify the reliable PM adsorption mechanism of S-PAcH, we carefully considered any factors to cause artifacts. We suspected that water in the atmosphere could be readily adsorbed on highly hydrophilic S-PAcH, possibly leading to misinterpretation of XPS data. Specifically, we speculated that the original broad O1s XPS spectrum of the PM/S-PAcH at ~533 eV (Fig. 2g and Supplementary Fig. 15) could be interfered with the adsorbed water whose O1s peak strongly appears at 533.1 eV [Heine et al., *J. Am. Chem. Soc.* 138 (2016) 13246].

To avoid the possible interference of adsorbed water, we performed the XPS analysis of PM/S-PAcH immediately after the sample was vacuum-dried at 50 °C for 48 h. As shown in **Fig. R2**, the reanalyzed O1s XPS spectrum of PM/S-PAcH exhibited the narrower peak at a lower binding energy without displaying the peak at ~533 eV compared with the original counterpart. This result clearly confirms that our previous XPS spectra were contaminated with adsorbed water. Hence, the XPS spectra of PM/S-PAcH were reanalyzed with the sample immediately after vacuum drying.

Fig. R2. Original (top) and reanalyzed (bottom) high-resolution O1s XPS spectra of the PM/S-PAcH(L) precipitates: (a) Au, (b) Pd and (c) Pt.

Fig. R3 shows the reanalyzed N1s, O1s, and C1s XPS spectra of PM/S-PAcH. The N1s XPS peak of PM/S-PAcH was deconvoluted into three peaks at 399.7 ($-\text{NO}_2$), 400.5 (**N-metal-N**), and 401.9 (**protonated amine**) eV (**Figs. R3a-c**), which were absent for pristine S-PAcH (original Supplementary Fig. 2a) [Zhang et al., *Sep. Purif. Technol.* 292 (2022) 121021]. Compared with the original spectra, the reanalyzed N1s spectra of PM/S-PAcH showed a new peak at 399.7 eV ($-\text{NO}_2$) without displaying the peak at 399.5 eV (N-metal-O). This result indicates that (1) protonated amine and $-\text{NO}_2$ groups and N-metal-N chelation bonding are formed in PM/S-PAcH after PM adsorption.

The deconvolution of the O1s spectra was not informative because both peaks corresponding to $-\text{NO}_2$ and C-O-C were overlapped at 532.3 eV (**Figs. R3d-f**) [Luo et al., *Adv. Mater.* 30 (2018) 1706498]. Hence, we focused on the C1s spectrum, which was deconvoluted to two peaks at 284.8 (C-C) and 288.0 (O=C-N) eV for both pristine S-PAcH and PM/S-PAcH (**Figs. R3g-i and R4**) [Zhang et al., *Sep. Purif. Technol.* 292 (2022) 121021]. This result suggests that (2) the oxygen atom in the carbonyl (C=O) group (carbonyl oxygen) of S-PAcH is not protonated in PM/S-PAcH after PM adsorption, unlike our original interpretation from the O1s spectra that protonation on the carbonyl oxygen ($\text{C}=\text{OH}^+$) occurs.

Fig. R3. Deconvolution of the high-resolution (a–c) N1s, (d–f) O1s, and (g–i) C1s XPS spectra of the PM/S-PAcH(L) precipitates: (a, d, g) Au, (b, e, h) Pd and (c, f, i) Pt.

Fig. R4. Deconvolution of the high-resolution C1s XPS peaks of S-PAcH.

Based on the reanalyzed XPS data, we carefully identified the PM adsorption mechanism of S-PAcH at low pH, as illustrated in **Fig. R5**. The primary amine (–NH₂) groups of S-PAcH are

protonated preferentially over its secondary amine ($-\text{NH}-$) groups under acidic conditions owing to their stronger basicity (*i.e.*, higher electron-donating ability) [Smith, *Organic chemistry 4th edition*, McGraw-Hill (2014)]. S-PAcH with protonated amine ($-\text{NH}_3^+$) groups adsorbs anionic PM species (*i.e.*, AuCl_4^- , PdCl_4^{2-} , and PtCl_6^{2-}) via long-range electrostatic interaction [Lin et al., *J. Mater. Chem. A* 5 (2017) 13557], followed by ion-exchange and chelation mainly with the unshared electron-bearing nitrogen atoms of its unprotonated amine ($-\text{NH}-$) groups (**Fig. R5a**) [Yang et al., *J. Mater. Chem. A* 8 (2020) 3438; Ain et al., *Spectrochim. Acta, Part A* 115 (2013) 683]. Meanwhile, S-PAcH molecules coagulate owing to their screened electrostatic charges. Subsequently, the $-\text{NH}_3^+$ groups of S-PAcH are deprotonated to $-\text{NH}_2$ while protonating the adsorbed PM ions via the acid–base reaction between the $-\text{NH}_3^+$ groups (acid) and PM ions (base) (Fig. R5b) [Zhang et al., *Sep. Purif. Technol.* 292 (2022) 121021]. The hydrazide groups of S-PAcH then reduce adsorbed PM ions to NPs while their $-\text{NH}_2$ groups being converted into $-\text{NO}_2$ (Figs. R5b and c) [Zhang et al., *Sep. Purif. Technol.* 292 (2022) 121021]. Continuous PM reduction leads to NP growth and induces intra/intermolecular chain fusion through chelation (N–metal–N) between the NPs and unshared electron-bearing nitrogen atoms of $-\text{NH}-$ groups in neighboring PAcH chains, leading to the rapid formation of large and robust precipitates (Fig. R5c). This mechanism is consistent with the presence of XPS spectra corresponding to $-\text{NO}_2$ and N–metal–N for PM/S-PAcH. Meanwhile, a small fraction of the adsorbed PM species exists as an ionic state in PM/S-PAcH via electrostatic and chelation interaction (Fig. R5c), as evidenced by the presence of the PM ionic XPS peak (original Fig. 2e and Supplementary Fig. 11) and N1s XPS peak corresponding to the protonated amine (**Figs. R3a–c**) for PM/S-PAcH.

Fig. R5. Revised PM adsorption mechanism of S-PAcH

(2) Sustainability and economic viability

Strictly speaking, the sustainability of the adsorption process needs to be assessed based on its whole cyclic process, including use and regeneration. We also need to consider environmental impact and social acceptance together with economic viability [Büyükoçkan and Karabulut, *J. Environ. Manage.* 217 (2018) 253]. Even if any PM adsorbent is reusable, it cannot be regarded as “completely sustainable or environmentally sustainable” if environmentally hazardous reagents, such as thiourea (cytotoxic), HCl (corrosive), and FeCl_3 (oxidative) are used for the regeneration of the adsorbent [Sharma et al., *Environ. Sci. Technol.* 33 (1999) 2645; Akcil et

al., *Waste Manage.* 45 (2015) 258]. In this meaning, the calcination process is also not sustainable because it emits gas (e.g., CO₂) waste and does not allow for the reuse of the adsorbent. As demonstrated above and in the original manuscript, our S-PAcH polymer can be either regenerated for reuse or calcinated for PM recovery. **Although S-PAcH cannot achieve “complete sustainability”, it can be sustainably reused at least.**

To evaluate the economic viability, we first estimated the price of S-PAcH. In fact, it is very difficult to accurately estimate the price of S-PAcH, which is a lab-scale sample, because the accurate information of additional production costs (e.g., facilities, plant planning, labor, etc.) and the industrial-grade prices of raw materials is not available for us, who are academic field researchers. Nevertheless, we made our best efforts to evaluate the final price of S-PAcH based on the reagent-grade prices of the raw materials required for its lab-scale synthesis.

Table R2 summarizes the amounts and unit prices of the raw materials used for synthesizing S-PAcH(L) of 1 kg. The pure price of S-PAcH(L) calculated based on its raw material prices was ~\$6961.3 kg⁻¹. If we assume that raw material costs occupy approximately 60–90% of the total production cost [Meneses et al., *Heliyon* 8 (2022) e09028], the final price of S-PAcH(L) can be reasonably estimated to be \$7734.8–11602.2 kg⁻¹. However, this calculation is a very rough estimation, and the accurate total price should be estimated by manufacturers.

Table R2. Unit prices, used amounts, and respective costs of the raw materials used for synthesizing S-PAcH(L) of 1 kg.

Raw material	Manufacturer	Unit price	Used amount	Cost
CDx	Sigma-Aldrich	\$997.4 kg ⁻¹	3.6 g	\$3.6
BiBr	Sigma-Aldrich	\$409.4 kg ⁻¹	24.5 g	\$10.0
CuBr	Sigma-Aldrich	\$3401 kg ⁻¹	8.3 g	\$28.2
MAc	Sigma-Aldrich	\$52.8 L ⁻¹	4.6 L	\$242.8
Alumina	Sigma-Aldrich	\$156.5 kg ⁻¹	1.3 kg	\$195.6
TBABr	Sigma-Aldrich	\$1706.8 kg ⁻¹	1.0 kg	\$1706.8
Hydrazine hydrate	Sigma-Aldrich	\$379.8 kg ⁻¹	7.6 kg	\$2886.5
Na ₂ CO ₃	Daejung Chemical	\$141.4 kg ⁻¹	46.1 g	\$6.5
DCM	Daejung Chemical	\$6.8 L ⁻¹	214.3 mL	\$1.5
THF	Daejung Chemical	\$21.2 L ⁻¹	30.0 L	\$634.6
Methanol	Daejung Chemical	\$7.2 L ⁻¹	162.5 L	\$1175
NMP	Daejung Chemical	\$89.7 L ⁻¹	29.1 mL	\$2.6
Silica	Alfa Aesar	\$53.8 kg ⁻¹	1.3 kg	\$67.3
DI water	Millipore	\$1.5 L ⁻¹	214.3 mL	\$0.3
Total cost:				\$6961.3 kg ⁻¹
Expected final price ^a :				\$7734.8 –11602.2 kg ⁻¹

^aEstimated with the assumption that raw material costs occupy approximately 60–90% of the total production cost [Meneses et al., *Heliyon* 8 (2022) e09028].

We then calculated the price (value) ratio of adsorbed PM to S-PAcH(L) in PM/S-PAcH(L)s formed with different PM concentrations based on the weight and price of adsorbed PM and S-PAcH(L), as summarized in **Table R3**. Two PM concentrations, (1) 200 mg L⁻¹, which is the standard PM concentration examined in the manuscript, and (2) PM concentrations in the above-prepared leachates of real-world CPU and spent Pd and Pt catalyst samples, were considered.

As shown in **Table R3**, at the PM concentration of 200 mg L⁻¹, the price of adsorbed (recovered) PM was 1.9 to 8.5 times higher than that of S-PAcH(L). Although more extensive cost analysis is needed, in this case where the value of recovered PM exceeds that of the S-PAcH(L) adsorbent owing to high PM adsorption capacity, simple calcination for PM recovery could be more cost-effective than a relatively complex regeneration process [Biswas et al., *Chem. Eng. J.* 407 (2021) 127225; Yang et al., *Carbohydr. Polym.* 111 (2014) 768]. Conversely, in the case of real-world leachates with low PM concentrations (~10 mg L⁻¹), the price of S-PAcH(L) was higher than that of adsorbed PM. In this case, the regeneration (reuse) of S-PAcH(L) could be more cost-effective than calcination which results in the loss of S-PAcH(L).

Table R3. Weight and cost ratios of adsorbed PM to S-PAcH(L) in PM/S-PAcH(L)s formed with different PM concentrations.

PM concentration	PM	Unit price of PM	Weight ratio of adsorbed PM to S-PAcH(L)	Price ratio of adsorbed PM to S-PAcH(L)
Model solution PM: 200 mg L ⁻¹	Au	\$65819 kg ⁻¹	1	5.7–8.5
	Pd	\$33256 kg ⁻¹	1	2.9–4.3
	Pt	\$31195 kg ⁻¹	0.71	1.9–2.9
Real-world leachate Au: 9.6 mg L ⁻¹ Pd: 10.8 mg L ⁻¹ Pt: 11.1 mg L ⁻¹	Au	\$65819 kg ⁻¹	0.048	0.27–0.41
	Pd	\$33256 kg ⁻¹	0.054	0.16–0.23
	Pt	\$31195 kg ⁻¹	0.056	0.15–0.22

Furthermore, compared with commercial amine polymer adsorbents (bPEI and PAAM), S-PAcH(L) exhibited significantly higher PM adsorption capacity and selectivity with high recoverability by forming reduction-induced large and strong precipitate, as demonstrated in the original manuscript. Hence, it is evident that these key advantages of S-PAcH(L) over commercial amine polymers enable more cost-effective PM recovery by considerably reducing separation processes.

We also compared the cost effectiveness of S-PAcH(L) with that of conventional reducing agents (hydrazine, and NaBH₄). To fairly assess the cost effectiveness of S-PAcH(L), hydrazine, and NaBH₄, their costs required for recovering the same amount of PM were compared. Hence, we first identified the doses of the reducing agents that could achieve the same R_e as that achieved by the dose (0.2 g L⁻¹) of S-PAcH(L) at a given PM ion concentration (200 mg L⁻¹), as shown in **Fig. R6**.

Although both reducing agents exhibited lower R_e for all PMs compared with S-PAcH(L) at the same dose (0.2 g L⁻¹), their R_e values increased with their doses, except for hydrazine whose R_e for Pt remained marginal (~0%) even at the highest dose investigated (5.0 g L⁻¹) owing to its very low Pt reduction capability [Wu et al., *Angew. Chem. Int. Ed.* 60 (2021) 17587].

Fig. R6. Recovery efficiency (R_e) of S-PAcH(L) (0.2 g L^{-1}) and reducing agents (hydrazine and NaBH_4) for PMs as a function of reducing agent doses (initial PM ion concentration = 200 mg L^{-1} , solution pH = 2, contact time = 3 h).

Based on the prices of materials and their doses required for achieving the same R_e at a given PM ion concentration of 200 mg L^{-1} (determined from **Fig. R6**), we calculated their costs for PM recovery, as summarized in **Table R4**. The material price of S-PAcH(L) was notably higher than those of hydrazine and NaBH_4 . Nevertheless, the cost of S-PAcH(L) for PM recovery was similar or even lower compared with hydrazine and slightly higher compared with NaBH_4 owing to the superior PM adsorption capacity of S-PAcH(L). We also need to consider the recoverability of adsorbents and reducing agents to assess their cost effectiveness for practical PM recovery. The small molecular size of reducing agents (unreacted and/or reacted forms) renders their recovery/collection difficult [Chen et al., *Chemosphere* 49 (2002) 363], which requires additional costs for purification/recovery processes. Particularly, in a PM reduction process using NaBH_4 , toxic impurities, such as boric acid (H_3BO_3) and sodium metaborate (NaBO_2), are generated, which should be removed by employing additional separation processes [Chen et al., *Chemosphere* 49 (2002) 363]. Hence, we believe that our S-PAcH polymers with higher PM adsorption capacity and recoverability would be more cost-effective compared with reducing agents.

Table R4. Prices, required doses, and respective costs of the materials used for PM recovery.

Materials	Manufacturer	Material price (\$ kg^{-1})	Dose (g L^{-1})			Cost (\$ L^{-1})		
			Au	Pd	Pt	Au	Pd	Pt
S-PAcH(L)	Lab-made	7734.8–11602.2	0.2	0.2	0.2	1.55–2.32	1.55–2.32	1.55–2.32
Hydrazine	Sigma-Aldrich	379.8	5.0	>5.0	>>5.0	1.90	>1.90	>>1.90
NaBH_4	Daejung Chemical	157.7	>5.0	>5.0	0.36	>0.79	>0.79	0.06

We further compared the PM R_e and selectivity of S-PAcH(L) with those of commercial amine polymers and reducing agents with real-world leachates. The real-world CPU and spent Pd/Pt catalyst leachate solutions contained various coexisting ions; the CPU leachate mainly contained Au (9.6 mg L^{-1}), Ni (84.5 mg L^{-1}), and Cu (102.9 mg L^{-1}) ions while spent Pd and Pt catalysts contained Al (2.9 and 3.7 mg L^{-1} , respectively) as a main coexisting metal ion

together with Pd (10.8 mg L⁻¹) and Pt (11.1 mg L⁻¹), respectively.

Fig. R7 showed that for all the real-world leachates, S-PACH(L) exhibited ~100% R_e for all PMs but negligible R_e for other coexisting metal cations, similar to the case for simulated feed solutions, as demonstrated in the original manuscript. This result highlights the excellent selectivity of S-PACH(L) toward all PMs. Compared with S-PACH(L), commercial amine polymers (bPEI and PAAM) and reducing agents (hydrazine and NaBH₄) exhibited lower R_e for PMs and higher R_e for coexisting metal ions, and thus, significantly lower selectivity toward PMs. This result demonstrates that S-PACH(L) has significantly higher PM adsorption capacity and selectivity compared with commercial amine polymers and reducing agents even with real-world leachate feed solutions.

Fig. R7. Recovery efficiency (R_e) of the polymers (S-PACH(L), bPEI, and PAAM) and reducing agents (hydrazine and NaBH₄) with the real-world CPU and spent Pd and Pt catalyst leachate solutions: (a) CPU leachate, (b) spent Pd catalyst leachate, and (c) spent Pt catalyst leachate. (polymer and reducing agent concentration = 0.2 g L⁻¹, solution pH = 2, contact time = 3 h).

The accurate economic analysis of the S-PACH-based PM recovery process is not possible in this study owing to the lack of the cost information of each unit process. Nevertheless, we believe that our S-PACH polymers with higher PM adsorption capacity/selectivity and recoverability would be more cost-effective compared with commercial amine polymers and reducing agents, based on the above considerations. The cost-effectiveness of S-PACH can also be optimized by calcinating or regenerating it depending on the PM concentration (PM adsorption capacity) in feed solutions.

In the revised manuscript, we made the following corrections.

(1) We included the protocol of regeneration experiments in the “Methods” section (Lines 467-480).

(2) We included the results and discussion regarding the reusability of our S-PACH adsorbent as follows, “Furthermore, S-PACH(L) was reusable by desorbing PM from PM/S-PACH(L) following a well-established regeneration protocol⁵⁷. The R_e value of S-PACH(L) very slightly decreased with increasing the number of adsorption–desorption cycles (Figs. 5g–i); S-PACH(L) underwent ~5% (for Au and Pd) and ~14% (for Pt) reductions in its R_e after seven adsorption–desorption cycles, corresponding to ~0.7% (for Au and Pd) and ~2% (for Pt) reductions in R_e per adsorption–desorption cycle. Compared with other reported PM adsorbents, S-PACH(L) exhibited a lower reduction in R_e per adsorption–desorption cycle (Supplementary Table 11), confirming its excellent reusability.” (Lines 333-340), together with providing the above Fig. R1 and Table R1 as new Figs. 5g–i and Supplementary Table 11.

(3) We revised the PM adsorption mechanism of S-PAcH as follows, “The N1s XPS peak of PM/S-PAcH was broader than that of the pristine S-PAcH and deconvoluted into three peaks at 399.7 (–NO₂), 400.5 (N–metal–N), and 401.9 (protonated amine) eV²⁰, which were absent for S-PAcH (Fig. 2f and Supplementary Figs. 2 and 16). Deconvolution of the C1s peak revealed two peaks at 284.8 (C–C) and 288.0 (O=C–N) eV for both S-PAcH and PM/S-PAcH (Fig. 2g and Supplementary Fig. 17)²⁸. These results suggest that protonated amines, –NO₂ groups, and N–metal–N chelation bonding are formed while carbonyl oxygen atoms remaining unprotonated in PM/S-PAcH after PM adsorption. Given the results above, the PM adsorption mechanism of S-PAcH at low pH can be depicted as illustrated in Figs. 2h–j. The primary amines (–NH₂) of S-PAcH are protonated preferentially over its secondary amines (–NH–) under acidic conditions owing to their higher basicity (*i.e.*, electron donating nature)³³. S-PAcH with protonated primary amines (–NH₃⁺) adsorbs anionic PM species (*i.e.*, AuCl₄[–], PdCl₄^{2–}, and PtCl₆^{2–}) via long-range electrostatic interactions^{30,40}, followed by ion-exchange and chelation mainly with the unshared electron-bearing nitrogen atoms of its unprotonated –NH– (Fig. 2h)^{14,41}. Meanwhile, S-PAcH molecules coagulate owing to their screened electrostatic charges. Subsequently, –NH₃⁺ of S-PAcH are deprotonated to –NH₂ while protonating the adsorbed PM ions via the acid (–NH₃⁺)–base (PM ion) reaction (Fig. 2i)²⁸. The hydrazide groups of S-PAcH then reduce the adsorbed PM ions to NPs while their –NH₂ being converted into –NO₂ (Figs. 2i and j)²⁸. Continuous PM reduction leads to NP growth and induces intra/intermolecular chain fusion through chelation (N–metal–N) between the NPs and unshared electron-bearing nitrogen atoms of –NH– in neighboring PAcH chains, leading to the rapid formation of large and robust precipitates (Fig. 2j). A small fraction of the adsorbed PM species exists as an ionic state in PM/S-PAcH via ion electrostatic and chelation interaction (Fig. 2j), as evidenced by the ionic PM and N1s (corresponding to the protonated amine) XPS peaks detected for PM/S-PAcH.” (Lines 156-180).

(4) We replaced original Figs. 2f–j with the above Figs. R3a and g and R5 with providing the above Fig. R4 as a new Supplementary Fig. 2c. We also replaced original Supplementary Figs. S14 and S15 with the above Figs. R3b, c, h, and i.

(5) We included the comment on the cost effectiveness of our S-PAcH(L) compared with reducing agents as follows, “Moreover, considering the difficulty in recovering small molecular-sized reducing agents⁵¹, we believe that our PAcH with higher PM adsorption performance and recoverability would be more cost-effective at recovering PMs compared with reducing agents (Supplementary Fig. 29, Supplementary Tables 7 and 8, and Supplementary Text).” (Lines 265-269), together with providing the above Fig. R6, Tables R2 and R4, and related explanations as new Supplementary Fig. 29, Supplementary Tables 7 and 8, and Supplementary text.

(6) We also included the comment related with the cost effectiveness of our S-PAcH(L) as follows, “We further assessed the feasibility of S-PAcH for its use in PM recovery from the leachates of real-world CPU and spent Pd/Pt catalysts (Figs. 5a–c and Supplementary Fig. 33). S-PAcH(L) completely recovered PMs (*i.e.*, ~100% R_e) from all the real-word leachate solutions without absorbing coexisting metal ions (Figs. 5d–f) and organic pollutants (Supplementary Fig. 34). Compared with S-PAcH(L), commercial amine polymers and reducing agents exhibited lower R_e for PMs and higher R_e for coexisting metal ions, and thus, displaying significantly lower selectivity toward PMs (Figs. 5d–f). This result highlights the practically feasible, excellent PM recovery performance and selectivity of S-PAcH(L), which are attributable to its high reduction capability combined

with its electrostatic and chelation interaction-mediated adsorption mechanism.” (Lines 324-333), together proving the above Fig. R7 as a new Figs. 5d–f.

(7) We further highlighted the cost effectiveness of our S-PAcH(L) as follows, “We believe that PM recovery by our S-PAcH is cost-effective owing to the excellent PM adsorption capacity and selectivity of S-PAcH combined with its facile collection. The cost-effectiveness of S-PAcH can also be optimized by calcinating or regenerating it depending on the PM concentration (PM adsorption capacity) in feed solutions (Supplementary Table 12 and Supplementary Text).” (Lines 341-345), together with providing the above Table R3 and related explanations as new Supplementary Table 12 and Supplementary text.

Comment 1-2: Line 84: Please specify the solubility limit.

Response to Comment 1-2: Thank you for the valuable suggestion. We determined the solubility limit of S-PAcH based on the visual transparency and hydrodynamic diameter (H_R) (determined by dynamic light scattering (DLS)) of S-PAcH(L) aqueous solutions at different S-PAcH(L) concentrations (Fig. R8). At the S-PAcH(L) concentration of ≤ 10 wt.%, the S-PAcH(L) solution was transparent, and the H_R of S-PAcH(L) was minimal and constant, indicating complete dissolution of S-PAcH(L) in water. In contrast, at the S-PAcH(L) concentration of >10 wt.%, the S-PAcH(L) solution appeared turbid, and the H_R of S-PAcH(L) increased drastically with the S-PAcH(L) concentration, indicating incomplete dissolution of S-PAcH(L) in water. Based on this experiment, we can conclude that the solubility limit of S-PAcH in water is 10 wt.%. Considering that the dose of the S-PAcH was 0.2 g L^{-1} (approximately 0.02 wt.%) in this study, the solubility limit of the S-PAcH is sufficiently high for practical PM recovery.

Fig. R8. (a) Photographs and (b) hydrodynamic diameter (H_R) of the S-PAcH(L) aqueous solutions at different S-PAcH(L) concentrations.

In the revised manuscript, we specified the solubility limit as follows, “S-PAcH possessed a spherical morphology with a diameter of approximately 9 nm, exhibiting excellent solubility in water (*i.e.*, the solubility limit of ~ 10 wt.%) (Fig. 1b and Supplementary Fig. 5).” (lines 91-93), together with providing the above Fig. R8 as a new Supplementary Fig. 5.

Comment 1-3: Line 140: What is the proportion of the ionic-to-reduced metal species?

Response to Comment 1-3: Thank you for the valuable comment. As the reviewer suggested, we determined the proportion of ionic-to-reduced metal species from the deconvoluted PM XPS peaks shown in original Fig. 2e and Supplementary Fig. S11. The ratio of peak areas can be correlated with the realistic molecular ratio of ionic to reduced PM(0) species.

Table R5 summarizes the binding energies and area percentages of the deconvoluted PM XPS peaks (Au4f, Pd3d, and Pt4f) of PM/S-PAcH(L) after PM recovery. It can be found from **Table R5** that approximately 88.8% of Au ions, 86.7% of Pd ions, and 80.6% of Pt ions were reduced to metal states (PM(0)). This result supports the high reduction capability of PM/S-PAcH(L), which facilitates easy recovery after use by inducing the formation of a large and strong complex with PM. The varying reduction ratios may be attributed to the intrinsic reduction potential and chemical affinity for each PM.

In contrast, no reduced PM(0) peak was observed for bPEI and PAAm after PM recovery, indicating that PM ions were not reduced by bPEI and PAAm, as mentioned in the original manuscript (Lines 141-142, Supplementary Figs. 12 and 13). This result suggests that commercial amine polymers adsorb PM ions by mainly through electrostatic and chelation interactions, not reduction, owing to their lack of reduction ability.

Table R5. Binding energies and area percentages of the deconvoluted PM XPS peaks (Au4f, Pd3d, and Pt4d) of the PM/S-PAcH(L) precipitates.

Sample	Peak	Binding energy (eV)	Peak area percentage (%)	Peak assignment
Au/S-PAcH(L)	Au4f	84.0/87.8	88.8	Au(0)
		83.6/87.2	11.2	Au(III)
Pd/S-PAcH(L)	Pd3d	336.4/342.0	86.7	Pd(0)
		338.1/343.5	13.3	Pd(II)
Pd/S-PAcH(L)	Pt4f	72.1/75.5	80.6	Pt(0)
		73.2/76.7	19.4	Pt(IV)

In the revised manuscript, we included the proportion of ionic-to-reduced metal species as follows, “Moreover, X-ray photoelectron spectroscopy (XPS) revealed that the PM/S-PAcH precipitates exhibited two deconvoluted PM peaks corresponding to ionic and reduced PM (PM(0)) metal states with a high fraction of the metal state (80–89%) (Fig. 2e, Supplementary Fig. 12 and Supplementary Table 2)” (Lines 147-151), together with providing the above Table R5 as a new Supplementary Table 2.

Comment 1-4: Line 168: The authors state that the PM species are unlikely to interact with the adsorbent at low initial concentration. This may not be correct, as the adsorption here is majorly assessed from the precipitation phenomenon, which occurs only above a certain threshold concentration, for obvious reasons.

Response to Comment 1-4: We appreciate the reviewer for the insightful comment. First of all, we apologize for causing confusion. Let us clarify our claim. It can be seen from original Figs. 3a–c that the R_e of commercial amine polymers for all PMs is low at low initial PM ion concentrations (C_i , $<50 \text{ mg L}^{-1}$). We speculated that commercial amine polymers, whose PM adsorption is mainly governed by electrostatic and chelation interactions, are likely to lead to low R_e at low C_i ($<50 \text{ mg L}^{-1}$) owing to their reduced interaction probability with PM ions.

By contrast, our S-PAcH polymer, in particular S-PAcH(L), exhibited very high R_e ($\sim 100\%$)

even at low C_i ($<50 \text{ mg L}^{-1}$, lower than a threshold concentration to induce significant precipitation), in which microscale precipitation was not induced (original Fig. S5). We attributed the superior PM adsorption capability of S-PAcH(L) to its strong reduction capability combined with its electrostatic and chelating adsorption mechanisms, as mentioned in the original manuscript (Lines 206-209).

To verify our claim, we closely examined the PM adsorption behavior of S-PAcH(L) at very low C_i (1 mg L^{-1}), as shown in Fig. R9. Although microscale precipitates were not formed after the addition of S-PAcH(L) to the PM solutions at the very low C_i (Figs. R9a–c), a distinct increase in the hydrodynamic diameter (H_R) of S-PAcH(L) was observed (Fig. R9d). Moreover, the S-PAcH(L)-added Au solution exhibited the characteristic red color of Au NPs, indicating PM reduction by S-PAcH(L). Despite their small sizes, PM/S-PAcH(L) particles were completely screened *via* a PSF membrane ($\text{MWCO} = 20 \text{ kg mol}^{-1}$), as evidenced by PM/S-PAcH(L) particles collected by the membrane (Figs. R9e–g) and clear permeate solutions (Figs. R9h–j). This result clearly demonstrates that even without microscale precipitation, a strong reduction mechanism enabled by S-PAcH(L) can achieve high R_e for all PMs at low PM ion concentrations ($<50 \text{ mg L}^{-1}$).

Fig. R9. a–c, Photographs of the (a) Au, (b) Pd, and (c) Pt aqueous solutions after the addition of S-PAcH(L) (S-PAcH(L) concentration = 0.2 g L^{-1} , initial PM ion concentration (C_i) = 1 mg L^{-1} , solution pH = 2, contact time = 3 h). d, Corresponding DLS curves of pristine S-PAcH(L) and PM/S-PAcH(L). e–j, Photographs of the (e–g) PM/S-PAcH(L) particles screened by a PSF membrane and (h–j) permeate solutions: (e, h) Au, (f, i) Pd, and (g, j) Pt.

In the revised manuscript, we corrected our claim as follows, “Commercial amine polymers achieved the maximum R_e for all PMs at a certain initial PM ion concentration (C_i) (Figs. 3a–c) because their interaction probability with PM ions becomes low at low C_i while their adsorption sites are saturated at high C_i ⁴⁴. By contrast, S-PAcH, in particular S-PAcH(L), exhibited very high R_e ($\sim 100\%$) for all PMs even at low C_i ($<50 \text{ mg L}^{-1}$), in which microscale precipitation was not induced (Figs. 3a–c and Supplementary Figs. 6

and 21), demonstrating its superior PM recovery performance.” (Lines 186-191), together with providing the above Fig. R9 and related discussion as a new Supplementary Fig. S21.

Comment 1-5: Line 190: The authors state that the adsorption can be associated with the homogeneous monolayer formation. While at a later part in the same paragraph, they state that the adsorption is very rapid and hence reliable kinetics could not be followed. Nevertheless, the authors speculate that the initial chemisorption would be the rate-determining step. While this is agreeable, the correlation between this kinetics and nanoparticle growth is not clearly linked. A 3-D nanoparticle growth is obviously not a monolayer, but a reductive multilayer adsorption process should have taken place. Another important associated question is the electron-transfer mechanism, leading to reductive adsorption. The authors state that the hydrazine moieties in the neighbouring PAcH chains reduce the PMs through electron transfer. If this is the case, would the degree of metal ion reduction depend on the amount (molarity) of hydrazine groups?

Response to Comment 1-5: Thank you for the very insightful comment. Let us justify our claim as follows.

(1) Correlation between adsorption kinetics and nanoparticle (NP) growth

As mentioned in the original manuscript, PM species are (1) chemisorbed *via* electrostatic and chelation interactions and subsequently (2) reduced by the hydrazide groups of S-PAcH. Specifically, as illustrated in the revised PM adsorption mechanism of S-PAcH (please refer to our response to the above comment 1-1), anionic PM species are chemisorbed by the protonated hydrazide group and then reduced to the PM metal state (PM(0)). NP growth is enabled by the collective reduction of PM ions adsorbed onto neighboring PAcH chains. This mechanism suggests that **NPs are formed by the post-transition (reduction) of chemisorbed PM ions; hence, it is reasonable to postulate that NP growth is dictated by chemisorption *via* electrostatic and chelation interactions**, as mentioned in the original manuscript (Lines 202-204).

As demonstrated in the original manuscript (Fig. 3d), the PM adsorption of S-PAcH occurs *via* homogeneous monolayer formation. This is consistent with our proposed mechanism because the chemisorption process *via* electrostatic and chelation interactions is generally regarded as homogeneous monolayer adsorption [Qin et al., *Chem. Eng. J.* 428 (2022) 132493; Hu et al., *J. Mol. Liq.* 367 (2022) 120586; Gurung et al., *Chem. Eng. J.* 228 (2013) 405; Zhou et al., *J. Hazard. Mater.* 172 (2009) 439], and **chemisorption dictates subsequent NP formation**. Hence, we can reasonably conclude that PM adsorption (chemisorption and subsequent reduction) by S-PAcH follows the homogeneous monolayer adsorption mechanism.

In the revised manuscript, we justified our claim by adding the following phrase, “This result indicates that PM species are adsorbed on S-PAcH(L) primarily *via* homogeneous monolayer formation⁴⁸, which is consistent with our proposed mechanism that homogeneous-monolayered, chemisorbed PM ions⁴⁵ are subsequently reduced to PM NPs.” (Lines 213-216).

(2) Dependency of the degree of PM reduction on the amount of hydrazine groups

As illustrated in the revised PM adsorption mechanism of S-PAcH, the hydrazide groups of S-PAcH reduce adsorbed PM ions to NPs while their $-NH_2$ groups being converted into $-NO_2$ groups *via* an electron-transfer mechanism (PM ions receive electrons from adjacent hydrazide groups) [Zhang et al., *Sep. Purif. Technol.* 292 (2022) 121021]. As also mentioned in the

original manuscript, the high local density of neighboring hydrazide groups enabled by the star-shaped architecture is beneficial for growing PM NPs, and thus, enhancing the propensity of precipitation (*i.e.*, forming large and robust precipitates) by promoting intra/intermolecular fusion. Nevertheless, we speculated that the degree of PM reduction (*i.e.*, the ratio of the ionic to reduced PM) would be predominantly determined by the inherent reduction capability of the hydrazide group, and thus, independent of the amount of the hydrazide group (*i.e.*, the concentration of hydrazine groups in the feed solution). To confirm this, we determined the degree of PM reduction from the XPS spectra of PM/S-PAcH(L)s formed with different S-PAcH(L) concentrations (*i.e.*, different concentration of hydrazide groups) in the same PM feed solutions. Specifically, S-PAcH(L) (2.5–40 mg) was added to the PM (200 mg L⁻¹) solutions (50 mL) at pH 2 and stirred at 200 rpm for 3 h. Collected PM/S-PAcH(L)s were vacuum-dried for XPS analysis, as shown in **Fig. R10**.

Fig. R10. Deconvolution of the high-resolution (a) Au4f, (b) Pd3d and (c) Pt4f XPS peaks of the PM/S-PAcH(L) precipitates formed with different S-PAcH(L) concentrations. The precipitates were formed by allowing contact between S-PAcH(L) (0.05–0.8 g L⁻¹) and the PM (200 mg L⁻¹) aqueous solutions (pH = 2) for 3 h.

Table R6 summarizes the area percentages of reduced metal state (PM(0)) peaks in the deconvoluted PM XPS peaks (Au4f, Pd3d, and Pt4f) of the PM/S-PAcH(L) precipitates. It is evident that **the area percentage of the reduced PM(0) peak (*i.e.*, the degree of PM reduction) is nearly identical regardless of the S-PAcH(L) concentration (*i.e.*, the amount of hydrazide groups) for all PMs.** Hence, we can reasonably claim that **the degree of PM reduction (*i.e.*, the PM reduction rate) is not affected by the amount of hydrazide groups, while the extent/propensity of precipitation (*i.e.*, the size and strength of precipitates) is affected by the local density of hydrazide groups (*i.e.*, the star-shaped architecture can increase the local density of hydrazide groups).**

Table R6. Area percentages of reduced metal state (PM(0)) peaks in the deconvoluted PM XPS peaks (Au4f, Pd3d, and Pt4f) of the PM/S-PAcH(L) precipitates formed with different S-PAcH(L) concentrations.

S-PAcH(L) concentration (mg L ⁻¹)	Peak area percentage of PM(0) (%)		
	Au	Pd	Pt
0.05	88.9	86.5	80.1
0.1	88.9	86.4	80.6
0.2	88.8	86.7	80.6
0.4	88.4	86.7	80.5
0.8	88.5	86.5	80.2

In the revised manuscript, we supplemented the PM reduction mechanism depending on the amount of hydrazide groups as follows, “PM/S-PAcH precipitates formed with different S-PAcH concentrations exhibited the nearly identical fraction of the PM metal state (Supplementary Fig. 15 and Supplementary Table 3), indicating that the degree of PM reduction is determined by the inherent reduction capability of the hydrazide group of S-PAcH” (Lines 153-156), together with providing the above Fig. R10 and Table R6 as new Supplementary Fig. 15 and Supplementary Table 3, respectively.

Comment 1-6: Line 241: Here, the reduction of 4-NP would be more appropriate than degradation, as no mineralization is happening here. Similar is the case with methyl orange.

Response to Comment 1-6: Thank you for the kind comment. We totally agree with the reviewer that “reduction” is a more appropriate expression than “degradation” because the reaction between NaBH₄ and the dyes (4-NP and MO) is reduction catalyzed by metal NPs [Sargin et al., *Sep. Purif. Technol.* 247 (2020) 116987]. Specifically, the nitro (–NO₂) group of 4-NP is reduced to the amine (–NH₂) group, while the azo (–N=N–) group of MO is reduced to two –NH₂ groups by highly reducible NaBH₄ [Ruan et al., *J. Phys. Chem. C* 121 (2017) 25882; Kgatele et al., *Catalysts* 11 (2021) 428]. Because the dyes (4-NP and MO) were converted into other organic compounds without undergoing mineralization, “reduction” is a correct expression.

In the revised manuscript, we corrected “degradation” to “reduction” throughout the manuscript.

Comment 1-7: Lines 453 & 458: 'R' to be subscript.

Response to Comment 1-7: Thank you for pointing out the typo.

In the revised manuscript, we corrected HR to H_R.

Reviewer #2

General comment: This manuscript reports a method of Precious Metals recovery by Hydrazide-Functionalized Star-Shaped Polymers. Various characterization techniques were employed and lots of experiments had been done to verified the effectiveness of S-PAcH for the adsorption of PMs. Although some interesting results were present, I still believed that this study fail to provide enough new insight on the underlying working mechanism. Hence, this manuscript is not appropriate for Nature Communications.

Response to General Comment: We greatly appreciate the reviewer for many valuable and insightful comments, which helped us significantly improve the scientific and practical impacts of our manuscript. We carefully addressed all the issues raised by the reviewer. Please refer to our responses to the reviewer comments as follows.

Comment 2-1: The explanation of the reaction mechanism is uncertain, the new insights need to be illustrated and the novelty need to be justified as well.

Response to Comment 2-1: Thank you for the valuable comment. We agree with the reviewer that the reaction mechanism of our proposed S-PAcH needs to be clarified with providing new insights and justifying the novelty.

To identify the reliable PM adsorption mechanism of S-PAcH, we carefully considered any factors to cause artifacts. We suspected that water in the atmosphere could be readily adsorbed on highly hydrophilic S-PAcH, possibly leading to misinterpretation of XPS data. Specifically, we speculated that the original broad O1s XPS spectrum of the PM-adsorbed S-PAcH (PM/S-PAcH) at ~533 eV (Fig. 2g and Supplementary Fig. 15) could be interfered with the adsorbed water whose O1s peak strongly appears at 533.1 eV [Heine et al., *J. Am. Chem. Soc.* 138 (2016) 13246].

To avoid the possible interference of adsorbed water, we performed the XPS analysis of PM/S-PAcH immediately after the sample was vacuum-dried at 50 °C for 48 h. As shown in **Fig. R1**, the reanalyzed O1s XPS spectrum of PM/S-PAcH exhibited the narrower peak at a lower binding energy without displaying the peak at ~533 eV compared with the original counterpart. This result clearly confirms that our previous XPS spectra were contaminated with adsorbed water. Hence, the XPS spectra of PM/S-PAcH were reanalyzed with the sample immediately after vacuum drying.

Fig. R1. Original (top) and reanalyzed (bottom) high-resolution O1s XPS spectra of the PM/S-PAcH(L) precipitates: (a) Au, (b) Pd and (c) Pt.

Fig. R2 shows the reanalyzed N1s, O1s, and C1s XPS spectra of PM/S-PACl. The N1s XPS peak of PM/S-PACl was deconvoluted into three peaks at 399.7 ($-\text{NO}_2$), 400.5 (N-metal-N), and 401.9 (protonated amine) eV (**Figs. R2a-c**), which were absent for pristine S-PACl (original Supplementary Fig. 2a) [Zhang et al., *Sep. Purif. Technol.* 292 (2022) 121021]. Compared with the original spectra, the reanalyzed N1s spectra of PM/S-PACl showed a new peak at 399.7 eV ($-\text{NO}_2$) without displaying the peak at 399.5 eV (N-metal-O). This result indicates that (1) protonated amine and $-\text{NO}_2$ groups and N-metal-N chelation bonding are formed in PM/S-PACl after PM adsorption.

The deconvolution of the O1s spectra was not informative because both peaks corresponding to $-\text{NO}_2$ and C-O-C were overlapped at 532.3 eV (**Figs. R2d-f**) [Luo et al., *Adv. Mater.* 30 (2018) 1706498]. Hence, we focused on the C1s spectrum, which was deconvoluted to two peaks at 284.8 (C-C) and 288.0 (O=C-N) eV for both pristine S-PACl and PM/S-PACl (**Figs. R2g-i and R3**) [Zhang et al., *Sep. Purif. Technol.* 292 (2022) 121021]. This result suggests that (2) the oxygen atom in the carbonyl (C=O) group (carbonyl oxygen) of S-PACl is not protonated in PM/S-PACl after PM adsorption, unlike our original interpretation from the O1s spectra that protonation on the carbonyl oxygen (C=OH^+) occurs.

Fig. R2. Deconvolution of the high-resolution (a-c) N1s, (d-f) O1s, and (g-i) C1s XPS spectra

of the PM/S-PAcH(L) precipitates: (a, d, g) Au, (b, e, h) Pd and (c, f, i) Pt.

Fig. R3. Deconvolution of the high-resolution C1s XPS peaks of S-PAcH.

Fig. R4. Revised PM adsorption mechanism of S-PAcH

Based on the reanalyzed XPS data, we carefully identified the PM adsorption mechanism of S-PAcH at low pH, as illustrated in **Fig. R4**. The primary amine ($-\text{NH}_2$) groups of S-PAcH are protonated preferentially over its secondary amine ($-\text{NH}-$) groups under acidic conditions owing to their stronger basicity (*i.e.*, higher electron-donating ability) [Smith, *Organic chemistry 4th edition*, McGraw-Hill (2014)]. S-PAcH with protonated amine ($-\text{NH}_3^+$) groups adsorbs anionic PM species (*i.e.*, AuCl_4^- , PdCl_4^{2-} , and PtCl_6^{2-}) via long-range electrostatic interaction [Lin et al., *J. Mater. Chem. A* 5 (2017) 13557], followed by ion-exchange and chelation mainly with the unshared electron-bearing nitrogen atoms of its unprotonated ($-\text{NH}-$) groups (**Fig. R4a**) [Yang et al., *J. Mater. Chem. A* 8 (2020) 3438; Ain et al., *Spectrochim. Acta, Part A* 115 (2013) 683]. Meanwhile, S-PAcH molecules coagulate owing to their screened electrostatic charges. Subsequently, the $-\text{NH}_3^+$ groups of S-PAcH are deprotonated to $-\text{NH}_2$ while protonating the adsorbed PM ions via the acid–base reaction between the $-\text{NH}_3^+$ groups (acid) and PM ions (base) (**Fig. R4b**) [Zhang et al., *Sep. Purif. Technol.* 292 (2022) 121021]. The hydrazide groups of S-PAcH then reduce adsorbed PM ions to NPs while their $-\text{NH}_2$ groups being converted into $-\text{NO}_2$ (**Figs. R4b and c**) [Zhang et al., *Sep. Purif. Technol.* 292 (2022) 121021]. Continuous PM reduction leads to NP growth and induces intra/intermolecular chain fusion through chelation (N–metal–N) between the NPs and unshared electron-bearing nitrogen atoms of $-\text{NH}-$ groups in neighboring PAcH chains, leading to the rapid formation of

large and robust precipitates (Fig. R4c). This mechanism is consistent with the presence of XPS spectra corresponding to $-\text{NO}_2$ and N–metal–N for PM/S-PAcH. Meanwhile, a small fraction of the adsorbed PM species exists as an ionic state in PM/S-PAcH *via* electrostatic and chelation interaction (Fig. R4c), as evidenced by the presence of the PM ionic XPS peak (original Fig. 2e and Supplementary Fig. 11) and N1s XPS peak corresponding to the protonated amine (Figs. R2a–c) for PM/S-PAcH.

This revised PM adsorption mechanism of S-PAcH revealed a unique PM adsorption mechanism that can be exploited for developing standalone adsorbents—**how the star-shape architecture of the hydrazide-functionalized polymer can perform as a standalone adsorbent**. Based on this mechanism, **our study provided new insights into the effects of the architecture (linear vs star-shape) and chemistry (hydrazide vs ethyleneimine vs allylamine) of the standalone polymer on PM adsorption performance and mechanism, which have not been identified previously**. Specifically, we found that the star-shaped polymer architecture with hydrazide chemistry (S-PAcH) can achieve superior PM adsorption capacity and selectivity and easier recovery (*via* strong precipitation) compared with its linear counterpart and the other chemistries. The benefits of S-PAcH were attributed to its high reduction capability combined with its adsorption mechanism *via* electrostatic and chelation interactions. This finding highlights the beneficial feature of S-PAcH with both adsorbent and reductant functions, which synergistically improve its PM recovery efficiency and selectivity above those achievable by amine polymers with an adsorption function only or reducing agents with a reduction function only, as we responded to the below comments 2-4 and 2-8 (Please also refer to our response to the comments 2-4 and 2-8). Hence, **our proposed mechanism provides new and important insights into PM adsorption mechanisms which allow us to rationally design high-performance standalone adsorbents**.

In the revised manuscript, we made the following corrections.

(1) We revised the PM adsorption mechanism of S-PAcH as follows, “The N1s XPS peak of PM/S-PAcH was broader than that of the pristine S-PAcH and deconvoluted into three peaks at 399.7 ($-\text{NO}_2$), 400.5 (N–metal–N), and 401.9 (protonated amine) eV²⁰, which were absent for S-PAcH (Fig. 2f and Supplementary Figs. 2 and 16). Deconvolution of the C1s peak revealed two peaks at 284.8 (C–C) and 288.0 (O=C–N) eV for both S-PAcH and PM/S-PAcH (Fig. 2g and Supplementary Fig. 17)²⁸. These results suggest that protonated amines, $-\text{NO}_2$ groups, and N–metal–N chelation bonding are formed while carbonyl oxygen atoms remaining unprotonated in PM/S-PAcH after PM adsorption. Given the results above, the PM adsorption mechanism of S-PAcH at low pH can be depicted as illustrated in Figs. 2h–j. The primary amines ($-\text{NH}_2$) of S-PAcH are protonated preferentially over its secondary amines ($-\text{NH}-$) under acidic conditions owing to their higher basicity (*i.e.*, electron donating nature)³³. S-PAcH with protonated primary amines ($-\text{NH}_3^+$) adsorbs anionic PM species (*i.e.*, AuCl_4^- , PdCl_4^{2-} , and PtCl_6^{2-}) *via* long-range electrostatic interactions^{30,40}, followed by ion-exchange and chelation mainly with the unshared electron-bearing nitrogen atoms of its unprotonated $-\text{NH}-$ (Fig. 2h)^{14,41}. Meanwhile, S-PAcH molecules coagulate owing to their screened electrostatic charges. Subsequently, $-\text{NH}_3^+$ of S-PAcH are deprotonated to $-\text{NH}_2$ while protonating the adsorbed PM ions *via* the acid ($-\text{NH}_3^+$)–base (PM ion) reaction (Fig. 2i)²⁸. The hydrazide groups of S-PAcH then reduce the adsorbed PM ions to NPs while their $-\text{NH}_2$ being converted into $-\text{NO}_2$ (Figs. 2i and j)²⁸. Continuous PM reduction leads to NP growth and induces intra/intermolecular chain fusion through chelation (N–metal–N) between the

NPs and unshared electron-bearing nitrogen atoms of –NH– in neighboring PAcH chains, leading to the rapid formation of large and robust precipitates (Fig. 2j). A small fraction of the adsorbed PM species exists as an ionic state in PM/S-PAcH via ion electrostatic and chelation interaction (Fig. 2j), as evidenced by the ionic PM and N1s (corresponding to the protonated amine) XPS peaks detected for PM/S-PAcH.” (Lines 156-180).

(2) We replaced original Figs. 2f–j with the above Figs. R2a and g and R4 with providing the above Fig. R3 as a new Supplementary Fig. 2c. We also replaced original Supplementary Figs. S14 and S15 with the above Figs. R2b, c, h, and i.

(3) We highlighted the novelties and benefits of our proposed mechanism of S-PAcH for PM recovery as follows, “The superior PM recovery performance of S-PAcH was attributed to its strong reduction capability combined with its chemisorption mechanism.” (Lines 27-29), “the unprecedentedly high-capacity and rapid PM adsorption of S-PAcH(L) can be attributed to its high reduction capability combined with its effective adsorption mechanism via strong electrostatic and chelation interactions” (Lines 236-239), and “This result highlights the practically feasible, excellent PM recovery performance and selectivity of S-PAcH(L), which are attributable to its high reduction capability combined with its electrostatic and chelation interaction-mediated adsorption mechanism.” (Lines 330-333).

Comment 2-2: What is the reference for H_R with adsorption capacity?

Response to Comment 2-2: Thank you for the valuable comment. An adsorption process, in which adsorbates (*e.g.*, atoms, ions, and molecules) adhere to the surface of an adsorbent via various interactions, creates an adsorbate surface layer. Hence, it is rationally postulated that the size (or hydrodynamic diameter (H_R)) of an adsorbent would increase with increasing the amounts of adsorbates bound by the adsorption process. Although the H_R of the adsorbent, which is determined by DLS, cannot quantify adsorption capacity, it can provide the qualitative information of the degree of adsorption. In fact, in several other reports, the particle size was monitored to confirm whether adsorption occurs [Jung et al., *Water Res.* 244 (2023) 120543; Culver et al., *Analyst* 142 (2017) 3183; Jin et al., *Colloids Surf. A* 597 (2020) 124791]. Hence, we monitored the change in H_R of the polymers to qualitatively analyze their adsorption degree and associated precipitation behavior.

In the revised manuscript, we justified the H_R analysis by including the following sentence, “The H_R of the polymers was characterized to qualitatively analyze their adsorption degree and associated precipitation behavior⁵⁸.” (Lines 401-402), together with citing the relevant reference [Jung et al., *Water Res.* 244 (2023) 120543].

Comment 2-3: The redox reaction of precious metals requires overcoming the potential complexation with organic pollutants, but this lacks experimental evidence.

Response to Comment 2-3: We appreciate the reviewer’s insightful comment. As the reviewer mentioned, organic pollutants coexistent in the feed solution could potentially disturb the redox reaction between the polymer adsorbent and PM ions by forming complexes with the adsorbent, thus impairing the PM adsorption performance of the adsorbent [Lam et al., *Chem. Eng. J.* 145 (2008) 185]. For this reason, in many e-waste recovery processes, organic pollutants are pre-screened from PM leachates via various pretreatments [Akcil et al., *Waste Manag.* 45 (2015)

258; Ahirwar and Tripathi, *Environ. Nanotechnol. Monit. Manag.* 15 (2021) 100409; Islam et al., *J. Clean. Prod.* 323 (2021) 129015]. Hence, the effect of coexistent organic pollutants on PM recovery performance may not be a critical concern.

Nevertheless, as the reviewer suggested, we identified the effect of coexisting organic pollutants on PM recovery by S-PAcH(L) by evaluating the PM recovery efficiency (R_e) of S-PAcH(L) using feed solutions containing various organic pollutants, as shown in **Fig. R5**. We selected natural organic matter (NOM), fulvic acid (FA), and humic acid (HA) as organic pollutants which are present in industrial water [Luo et al., *Chemosphere* 303 (2022) 135183]. We also selected polychlorinated biphenyls (PCBs), polybrominated diphenyl ethers (PBDEs), polyaromatic hydrocarbons (PAHs), and phthalate esters (PAEs) as organic pollutants which can be generated during the leaching process of e-waste [Yu et al., *Sci. Total Environ.* 694 (2019) 133643]. PM (200 mg L^{-1}) aqueous solutions at pH 2 were mixed with each organic pollutant (200 mg L^{-1}) or all combined pollutants (200 mg L^{-1} each). S-PAcH(L) (10 mg) was added to the organic pollutant-containing PM solutions (50 mL) and stirred at 200 rpm for 3 h . The mixture was then filtered through a PSF ultrafiltration membrane, and the supernatant was collected. The PM ion concentrations of the solutions obtained before and after the addition of S-PAcH(L) were measured using an ICP-OES to quantify the R_e .

Fig. R5 showed that the R_e values of S-PAcH(L) for all PMs were similar for the PM solutions with and without (control) organic pollutants. This result clearly demonstrates that coexisting organic pollutants cannot impair the PM reduction capability, and thus, recovery performance of S-PAcH(L). This could be attributed to 1) poor affinity between highly hydrophilic S-PAcH and relatively hydrophobic organic pollutants (i.e., low hydrophobic interactions) [Yu et al., *Sci. Total Environ.* 694 (2019) 133643] and 2) rapid PM adsorption by S-PAcH(L), which could significantly reduce the probability of organic pollutants to form complexes with S-PAcH(L).

Fig. R5. PM recovery efficiency (R_e) of S-PAcH(L) for Au, Pd, and Pt for PM aqueous solutions with and without (control) organic pollutants (S-PAcH(L) concentration = 0.2 g L^{-1} , initial PM concentration = 200 mg L^{-1} , organic pollutant concentration = 200 mg L^{-1} each, solution pH = 2, contact time = 3 h).

We also evaluated the effect of coexisting organic pollutants on PM recovery by S-PAcH(L) using real waste solutions, which are the leachates of real-world CPU (Au) and spent catalyst (Pd and Pt) samples, as displayed in **Fig. R6**. Experiments were performed as follows. CPU (Au, Intel) was obtained from an end-of-life computer, and spent catalysts (Pd and Pt, Sigma Aldrich) were obtained after their use in hydrogenation reactions. Real-world leachate feed

solutions were prepared by following a previously reported protocol [Zhang et al., *Sep. Purif. Technol.* 292 (2022) 121021]. Each real-world sample (20 g) was immersed in aqua regia (500 mL) for 3 d. The mixture was filtered through a cellulose filter paper (JIS P 3801, pore size = 1 μm , Advantec) to remove undissolved solids and further diluted to 1 L with DI water while adjusting pH to 2 using 1 N NaOH aqueous solution. S-PAcH(L) (10 mg) was added to each leachate solution (50 mL) and stirred at 200 rpm for 3 h. The mixture was then filtered through a PSF ultrafiltration membrane, and the supernatant was collected. The PM ion and total organic carbon (TOC) concentrations of the solutions obtained before and after the addition of PAcH(L) were measured using ICP-OES and TOC analyzer (TOC-L, Shimadzu, Japan) respectively. PM ion and TOC concentrations in the real-world leachates were 9–11 and 1.4–2.1 mg L^{-1} , respectively.

Fig. R6 showed that S-PAcH(L) completely recovered all PMs (*i.e.*, ~100% R_e) from all the real-world CPU and spent catalyst leachates without adsorbing coexisting organic pollutants. This result further demonstrates the high adsorption selectivity of S-PAcH(L) towards PM ions over organic pollutants, which attributed to its rapid PM adsorption capability combined with its poor affinity with organic foulants. Therefore, we can reasonably claim that our S-PAcH adsorbent can maintain its high PM reduction capability and recovery performance even with organic pollutant-containing real feed solutions by overcoming its potential complexation with coexisting organic pollutants. Thanks to the reviewer' insightful comment, we were able to further highlight the beneficial feature of our S-PAcH adsorbent.

Fig. R6. (a) PM ion and (b) TOC concentrations in the real-world CPU and spent catalyst (Pd and Pt) leachates before and after the addition of S-PAcH(L) (S-PAcH(L) concentration = 0.2 g L^{-1} , solution pH = 2, contact time = 3 h).

In the revised manuscript, we made the following corrections.

(1) We included the results and discussion on the effect of coexisting organic pollutants on PM recovery by S-PAcH(L) for model PM solutions as follows, “S-PAcH(L) also maintained its high R_e for PMs even for simulated feed solutions containing model organic pollutants that can possibly impair the PM adsorption performance of adsorbents by forming adsorbent–pollutant complexes (Supplementary Fig. 30 and Supplementary Text). This result demonstrates the high adsorption selectivity of PAcH(L) toward PMs over organic pollutants, which can be attributed to its rapid PM adsorption capability combined with its poor affinity with relatively hydrophobic organic pollutants⁵².” (Lines

269-274), together with providing the above Fig. R5 and related discussion as new Supplementary Fig. 30 and Supplementary text.

(2) We also included the results and discussion on the effect of coexisting organic pollutants on PM recovery by S-PAcH(L) for real-world leachate solutions as follows, “S-PAcH(L) completely recovered PMs (*i.e.*, ~100% R_e) from all the real-world leachate solutions without absorbing coexisting metal ions (Figs. 5d–f) and organic pollutants (Supplementary Fig. 34).” (Lines 325-328), together with providing the above Fig. R6 as a new Supplementary Fig. 34.

Comment 2-4: The adsorbent is incinerated to obtain high-purity precious metals, and the author should conduct economic evaluations, which are necessary for the sustainability aspect.

Response to Comment 2-4: We appreciate the reviewer’s very insightful comment and useful suggestion. We totally agree with the reviewer that the sustainable and economical use of adsorbents is critical for their practical application. Let us respond to the reviewer’s comment point by point as follows.

(1) Sustainable use (reusability) of our S-PAcH adsorbent

As mentioned in the manuscript, our S-PAcH adsorbs PMs *via* reduction combined with electrostatic and chelation interactions. Careful reanalysis of its PM adsorption mechanism revealed that the primary amine ($-NH_2$) groups of S-PAcH are oxidized to $-NO_2$ groups when they reduce PM ions to PM NPs, as we responded to the above comment 2-1.

Our PM-adsorbed S-PAcH (PM/S-PAcH) precipitates can be either calcinated (as demonstrated in the original manuscript) or regenerated for reuse as demonstrated below. To regenerate PM-adsorbed adsorbents by desorbing PMs, agents with high affinity for PMs, such as thiourea, cyanide, and thiocyanate, are commonly used [Ruan et al., *J. Phys. Chem. C* 121 (2017) 25882]. In our study, we used thiourea and Fe^{3+} to regenerate S-PAcH through PM desorption. Thiourea with high affinity for PMs facilitates the desorption of PM species (especially PM ions) from PM/S-PAcH, while Fe^{3+} with high oxidation ability oxidizes adsorbed PM NPs to PM ions, thus assisting PM desorption [Zhou et al., *Environ. Sci. Technol.* 57 (2023) 3334]. Meanwhile, the reduced form of Fe^{3+} (Fe^{2+}) can reduce the $-NO_2$ groups (oxidized amine groups) of S-PAcH to primary amine ($-NH_2$) groups, thus regenerating S-PAcH [Hofstetter et al., *Environ. Sci. Technol.* 40 (2006) 235].

The reusability of our S-PAcH was assessed with the leachates of real-world CPU and spent Pd and Pt catalyst samples, which were prepared as described in our response to the above comment 2-3. S-PAcH(L) (10 mg) was added to each leachate solution (50 mL) and stirred at 200 rpm for 3 h. The mixture was then filtered through a PSF ultrafiltration membrane, and the supernatant was collected. The PM ion concentration of the permeate solution was measured using an ICP-OES to calculate the R_e . The filtrated PM/S-PAcH(L) was put into the aqueous solution containing thiourea (1M), $FeCl_3$ (1M), and HCl (1M) and sonicated for 30 min to desorb PM species from PM/S-PAcH(L). The mixture was filtered through a PSF ultrafiltration membrane, and the PM concentration of the permeate solution (C_D) was measured using an ICP-OES to calculate the PM desorption efficiency (D_e), as given by:

$$D_e = \frac{C_D}{C_L \times R_e} \times 100 \quad (\text{Eq. R1})$$

where C_L is the PM ion concentration of the real-world leachate. The filtrated S-PAcH(L) was freeze-dried and then used to repeat the above adsorption–desorption process seven times.

Fig. R7 shows the R_e and D_e values as a function of the number of adsorption–desorption

cycles. The R_e value of S-PAcH(L) very slightly decreased with increasing the number of adsorption–desorption cycles; S-PAcH(L) underwent ~5% (for Au and Pd) and ~14% (for Pt) reductions in its R_e after seven adsorption–desorption cycles, corresponding to ~0.7% (for Au and Pd) and ~2% (for Pt) reductions in R_e per adsorption–desorption cycle. **Compared with other reported PM adsorbents, S-PAcH(L) exhibited a lower reduction in R_e per adsorption–desorption cycle (Table R1), confirming its excellent reusability (sustainable use).**

Fig R7. Recovery (R_e) and desorption (D_e) efficiency of S-PAcH(L) with real-world leachate solutions as a function of the number of adsorption–desorption cycles: (a) CPU leachate (Au), (b) spent Pd catalyst leachate, and (c) spent Pt catalyst leachate (S-PAcH(L) concentration = 0.2 g L⁻¹, solution pH = 2, and adsorption time = 3 h).

Table R1. Reductions in R_e per adsorption–desorption cycle of reported PM adsorbents.

Adsorbent	PM	Reduction in R_e per adsorption–desorption cycle	Ref.
ADH@BC hybrid membrane ^a	Au	0.4	[1]
Poly-Cys-g-PDA@GPUF ^b	Au	2.5	[2]
2-Mercaptobenzothiazole-impregnated amine-functionalized resin	Pd	4.8	[3]
AHPP-MOF ^c	Pd	2.1	[4]
MNP-G3 ^d	Pd	1.7	[5]
Poly(allylamine hydrochloride)-modified E. coli	Pt	4.4	[6]
S-PAcH(L)	Au	0.7	This study
	Pd	0.7	
	Pt	2	

^aAdipic dihydrazide-grafted bacterial cellulose hybrid membrane. ^bCysteine polymer brush-grafted polydopamine-modified graphene-based polyurethane foam. ^c4-amino-3-hydroxybenzoic acid-modified Zr-based metal-organic framework. ^dMagnetic nanoparticle modified by third-generation dendrimer. ([1] Zhang et al., *Sep. Purif. Technol.* 292 (2022) 121021; [2] Xue et al., *React. Funct. Polym.* 136 (2019) 138; [3] Sharma and Rajesh, *Chem. Eng. J.* 283 (2016) 999; [4] Tang et al., *Chem. Eng. J.* 407 (2021) 127223; [5] Yen et al., *J. Hazard. Mater.* 322 (2017) 215; [6] Mao et al., *Water Res.* 44 (2010) 5919)

(2) Assessment of sustainability and economic analysis

Strictly speaking, the sustainability of the adsorption process needs to be assessed based on its whole cyclic process, including use and regeneration. We also need to consider environmental impact and social acceptance together with economic viability [Büyüközkan and Karabulut, *J. Environ. Manage.* 217 (2018) 253]. Even if any PM adsorbent is reusable, it cannot be regarded as “completely sustainable or environmentally sustainable” if environmentally hazardous reagents, such as thiourea (cytotoxic), HCl (corrosive), and FeCl₃ (oxidative) are used for the regeneration of the adsorbent [Sharma et al., *Environ. Sci. Technol.* 33 (1999) 2645; Akcil et al., *Waste Manage.* 45 (2015) 258]. In this meaning, the calcination process is also not sustainable because it emits gas (e.g., CO₂) waste and does not allow for the reuse of the adsorbent. As demonstrated above and in the original manuscript, our S-PAC_H polymer can be either regenerated for reuse or calcinated for PM recovery. **Although S-PAC_H cannot achieve “complete sustainability”, it can be sustainably reused at least.**

To evaluate the economic viability, we first estimated the price of S-PAC_H. In fact, it is very difficult to accurately estimate the price of S-PAC_H, which is a lab-scale sample, because the accurate information of additional production costs (e.g., facilities, plant planning, labor, etc.) and the industrial-grade prices of raw materials is not available for us, who are academic field researchers. Nevertheless, we made our best efforts to evaluate the final price of S-PAC_H based on the reagent-grade prices of the raw materials required for its lab-scale synthesis.

Table R2 summarizes the amounts and unit prices of the raw materials used for synthesizing S-PAC_H(L) of 1 kg. The pure price of S-PAC_H(L) calculated based on its raw material prices was ~\$6961.3 kg⁻¹. If we assume that raw material costs occupy approximately 60–90% of the total production cost [Meneses et al., *Heliyon* 8 (2022) e09028], the final price of S-PAC_H(L) can be reasonably estimated to be \$7734.8–11602.2 kg⁻¹. However, this calculation is a very rough estimation, and the accurate total price should be estimated by manufacturers.

Table R2. Unit prices, used amounts, and respective costs of the raw materials used for synthesizing S-PAC_H(L) of 1 kg.

Raw material	Manufacturer	Unit price	Used amount	Cost
CDx	Sigma-Aldrich	\$997.4 kg ⁻¹	3.6 g	\$3.6
BiBr	Sigma-Aldrich	\$409.4 kg ⁻¹	24.5 g	\$10.0
CuBr	Sigma-Aldrich	\$3401 kg ⁻¹	8.3 g	\$28.2
MAc	Sigma-Aldrich	\$52.8 L ⁻¹	4.6 L	\$242.8
Alumina	Sigma-Aldrich	\$156.5 kg ⁻¹	1.3 kg	\$195.6
TBABr	Sigma-Aldrich	\$1706.8 kg ⁻¹	1.0 kg	\$1706.8
Hydrazine hydrate	Sigma-Aldrich	\$379.8 kg ⁻¹	7.6 kg	\$2886.5
Na ₂ CO ₃	Daejung Chemical	\$141.4 kg ⁻¹	46.1 g	\$6.5
DCM	Daejung Chemical	\$6.8 L ⁻¹	214.3 mL	\$1.5
THF	Daejung Chemical	\$21.2 L ⁻¹	30.0 L	\$634.6
Methanol	Daejung Chemical	\$7.2 L ⁻¹	162.5 L	\$1175
NMP	Daejung Chemical	\$89.7 L ⁻¹	29.1 mL	\$2.6
Silica	Alfa Aesar	\$53.8 kg ⁻¹	1.3 kg	\$67.3
DI water	Millipore	\$1.5 L ⁻¹	214.3 mL	\$0.3
			Total cost:	\$6961.3 kg ⁻¹
			Expected final price ^a :	\$7734.8 –11602.2 kg ⁻¹

^aEstimated with the assumption that raw material costs occupy approximately 60–90% of the total production cost [Meneses et al., *Heliyon* 8 (2022) e09028].

We then calculated the price (value) ratio of adsorbed PM to S-PAcH(L) in PM/S-PAcH(L)s formed with different PM concentrations based on the weight and price of adsorbed PM and S-PAcH(L), as summarized in **Table R3**. Two PM concentrations, (1) 200 mg L⁻¹, which is the standard PM concentration examined in the manuscript, and (2) PM concentrations in the above-prepared leachates of real-world CPU and spent Pd and Pt catalyst samples, were considered.

As shown in **Table R3**, at the PM concentration of 200 mg L⁻¹, the price of adsorbed (recovered) PM was 1.9 to 8.5 times higher than that of S-PAcH(L). Although more extensive cost analysis is needed, in this case where the value of recovered PM exceeds that of the S-PAcH(L) adsorbent owing to high PM adsorption capacity, simple calcination for PM recovery could be more cost-effective than a relatively complex regeneration process [Biswas et al., *Chem. Eng. J.* 407 (2021) 127225; Yang et al., *Carbohydr. Polym.* 111 (2014) 768]. Conversely, in the case of real-world leachates with low PM concentrations (~10 mg L⁻¹), the price of S-PAcH(L) was higher than that of adsorbed PM. In this case, the regeneration (reuse) of S-PAcH(L) could be more cost-effective than calcination which results in the loss of S-PAcH(L).

Table R3. Weight and cost ratios of adsorbed PM to S-PAcH(L) in PM/S-PAcH(L)s formed with different PM concentrations.

PM concentration	PM	Unit price of PM	Weight ratio of adsorbed PM to S-PAcH(L)	Price ratio of adsorbed PM to S-PAcH(L)
Model solution PM: 200 mg L ⁻¹	Au	\$65819 kg ⁻¹	1	5.7–8.5
	Pd	\$33256 kg ⁻¹	1	2.9–4.3
	Pt	\$31195 kg ⁻¹	0.71	1.9–2.9
Real-world leachate Au: 9.6 mg L ⁻¹ Pd: 10.8 mg L ⁻¹ Pt: 11.1 mg L ⁻¹	Au	\$65819 kg ⁻¹	0.048	0.27–0.41
	Pd	\$33256 kg ⁻¹	0.054	0.16–0.23
	Pt	\$31195 kg ⁻¹	0.056	0.15–0.22

Furthermore, compared with commercial amine polymer adsorbents (bPEI and PAAm), S-PAcH(L) exhibited significantly higher PM adsorption capacity and selectivity with high recoverability by forming reduction-induced large and strong precipitate, as demonstrated in the original manuscript. Hence, it is evident that these key advantages of S-PAcH(L) over commercial amine polymers enable more cost-effective PM recovery by considerably reducing separation processes.

We also compared the cost effectiveness of S-PAcH(L) with that of conventional reducing agents (hydrazine, and NaBH₄). To fairly assess the cost effectiveness of S-PAcH(L), hydrazine, and NaBH₄, their costs required for recovering the same amount of PM were compared. Hence, we first identified the doses of the reducing agents that could achieve the same R_e as that achieved by the dose (0.2 g L⁻¹) of S-PAcH(L) at a given PM ion concentration (200 mg L⁻¹), as shown in **Fig. R8**.

Although both reducing agents exhibited lower R_e for all PMs compared with S-PAcH(L) at the same dose (0.2 g L⁻¹), their R_e values increased with their doses, except for hydrazine whose R_e for Pt remained marginal (~0%) even at the highest dose investigated (5.0 g L⁻¹) owing to

its very low Pt reduction capability [Wu et al., *Angew. Chem. Int. Ed.* 60 (2021) 17587].

Fig. R8. Recovery efficiency (R_e) of S-PAcH(L) (0.2 g L^{-1}) and reducing agents (hydrazine and NaBH_4) for PMs as a function of reducing agent doses (initial PM ion concentration = 200 mg L^{-1} , solution pH = 2, contact time = 3 h).

Based on the prices of materials and their doses required for achieving the same R_e at a given PM ion concentration of 200 mg L^{-1} (determined from **Fig. R8**), we calculated their costs for PM recovery, as summarized in **Table R4**. The material price of S-PAcH(L) was notably higher than those of hydrazine and NaBH_4 . Nevertheless, the cost of S-PAcH(L) for PM recovery was similar or even lower compared with hydrazine and slightly higher compared with NaBH_4 owing to the superior PM adsorption capacity of S-PAcH(L). We also need to consider the recoverability of adsorbents and reducing agents to assess their cost effectiveness for practical PM recovery. The small molecular size of reducing agents (unreacted and/or reacted forms) renders their recovery/collection difficult [Chen et al., *Chemosphere* 49 (2002) 363], which requires additional costs for purification/recovery processes. Particularly, in a PM reduction process using NaBH_4 , toxic impurities, such as boric acid (H_3BO_3) and sodium metaborate (NaBO_2), are generated, which should be removed by employing additional separation processes [Chen et al., *Chemosphere* 49 (2002) 363]. Hence, we believe that our S-PAcH polymers with higher PM adsorption capacity and recoverability would be more cost-effective compared with reducing agents.

Table R4. Prices, required doses, and respective costs of the materials used for PM recovery.

Materials	Manufacturer	Material price (\$ kg^{-1})	Dose (g L^{-1})			Cost (\$ L^{-1})		
			Au	Pd	Pt	Au	Pd	Pt
S-PAcH(L)	Lab-made	7734.8 –11602.2	0.2	0.2	0.2	1.55– 2.32	1.55– 2.32	1.55–2. 32
Hydrazine	Sigma-Aldrich	379.8	5.0	>5.0	>>5.0	1.90	>1.90	>>1.90
NaBH_4	Daejung Chemical	157.7	>5.0	>5.0	0.36	>0.79	>0.79	0.06

We further compared the R_e and selectivity of S-PAcH(L) with those of commercial amine polymers and reducing agents with real-world leachates. The real-world CPU and spent Pd/Pt catalyst leachate solutions contained various coexisting ions; the CPU leachate mainly

contained Au (9.6 mg L⁻¹), Ni (84.5 mg L⁻¹), and Cu (102.9 mg L⁻¹) ions while spent Pd and Pt catalysts contained Al (2.9 and 3.7 mg L⁻¹, respectively) as a main coexisting metal ion together with Pd (10.8 mg L⁻¹) and Pt (11.1 mg L⁻¹), respectively.

Fig. R9 showed that for all the real-world leachates, S-PAC(H)L exhibited ~100% R_e for all PMs but negligible R_e for other coexisting metal cations, similar to the case for simulated feed solutions, as demonstrated in the original manuscript. This result highlights the excellent selectivity of S-PAC(H)L toward all PMs. Compared with S-PAC(H)L, commercial amine polymers (bPEI and PAAm) and reducing agents (hydrazine and NaBH₄) exhibited lower R_e for PMs and higher R_e for coexisting metal ions, and thus, significantly lower selectivity toward PMs. This result demonstrates that **S-PAC(H)L has significantly higher PM adsorption capacity and selectivity compared with commercial amine polymers and reducing agents even with real-world leachate feed solutions.**

Fig. R9. Recovery efficiency (R_e) of the polymers (S-PAC(H)L, bPEI, and PAAm) and reducing agents (hydrazine and NaBH₄) with the real-world CPU and spent Pd and Pt catalyst leachate solutions: (a) CPU leachate, (b) spent Pd catalyst leachate, and (c) spent Pt catalyst leachate. (polymer and reducing agent concentration = 0.2 g L⁻¹, solution pH = 2, contact time = 3 h).

The accurate economic analysis of the S-PAC(H)-based PM recovery process is not possible in this study owing to the lack of the cost information of each unit process. Nevertheless, we believe that **our S-PAC(H) polymers with higher PM adsorption capacity/selectivity and recoverability would be more cost-effective compared with commercial amine polymers and reducing agents**, based on the above considerations. **The cost-effectiveness of S-PAC(H) can also be optimized by calcinating or regenerating it depending on the PM concentration (PM adsorption capacity) in feed solutions.**

In the revised manuscript, we made the following corrections.

(1) We included the protocol of regeneration experiments in the “Methods” section (Lines 467-480).

(2) We included the results and discussion regarding the reusability of our S-PAC(H) adsorbent as follows, “Furthermore, S-PAC(H)L was reusable by desorbing PM from PM/S-PAC(H)L following a well-established regeneration protocol⁵⁷. The R_e value of S-PAC(H)L very slightly decreased with increasing the number of adsorption–desorption cycles (Figs. 5g–i); S-PAC(H)L underwent ~5% (for Au and Pd) and ~14% (for Pt) reductions in its R_e after seven adsorption–desorption cycles, corresponding to ~0.7% (for Au and Pd) and ~2% (for Pt) reductions in R_e per adsorption–desorption cycle. Compared with other reported PM adsorbents, S-PAC(H)L exhibited a lower reduction in R_e per adsorption–desorption cycle (Supplementary Table 11), confirming its excellent

reusability.” (Lines 333-340), together with providing the above Fig. R7 and Table R1 as new Figs. 5g-i and Supplementary Table 11.

(3) We included the comment on the cost effectiveness of our S-PAcH(L) compared with reducing agents as follows, “Moreover, considering the difficulty in recovering small molecular-sized reducing agents⁵¹, we believe that our PAcH with higher PM adsorption performance and recoverability would be more cost-effective at recovering PMs compared with reducing agents (Supplementary Fig. 29, Supplementary Tables 7 and 8, and Supplementary Text).” (Lines 265-269), together with providing the above Fig. R8, Tables R2 and R4, and related explanations as new Supplementary Fig. 29, Supplementary Tables 7 and 8, and Supplementary text.

(4) We also included the comment related with the cost effectiveness of our S-PAcH(L) as follows, “We further assessed the feasibility of S-PAcH for its use in PM recovery from the leachates of real-world CPU and spent Pd/Pt catalysts (Figs. 5a–c and Supplementary Fig. 33). S-PAcH(L) completely recovered PMs (*i.e.*, ~100% R_e) from all the real-word leachate solutions without absorbing coexisting metal ions (Figs. 5d–f) and organic pollutants (Supplementary Fig. 34). Compared with S-PAcH(L), commercial amine polymers and reducing agents exhibited lower R_e for PMs and higher R_e for coexisting metal ions, and thus, displaying significantly lower selectivity toward PMs (Figs. 5d–f). This result highlights the practically feasible, excellent PM recovery performance and selectivity of S-PAcH(L), which are attributable to its high reduction capability combined with its electrostatic and chelation interaction-mediated adsorption mechanism.” (Lines 324-333), together proving the above Fig. R9 as new Figs. 5d–f.

(5) We further highlighted the cost effectiveness of our S-PAcH(L) as follows, “We believe that PM recovery by our S-PAcH is cost-effective owing to the excellent PM adsorption capacity and selectivity of S-PAcH combined with its facile collection. The cost-effectiveness of S-PAcH can also be optimized by calcinating or regenerating it depending on the PM concentration (PM adsorption capacity) in feed solutions (Supplementary Table 12 and Supplementary Text).” (Lines 341-345), together with providing the above Table R3 and related explanations as new Supplementary Table 12 and Supplementary text.

Comment 2-5: The lack of appropriate discussion that frames insights to existing concepts of PMs electro-sorption.

Response to Comment 2-5: We appreciate the reviewer for the very insightful comment. Electro-sorption is generally defined as a phenomenon, “metal adsorption on the surface of charged electrodes (*i.e.*, cathodes) via an electric potential-induced oxidation/reduction reaction in an electrolyte” [Foo and Hameed, *J. Hazard. Mater.* 170 (2009) 552]. The electro-sorption mechanism has been exploited for selective PM recovery [Kasper et al., *Electrochim. Acta* 259 (2018) 500; Halli et al., *ACS Sustainable Chem. Eng.* 6 (2018) 14631; Chen et al., *Sep. Purif. Technol.* 259 (2021) 118204]. Both conventional PM electro-sorption and our S-PAcH-induced PM adsorption processes have a common point in that they utilize an oxidation/reduction reaction for PM recovery, and PM ions are recovered via reduction. However, our S-PAcH-induced PM adsorption process has novel features that cannot be achieved by the conventional electro-sorption process as follows.

To achieve high PM selectivity in an electro-sorption process, a sufficiently low electric potential is required for selectively reducing PM ions with relatively low reduction potential while preventing the reduction of coexisting ions at the expense of slowed PM recovery [Ren

et al., *ACS Central Sci.* 5 (2019) 1396]. Conversely, a high electric potential is required for rapid PM recovery, but it reduces PM selectivity by inducing the electro-sorption of coexisting ions. This demonstrates a critical trade-off between PM selectivity and recovery rate in an electro-sorption process.

Unlike an electro-sorption process whose PM recovery relies solely on the reduction mechanism, PM recovery by our S-PAC_H is achieved by both reduction and adsorption (via electrostatic and chelation interactions) mechanisms. This beneficial feature of S-PAC_H enables rapid, high-capacity, and highly selective PM recovery without the trade-off between PM selectivity and recovery rate observed in an electro-sorption process. Specifically, the high reduction capability of S-PAC_H combined with its adsorption mechanism *via* electrostatic and chelation interactions can synergistically achieve rapid and high-capacity PM recovery. Meanwhile, the strong positive charge of S-PAC_H electrostatically repels coexisting metal cations, resulting in high PM selectivity. This highlights the benefit of S-PAC_H with both adsorption and reduction functions, enabling highly selective and rapid PM recovery, overcoming the trade-off imposed on the electric potential-dependent process.

In the revised manuscript, we included discussion on the comparison of our adsorption process with the conventional electro-sorption process as follows, “An electro-sorption process, in which PM ions are recovered *via* reduction on the electrode surface under electric potential, has also been employed for selective PM recovery¹⁰. Unfortunately, the electro-sorption process exhibits a trade-off between PM selectivity and recovery rate depending on the electric potential strength⁵⁶. In contrast, combined with its high reduction capability, the strong electrostatic repulsion of S-PAC_H toward coexisting metal cations enables highly selective and rapid PM recovery, overcoming the trade-off of the electro-sorption process.” (Lines 302-308).

Comment 2-6: In the context of the title's reference to "upcycling," it is incumbent upon us to establish a clear and academically rigorous definition of this process as it pertains to the recovery of precious metals. Subsequently, we must formulate a robust methodology to substantiate and validate this definition. How do we define this process concerning the recovery of precious metals?

Response to Comment 2-6: We appreciate the reviewer's very insightful and careful comment. First above all, we apologize for causing confusion. We strongly agree with the reviewer's comment that we need to clearly define the processes and provide their relevant methodologies. The term, “upcycling”, refers to “the reuse of a waste material after its upgrading to a value-added form” *via* mechanical/chemical processes, while “recycling” refers to “the reuse of a waste material without any upgrading” [Korley et al., *Science* 2021]. Based on these definitions, let us clearly define the processes of S-PAC_H(L) relevant to PM recovery proposed in our manuscript.

<Refinement into pure PM>

As mentioned in the original manuscript (Fig. 4f), we demonstrated the feasibility of PM recovery using S-PAC_H(L) *via* calcination-mediated refinement. Through the calcination process of PM/S-PAC_H(L), we obtained high-purity solid PM particles, which can be used as “raw materials” for various applications by proper processing. Because raw PM materials have no added value compared with their processed forms (*e.g.*, CPU and catalyst), our proposed refinement process can be classified as a “recycling” process.

<Recycling of PAcH(L) and PM>

As responded to the above comment 2-4, PM/S-PAcH(L) can be regenerated to pristine S-PAcH(L) for reuse. During this regeneration process, PM ions can be obtained *via* their desorption from PM/S-PAcH(L). Because the obtained PM ions have no added value, this recovery process can also be classified as a “recycling” process.

<Catalytic application to dye reduction>

As presented in the original manuscript, S-PAcH(L) can adsorb PM species as reduced NP forms owing to its high reducing ability imparted by its hydrazide groups. It is well known that PM NPs have high catalytic activity for various reactions (*e.g.*, dye reduction, hydrogen production, volatile organic compound oxidation, and hydrocarbon reforming) [Chen et al., *Joule* 5 (2021) 3097–3115; Wang et al., *Angew. Chem.* 120 (2008) 3644–3647; Freyschlag and Madix, *Mater. Today* 14 (2011) 134–142; Quinson et al., *ACS Catalysis* 13 (2023) 4903–4937]. Among various catalytic applications, in our study, we employed PM/S-PAcH(L) as a catalyst for dye reduction. As described in the original manuscript (Fig. 4f, g), we clearly demonstrated that PM/S-PAcH(L) can successfully catalyze dye (4-NP and MO) reduction with high catalytic activity. As a result, PM ions were recovered and upgraded to catalysts, which are value-added forms that can be used in new applications, *via* reduction-mediated NP formation by S-PAcH(L). Hence, this catalytic application process can be classified as an “upcycling” process.

In conclusion, our proposed strategy demonstrated both the recycling and upcycling of recovered PM using S-PAcH(L).

In the revised manuscript,

(1) First, we revised the wording of title “Upcycling” to “Recycling/Upcycling” to better convey the definitions of the PM recovery processes using our S-PAcH(L).

(2) We also clarified the definitions of “recycling” and “upcycling” by adding the following phrases, “The obtained materials were found to be solid PM particles with a purity of 99.9%, corresponding to 24 Karat, which can be reused as raw materials for various applications (referred to as a recycling process). Because S-PAcH can adsorb PMs in a reduced NP form, the PM/S-PAcH precipitates can also be directly utilized as a catalyst in chemical reactions (referred to as a value-added upcycling process⁵³.” (Lines 281-285), together with citing a relevant reference [Korley et al., *Science* 373 (2021) 66–69].

Comment 2-7: The highlights and innovation of this work are not enough in the whole manuscript.

Response to Comment 2-7: We appreciate the reviewer’s critical and valuable comment, which helped us to significantly improve the highlights and novel points of our work after thorough revisions. Let us clearly explain the highlights and novel points of our work point by point.

(Novel point-1)

Our present study is the first report on the “standalone use of the polymer (star-shape polymer) as an adsorbent”, which has not been attempted previously. As mentioned in the “Introduction” part, most adsorptive polymers are used as supported (anchored) forms not as standalone forms to facilitate their collection/recovery. The use of the polymer as a supported

adsorbent inevitably lowers its adsorption performance owing to the high weight fraction of the support. Our work demonstrated, for the first time, that the star-shaped architecture of the polymer with highly reducible (hydrazide) functional groups enables its easy collection by inducing reduction-mediated precipitation, allowing us to use it as a standalone adsorbent for PM recovery.

(Novel point-2)

Thanks to the standalone usability of S-PAcH as a PM adsorbent, our study provided new insights into the effects of the architecture (linear vs star-shape) and chemistry (hydrazide vs ethyleneimine vs allylamine) of the standalone polymer on PM adsorption performance and mechanism, which have not been identified previously. Because polymers are typically anchored onto porous substrates in most previous studies, the effects of the architecture and chemistry of standalone polymers on adsorption performance have not been investigated. By contrast, we clearly identified the structure-property-adsorption performance relationship of the standalone polymer adsorbent by performing systematic and comprehensive experiments. As a result, we found that the star-shaped polymer architecture with hydrazide chemistry can achieve superior PM adsorption capacity and selectivity and easier recovery compared with its linear counterpart and the other chemistries.

(Novel point-3)

Our study provided new insights into the PM adsorption mechanism that can be utilized for developing standalone adsorbents; how the star-shape architecture of the hydrazide-functionalized polymer can perform as a standalone adsorbent. Specifically, based on the proposed PM adsorption mechanism of S-PAcH, S-PAcH can achieve excellent PM adsorption capacity and selectivity while inducing rapid and strong precipitation owing to its high reduction capability combined with its adsorption mechanism *via* electrostatic and chelation interactions. This finding highlights the beneficial feature of S-PAcH with both adsorbent and reductant functions, which synergistically improve its PM recovery efficiency and selectivity above those achievable by amine polymers with an adsorption function only or reducing agents with a reduction function only. Please also refer to our responses to the comments 2-1, 2-4, and 2-8.

(Highlight-1)

Our S-PAcH can achieve unprecedented PM adsorption capacity and selectivity, which are not attainable by previously developed adsorbents, commercial amine polymers, and reducing agents. We clearly demonstrated that the standalone use of S-PAcH achieved superior PM adsorption capacity and selectivity compared with other reported adsorbents, commercial amine polymers, and reducing agents (hydrazine or NaBH₄). We further highlighted the superior PM adsorption capacity and selectivity of S-PAcH compared with commercial amine polymers and reducing agents for real-world leachate solutions. The superior PM adsorption performance of S-PAcH(L) can be attributed to its high reduction capability combined with its adsorption mechanism *via* electrostatic and chelation interactions. Please also refer to our responses to the comments 2-4 and 2-8.

(Highlight-2)

PM-adsorbed S-PAcH can be refined into high-purity PMs *via* calcination, upcycled to catalysts, or regenerated for reuse, highlighting its high practical feasibility. In the original manuscript (Lines 235-253), we clearly demonstrated that PM/S-PAcH can be refined to high-purity metal PMs *via* calcination or upcycled to catalysts for dye reduction, which are value-

added forms. Moreover, as we responded to the comment 2-4, we demonstrated that PM/S-PAcH can also be sustainably reused through a well-established regeneration method. This result clearly highlights the high commercial viability of our proposed strategy.

(Highlight-3)

Our S-PAcH enables a more commercially viable, cost-effective PM recovery process than commercial amine polymers and reducing agents. As we responded to the comment 2-4, we demonstrated that S-PAcH can achieve more cost-effective PM recovery compared with commercial amine polymers and reducing agents owing to its higher PM adsorption capacity/selectivity and recoverability even with real-world leachate feeds. This further highlights the high commercial viability of S-PAcH.

In the revised manuscript, we strengthened the highlights and novel points of our study as follows.

(1) We clarified the novel point of our study by rephrasing the following sentence, “In this study, we synthesize a star-shaped, hydrazide-functionalized polymer (poly(acryloyl hydrazide), S-PAcH)²⁶ and demonstrate, for the first time, that it can be used as a standalone adsorbent to achieve highly efficient, selective, and rapid PM recovery.” (Lines 62-64).

(2) We emphasized the novelty of our study by adding the following sentence, “Based on these experiments, we identified the effects of the architecture and chemistry of the standalone polymer on PM adsorption performance, which have not yet been investigated.” (Lines 75-77).

(3) We emphasized the novelty of our study by adding the following sentence, “Furthermore, the structural and physicochemical properties of S-PAcH before and after PM adsorption were comprehensively analyzed to identify its PM adsorption mechanism that can be utilized for designing standalone adsorbents.” (Lines 77-80).

(4) We emphasized the highlights of our study as follows, “Moreover, PM-adsorbed S-PAcH could be refined into high-purity PMs *via* calcination, directly utilized (upcycled) as catalysts for dye reduction, or regenerated for reuse, demonstrating its high practical feasibility.” (Lines 29-31) and “We also demonstrate that PM-adsorbed S-PAcH (PM/S-PAcH) precipitates can be refined into high-purity PMs, utilized directly as catalysts for dye reduction, or regenerated for reuse. Finally, we highlight the high commercial viability of S-PAcH by demonstrating its higher PM recovery performance and recoverability compared with commercial amine polymers and reducing agents using real-world leachate feed solutions.” (Lines 80-84).

(5) We highlighted the superior PM adsorption performance of S-PAcH as follows, “S-PAcH(L) was also more effective at recovering PMs than conventional reducing agents such as hydrazine and sodium borohydride (NaBH₄) (Supplementary Figs. 24–26 and Supplementary Text). This result highlights the beneficial feature of S-PAcH(L) with both adsorbent and reductant functions, which synergistically improves PM recovery performance above that achievable by amine polymers with an adsorption function only or reducing agents with a reduction function only; the unprecedentedly high-capacity and rapid PM adsorption of S-PAcH(L) can be attributed to its high reduction capability combined with its effective adsorption mechanism *via* strong electrostatic and chelation interactions” (Lines 231-239) and “Compared with reducing agents that can reduce coexisting cations as well as PM ions⁵¹, PAcH(L) also exhibited significantly higher selectivity toward PMs (Supplementary Fig. 28 and Supplementary Text), further

highlighting the benefit of its both adsorbent and reductant functions in selective PM recovery.” (Lines 261-265).

(6) We highlighted the commercial viability of S-PAcH as follows, “We further assessed the feasibility of S-PAcH for its use in PM recovery from the leachates of real-world CPU and spent Pd/Pt catalysts (Figs. 5a–c and Supplementary Fig. 33). S-PAcH(L) completely recovered PMs (*i.e.*, ~100% R_e) from all the real-world leachate solutions without absorbing coexisting metal ions (Figs. 5d–f) and organic pollutants (Supplementary Fig. 34). Compared with S-PAcH(L), commercial amine polymers and reducing agents exhibited lower R_e for PMs and higher R_e for coexisting metal ions, and thus, displaying significantly lower selectivity toward PMs (Figs. 5d–f). This result highlights the practically feasible, excellent PM recovery performance and selectivity of S-PAcH(L), which are attributable to its high reduction capability combined with its electrostatic and chelation interaction-mediated adsorption mechanism.” (Lines 324-333) and “We believe that PM recovery by our S-PAcH is cost-effective owing to the excellent PM adsorption capacity and selectivity of S-PAcH combined with its facile collection. The cost-effectiveness of S-PAcH can also be optimized by calcinating or regenerating it depending on the PM concentration (PM adsorption capacity) in feed solutions (Supplementary Table 12 and Supplementary Text).” (Lines 341-345).

Comment 2-8: In the context of experimental data pertaining to precious metal recovery, the inclusion of empirical findings from real-world samples would enhance the persuasiveness and credibility of the results.

Response to Comment 2-8: Thank you for the valuable suggestion. We fully agree with the reviewer’s comment. As also responded to the above comment 2-4, we evaluated the R_e and selectivity of our S-PAcH(L) for real-world leachate feed solutions and compared them with those of commercial amine polymers and reducing agents as follows.

CPU (Au, Intel) was obtained from an end-of-life computer, and spent catalysts (Pd and Pt, Sigma Aldrich) were obtained after their use in hydrogenation reactions. Real-world leachate feed solutions were prepared by following a previously reported protocol [Zhang et al., *Sep. Purif. Technol.* 292 (2022) 121021]. Each real-world sample (20 g) was immersed in aqua regia (500 mL) for 3 d. The mixture was filtered through a cellulose filter paper (JIS P 3801, pore size = 1 μm , Advantec) to remove undissolved solids and further diluted to 1 L with DI water while adjusting pH to 2 using 1 N NaOH aqueous solution. Polymer adsorbents (S-PAcH(L), bPEI and PAAm) or reducing agents (hydrazine and NaBH_4) (10 mg) were added to each leachate solution (50 mL) and stirred at 200 rpm for 3 h. The mixture was then filtered through a PSF ultrafiltration membrane, and the supernatant was collected. The metal ion concentrations of the leachate solutions before and after the addition of the adsorbents and reducing agents were measured using an ICP-OES.

The real-world CPU and spent Pd/Pt catalyst leachate solutions contained various coexisting ions; the CPU leachate mainly contained Au (9.6 mg L^{-1}), Ni (84.5 mg L^{-1}), and Cu (102.9 mg L^{-1}) ions while spent Pd and Pt catalysts contained Al (2.9 and 3.7 mg L^{-1} , respectively) as a main coexisting metal ion together with Pd (10.8 mg L^{-1}) and Pt (11.1 mg L^{-1}), respectively (Fig. R10).

Fig. R10. a–c, Photographs, SEM, and SEM-EDX mapping images of the (a) CPU and spent (b) Pd and (c) Pt catalysts. d, Metal composition of the CPU and spent Pd and Pt catalyst leachates.

Fig. R11 showed that for all the real-world leachates, S-PAcH(L) exhibited ~100% R_e for all PMs but negligible R_e for other coexisting metal cations, similar to the case for simulated feed solutions, as demonstrated in the original manuscript. This result highlights the excellent selectivity of S-PAcH(L) toward all PMs. Compared with S-PAcH(L), commercial amine polymers (bPEI and PAAm) and reducing agents (hydrazine and NaBH₄) exhibited lower R_e for PMs and higher R_e for coexisting metal ions, and thus, significantly lower selectivity toward PMs (Fig. R11). The superior PM adsorption capability and selectivity of S-PAcH(L) can be attributed to its high reduction capability combined with its adsorption mechanism *via* electrostatic and chelation interactions. This result highlights the beneficial feature of S-PAcH(L) with both adsorbent and reductant functions, which synergistically improve PM recovery performance above that achievable by amine polymers with an adsorption function only or reducing agents with a reduction function only. Particularly, the excellent PM

selectivity of S-PAcH(L) can be attributed to its strong reduction capability combined with electrostatic repulsion between positively charged S-PAcH(L) and coexisting metal cations, as mentioned in the original manuscript (Lines 226-230).

Fig. R11. Recovery efficiency (R_e) of the polymers (S-PAcH(L), bPEI, and PAAm) and reducing agents (hydrazine and NaBH₄) with the real-world CPU and spent Pd and Pt catalyst leachate solutions: (a) CPU leachate, (b) spent Pd catalyst leachate, and (c) spent Pt catalyst leachate. (polymer and reducing agent concentration = 0.2 g L⁻¹, solution pH = 2, contact time = 3 h).

Moreover, as we also responded to the above comment 2-3, we clearly demonstrated that S-PAcH(L) completely recovered all PMs (i.e., ~100% R_e) from all the real-world CPU and spent catalyst leachates without adsorbing coexisting organic pollutants. This result confirms the high adsorption selectivity of S-PAcH(L) towards PM ions over organic pollutants, which attributed to its rapid PM adsorption capability combined with its poor affinity with organic foulants.

In the revised manuscript, we added the following phrases.

(1) We described the protocol to recover PM from real-word samples in the “Methods” section (Lines 451-466).

(2) We included the results and discussion of the PM recovery performance of S-PAcH(L) with real-world samples as follows, “We further assessed the feasibility of S-PAcH for its use in PM recovery from the leachates of real-world CPU and spent Pd/Pt catalysts (Figs. 5a–c and Supplementary Fig. 33). S-PAcH(L) completely recovered PMs (i.e., ~100% R_e) from all the real-word leachate solutions without absorbing coexisting metal ions (Figs. 5d–f) and organic pollutants (Supplementary Fig. 34). Compared with S-PAcH(L), commercial amine polymers and reducing agents exhibited lower R_e for PMs and higher R_e for coexisting metal ions, and thus, displaying significantly lower selectivity toward PMs (Figs. 5d–f). This result highlights the practically feasible, excellent PM recovery performance and selectivity of S-PAcH(L), which are attributable to its high reduction capability combined with its electrostatic and chelation interaction-mediated adsorption mechanism.” (Lines 324-333), together with providing the above Figs. R10 and R11 as new Figs. 5a–f and Supplementary Fig. 33.

Comment 2-9: Turnover frequency (TOF) and turnover number (TON) are critical performance metrics for evaluating precious metal catalysts. Have TON and TOF been calculated for the catalyst used for organic molecule removal? Do they have obvious advantages over other reported catalysts?

Response to Comment 2-9: We really appreciate the reviewer's insightful suggestion, which allowed us to carefully assess the catalytic activity of our material. As the reviewer mentioned, TOF and TON are important measures of catalytic efficiency or catalytic activity. TON is defined as the consumed amount of a reactant per unit amount (mass or mole) of a catalyst. TON denotes the consumed amount of a reactant per unit amount (mass or mole) of a catalyst per unit time. The TON and TOF values of PM/S-PAcH(L), which acts as a heterogeneous catalyst for dye reduction, can be practically calculated by the following equations based on references [Costentin et al., *J. Am. Chem. Soc.* 134 (2012) 11235–11242; Kästner and Thünemann, *Langmuir* 32 (2016) 7383–7391; Saha et al., *Langmuir* 26 (2010) 2885–2893]:

$$\text{TON} = C_{\text{dye}}/C_{\text{cat}} \quad (\text{Eq. R2})$$

where C_{cat} denotes the lowest concentration of the catalyst (PM/S-PAcH(L)) that leads to complete dye reduction at a given dye concentration (C_{dye}).

$$\text{TOF} = \text{TON}/t \quad (\text{Eq. R3})$$

where t denotes the shortest reaction time to reach complete dye reduction.

We first examined dye reduction at different PM/S-PAcH(L) concentrations and times to determine C_{cat} and t , as shown in **Fig. R12**. At the given concentrations ($C_{\text{dye}} = 0.02 \text{ mM}$) of 4-MP and MO dyes, Pd/S-PAcH(L) led to complete dye reduction at 0.001 g L^{-1} (C_{cat}) for 2 min (t) for 4-NP and 1 min (t) for MO, Pt/S-PAcH(L) led to complete reduction at $C_{\text{cat}} = 0.001 \text{ g L}^{-1}$ for $t = 3 \text{ min}$. Au/S-PAcH(L) led to complete dye reduction at $C_{\text{cat}} = 0.005 \text{ g L}^{-1}$ for $t = 5 \text{ min}$.

Fig. R12. UV-vis spectra of organic dye (MO and 4-NP) solutions containing NaBH_4 after the addition of the Pt/S-PAcH(L) precipitates as a function of (a, c, e) PM/S-PAcH(L) concentrations (reaction time = 30 min) and (b, d, f) reaction times (Au/S-PAcH(L) concentration = 0.005 g L^{-1} and Pd/ and Pt/S-PAcH(L) concentration = 0.001 g L^{-1}): (a, b) Au/S-PAcH(L), (c, d) Pd/S-PAcH(L), and (e, f) Pt/S-PAcH(L).

Based on these results, we calculated the TON and TOF values of PM/S-PAcH(L) using the above Eqs. R2 and R3 and compared them with those of other reported catalysts, as

summarized in **Tables R5** and **R6**. Although fair comparison is difficult owing to different reaction conditions, our PM/S-PAcH(L) exhibited comparable (Au/S-PAcH(L)) and/or even higher (Pd and Pt/S-PAcH(L)) TON and TOF values compared with other reported catalysts, indicating its excellent catalytic activity and thus successful upcycling as a catalyst for dye reduction.

Table R5. TON and TOF values of the catalysts for 4-NP reduction.

Catalyst	$C_{4\text{-NP}}$ (mM)	NaBH_4 concentration (mM)	C_{cat} (g L^{-1})	t (min)	TON (mmol g^{-1})	TOF ($\text{mmol g}^{-1} \text{min}^{-1}$)	Ref.
Au/S-PAcH(L)	0.01	1	0.005	5	2	0.4	This work
Pd/S-PAcH(L)	0.01	1	0.001	2	10	5	
Pt/S-PAcH(L)	0.01	1	0.001	3	10	3.3	
Au NP	2	100	0.20	3.3	10.26	3.11	[1]
Au NP	0.05	13	0.08	5	0.60	0.12	[2]
APM-Au NP ^a	4	100	0.20	9	20.31	2.26	[3]
Au NP	2	30	0.20	10	10.15	1.01	[4]
Au NP	0.05	10	1.97	8.6	0.03	0.003	[5]
Pd NP	1.25	125	0.10	71	12.50	0.18	[6]
Pd NP@chitosan-MWCNT ^b	0.091	4.54	3.63	12	0.03	0.002	[7]
Pd NP@Sch-boehmite ^c	0.114	4.54	0.91	4	0.13	0.03	[8]
GG-s-Pt NP ^d	1	100	0.49	240	2.10	0.009	[9]
Pt NP	0.67	6.67	3.33	20	0.20	0.01	[10]

^aAspartam-capped Au NP. ^bPd NP-loaded crosslinked chitosan/MWCNT bead. ^cPd NP on Schiff base-modified boehmite. ^dPt NP stabilized by guar gum. ([1] Pei et al., *Environ. Sci. Pollut. Res.* 24 (2017) 21649–21659; [2] Majumdar et al., *Appl. Nanosci.* 6 (2016) 521–528; [3] Wu et al., *Nanoscale Res. Lett.* 10 (2015) 213; [4] Khan et al., *J. Photoch. Photobio. B* 170 (2017) 181–187; [5] Gao et al., *Nanoscale Res. Lett.* 9 (2014) 404; [6] Bordbar et al., *Environ. Sci. Pollut. Res.* 24 (2017) 4093–4104; [7] Sargin et al., *Sep. Purif. Technol.* 247 (2020) 116987; [8] Baran et al., *J. Organomet. Chem.* 900 (2019) 120916; [9] Pandey and Mishra, *Carbohydr. Polym.* 113 (2014) 525–531; [10] Ullah et al., *J. Photoch. Photobio. B* 173 (2017) 368–375)

Table R6. TON and TOF values of the catalysts for MO reduction.

Catalyst	C_{MO} (mM)	NaBH_4 concentration (mM)	C_{cat} (g L^{-1})	t (min)	TON (mmol g^{-1})	TOF ($\text{mmol g}^{-1} \text{min}^{-1}$)	Ref.
Au/S-PAcH(L)	0.01	1	0.005	5	2	0.4	This work
Pd/S-PAcH(L)	0.01	1	0.001	1	10	10	
Pt/S-PAcH(L)	0.01	1	0.001	3	10	3.3	
Pd NP@chitosan-MWCNT	0.1	79.3	4	1	0.025	0.025	[7]
DLP-Au NP ^a	0.1	79.3	4	8	0.025	0.003	[11]
MgAlCe-LDH@Au ^b	0.08	16	0.007	3.5	12.4	3.6	[12]
MA@Ag NP ^c	0.04	0.17	6.7	16	0.006	0.004	[13]
MA@Cu NP ^d	0.04	0.17	6.7	12	0.006	0.005	[13]

CuO-MgO	0.05	5	0.2	15	0.25	0.017	[14]
Cu-NMOF/Ce-doped-Mg-Al-LDH ^c	0.07	8.4	0.017	1	4	4	[15]
Cu-MOF ^f	0.07	8.4	0.017	3.5	4	1.1	[15]
Au/ZIF-11 ^g	0.22	3.5	0.015	7	15.3	2.2	[16]

^aDalspinin-mediated AuNP. ^bAu NP-loaded Ce-doped magnesium-aluminium layered double hydroxide. ^c*M. azedarach*-supported Ag NP. ^d*M. azedarach*-supported Cu NP. ^eCopper-based nanoscale metal-organic framework combined with Ce-doped magnesium-aluminum layered double hydroxide. ^fCopper-based metal-organic framework. ^gAu NP-decorated ZIF-11. ([11] Umamaheswari et al., *J. Photochem. Photobiol. B* 178 (2018) 33–39; [12] Iqbal et al., *J. Mater. Chem. A* 5 (2017) 6716; [13] Shah et al., *J. Organomet. Chem.* 938 (2021) 121756; [14] Alla et al., *RSC Adv.* 6 (2016) 61927; [15] Iqbal et al., *Inorg. Chem.* 57 (2018) 13270–13278; [16] Malik and Nath, *J. Water Process Eng.* 44 (2021) 102362)

In the revised manuscript, we included the comment on TON and TOF results as follows, “Turnover number (TON) and turnover frequency (TOF) are critical performance metrics for evaluating catalytic activity. PM/S-PaCH(L) exhibited comparable (Au/S-PaCH(L)) and/or even higher (Pd and Pt/S-PaCH(L)) TON and TOF values compared with other reported catalysts (Supplementary Fig. 32 and Supplementary Tables 9 and 10).” (Lines 295-298), together with specifying how to calculate TON and TOF in Supplementary Information and providing the above Fig. R12 and Tables R5 and R6 as new Supplementary Fig. S32 and Supplementary Tables 9 and 10, respectively.

Comment 2-10: The TOC (Table of Content) graph is necessary to help readers better understand the highlights of the work if the journal requires it.

Response to Comment 2-10: Thank you for the valuable suggestion. We totally agree with the reviewer in that the TOC graphic can help readers better understand the highlights of our work. Unfortunately, according to the journal guideline, the Nature Communications journal does not require the TOC graphic, and therefore, we did not provide it. Nevertheless, the following Fig. R13 can be provided as TOC if needed.

Fig. R13. Proposed TOC image.

Comment 2-11: The authors have mentioned that “In this study, we develop, for the first time, a star-shaped, hydrazide-functionalized polymer (poly(acryloyl hydrazide), S-PAcH).....” in terms of the star-shaped, hydrazide-functionalized materials, what is different between this work and doi.org/10.1016/j.cej.2022.137883? Please confirm whether it is the first time synthesis?

Response to Comment 2-11: Thank you for the valuable comment. We apologize for causing confusion. As we responded to the above comment 2-7, let us clarify the novel points of our study compared with the previous reported work. S-PAcH synthesized in the present study is essentially similar to that employed in our previous work (doi.org/10.1016/j.cej.2022.137883). Nevertheless, compared with our previous work, the present work has several novel points in terms of methodology, application, effects, and scientific insights as follows.

(1) Our present study is the first report on the use of the star-shaped polymer (S-PAcH) as a “standalone adsorbent”, which has not been attempted previously. As mentioned in the “Introduction” part, most adsorptive polymers are used as supported (anchored) forms not as standalone forms to facilitate their collection/recovery. Similarly, in our previous study (doi.org/10.1016/j.cej.2022.137883), S-PAcH was used as a “supported form for a separation membrane”; it was deposited on a porous support and then crosslinked to form a membrane separation layer for Cr(VI) removal. Unlike our previous study, the present work demonstrated, for the first time, that the star-shaped architecture of the polymer with highly reducible (hydrazide) functional groups enables its easy collection by inducing reduction-mediated precipitation, allowing us to use it as a “standalone adsorbent” for PM recovery.

(2) Thanks to the standalone usability of S-PAcH as a PM adsorbent, our present work provided new insights into the effects of the architecture (linear vs star-shape) and chemistry (hydrazide vs ethyleneimine vs allylamine) of the standalone polymer on PM adsorption performance and mechanism, which have not been identified previously. Because polymers are typically anchored onto porous substrates in most previous studies, the effects of the architecture and chemistry of standalone polymers on adsorption performance have not been investigated. By contrast, we clearly identified the structure-property-adsorption performance relationship of the standalone polymer adsorbent by performing systematic and comprehensive experiments. As a result, we found that the star-shaped polymer architecture with hydrazide chemistry can achieve superior PM adsorption performance and easier recovery compared with its linear counterpart and the other chemistries.

(3) We demonstrated, for the first time, that S-PAcH can achieve unprecedented PM adsorption capacity and selectivity as well as successful recycling/upcycling, which are not attainable by previously developed adsorbents. Whereas our previous study used S-PAcH as a building material of the membrane for Cr(IV) removal, our present study employed S-PAcH as a standalone adsorbent for PM recovery. Particularly, the present study clearly demonstrated the successful use of S-PAcH in PM adsorption in terms of its high PM adsorption capacity, selectivity, and recovery/upcycling.

(4) Our present study provided new insights into the PM adsorption mechanism that can be utilized for developing standalone adsorbents; how the star-shape architecture of the hydrazide-functionalized polymer can perform as a standalone adsorbent. Specifically, based on the proposed PM adsorption mechanism of S-PAcH, S-PAcH can achieve excellent PM adsorption capacity and selectivity while inducing rapid and strong precipitation owing to its high reduction capability combined with its adsorption mechanism *via* electrostatic and

chelation interactions. This finding highlights the beneficial feature of S-PAcH with both adsorbent and reductant functions, which synergistically improve its PM recovery efficiency and selectivity above those achievable by amine polymers with an adsorption function only or reducing agents with a reduction function only. Please also refer to our responses to the comments 2-1, 2-4, and 2-8.

In the revised manuscript, we strengthened the novelty of our study as follows.

(1) We clarified the novel point of our study by rephrasing the following sentence, “In this study, we synthesize a star-shaped, hydrazide-functionalized polymer (poly(acryloyl hydrazide), S-PAcH)²⁶ and demonstrate, for the first time, that it can be used as a standalone adsorbent to achieve highly efficient, selective, and rapid PM recovery.” (Lines 62-64), together with citing our previous work [doi.org/10.1016/j.cej.2022.137883] a new reference [26].

(2) We emphasized the novelty of our study by adding the following sentence, “Based on these experiments, we identified the effects of the architecture and chemistry of the standalone polymer on PM adsorption performance, which have not yet been investigated.” (Lines 75-77).

(3) We emphasized the novelty of our study by adding the following sentence, “Furthermore, the structural and physicochemical properties of S-PAcH before and after PM adsorption were comprehensively analyzed to identify its PM adsorption mechanism that can be utilized for designing standalone adsorbents.” (Lines 77-80).

Comment 2-12: The level of rigor in academic paper writing currently falls short of the desired standards.

Response to Comment 2-12: We appreciate the reviewer’s critical and constructive comment. First above all, we made our best efforts to **enhance the level of academic writing through proofreading by a professional editing agency ‘Editage’, as shown in the certificate.**

editage helping you get published

Since 2002, Editage has helped over 430,000 authors publish around 1.2 million research papers in scholarly journals across over 1000 disciplines through editorial, translation, transcription, and publication support services. Editage is a brand of Cactus Communications (cactusglobal.com), a science communication and technology company.

GLOBAL : +1(833) 979-0061 | request@editage.com **KOREA :** 02-3478-4396 | submit-korea@editage.com

CACTUS

editage.com | editage.co.kr | editage.jp | editage.cn | editage.com.br | editage.com.tw | editage.de

We also significantly enhanced the academic level and rigor of our manuscript by thorough revisions as follows.

- (1) We clarified the PM adsorption mechanism of our S-PAC_H adsorbent and provided new insights into the adsorption mechanism that can be utilized for developing standalone adsorbents, as responded to the above comment 2-1.
- (2) We further demonstrated the higher PM recovery performance and cost effectiveness of S-PAC_H compared with commercial amine polymers and reducing agents, as responded to the above comments 2-4 and 2-8. We also carefully performed the sustainability and economic analysis of our proposed strategy as responded to the above comment 2-4.
- (3) We further highlighted the practical feasibility of S-PAC_H by demonstrating its high PM adsorption capacity/selectivity and reusability with real-world leachate solutions, as responded to the above comments 2-3, 2-4, and 2-8.
- (4) We strengthened the academic depth of our manuscript by characterizing the catalytic activity of PM/S-PAC_H (as responded to the above comment 2-9) and the pH effect on the PM recovery performance of S-PAC_H (as responded to the below comment 2-14).

Please carefully refer to our responses to all your comments. We strongly believe that the academic level of our revised manuscript can sufficiently meet the standards of this journal.

Comment 2-13: In metal reduction processes, such as the direct reduction of Au from trivalent to zero valence, without transitioning to a monovalent state, have the authors paid attention to the intermediate state?

Response to Comment 2-13: We really appreciate the reviewer's insightful comment, which allowed us to carefully analyze the PM adsorption mechanism of S-PACH. In aqueous media, Au generally exists as three oxidation states (different valences), which are Au(0) (zerovalent), Au(I) (monovalent), and Au(III) (trivalent) [Lingane et al., *J. Electroanal. Chem.* 4 (1962) 332]. As shown in original Fig 2e, the XPS spectrum of the Au/S-PACH(L) precipitate displayed only Au(III) and Au(0) peaks with no Au(I) peak. This result clearly indicates that AuCl₄⁻ ions (corresponding to trivalent Au(III)) were reduced exclusively to Au(0) without transitioning to the intermediate state, AuCl₂⁻ (corresponding to monovalent Au(I)), by S-PACH(L). This result can be explained as follows.

It is known that the reduction reaction with higher standard reduction potential is favored [Dong et al., *Chem. Eng. J.* 283 (2016) 504; Bui et al., *Sep. Purif. Technol.* 248 (2020) 116989]. Hence, as summarized in **Table R7**, compared with the reduction from AuCl₄⁻ to AuCl₂⁻ (from Au(III) to Au(I)), that from AuCl₄⁻ to Au (from Au(III) to Au(0)) is more favorable owing to higher standard reduction potential. Although AuCl₂⁻ (Au(I)) can be generated by the reduction of AuCl₄⁻ (Au(III)), it would be readily reduced to Au (Au(0)) because very high reduction potential (1.15 V) is involved [Dong et al., *Chem. Eng. J.* 283 (2016) 504; Bui et al., *Sep. Purif. Technol.* 248 (2020) 116989]. Moreover, AuCl₂⁻ (Au(I)) is known as an unstable species, and thus, easily reduced to Au (Au(0)) [Gebremichael et al., *Appl. Surf. Sci.* 567 (2021) 150743]. Therefore, it is reasonable to claim that the intermediate Au(I) state is absent in the Au/S-PACH(L) precipitate.

Table R7. Standard reduction potential of the reduction reactions of Au species [Lingane et al., *J. Electroanal. Chem.* 4 (1962) 332].

Reaction	Standard reduction potential (V)
$\text{AuCl}_4^- + 2\text{e}^- \rightarrow \text{AuCl}_2^- + 2\text{Cl}^-$	0.93
$\text{AuCl}_4^- + 3\text{e}^- \rightarrow \text{Au} + 4\text{Cl}^-$	1.00
$\text{AuCl}_2^- + \text{e}^- \rightarrow \text{Au} + 2\text{Cl}^-$	1.15

Comment 2-14: In real waste solutions containing precious metals, various other ions are present, and the pH levels can vary significantly, including the presence of strong acids and bases. Have the authors taken these practical situations into consideration?

Response to Comment 2-14: We appreciate the reviewer's thoughtful comment. We agree with the reviewer that it is important to consider the effects of pH and coexisting ions on PM recovery.

<pH effect>

We already provided the *R_e* data of the polymers for all PMs in a wide pH range of 1 (acidic)–10 (basic) in original Supplementary Figs. 16–18. It was evident that S-PACH(L) exhibited high *R_e* (>99%) for all PMs at pH ≤8. The lower *R_e* of S-PACH(L) under basic conditions (pH >8)

can be attributed to reinforced electrostatic repulsion between S-PAcH(L) and anionic PM species [Ramesh et al., *Bioresour. Technol.* 99 (2008) 3801], as mentioned in original Supplementary Text. It should be noted that ~ 0 R_e value for Pd at $\text{pH} \geq 5$ is attributed to the precipitation of Pd ions [Park et al., *J. Hazard. Mater.* 181 (2010) 797], as mentioned in original Supplementary Text. Importantly, S-PAcH(L) displayed significantly higher R_e for all PMs at any pH investigated (original Supplementary Figs. 16–18).

Most PM recovery processes occur under strong acidic conditions (pH 0–2) [Lin et al., *J. Mater. Chem. A* 5 (2017) 13557; Jung et al., *Chem. Eng. J.* 438 (2022) 135618; Das, *Hydrometallurgy* 103 (2010) 180] because PMs are typically extracted by using strong acids such as concentrated HCl and aqua regia [Chaudhuri et al., *J. Clean. Prod.* 434 (2024) 139912; Ren et al., *Chem. Eng. J.* 479 (2024) 147585; Zupanc et al., *Angew. Chem. Int. Edit.* 62 (2023) e202214453]. Hence, we further evaluated the R_e of S-PAcH(L) for PMs under more acidic conditions ($\text{pH} < 1$). Because a pH meter is able to detect pH up to 0, we used the HCl molar concentration (M_{HCl}) as a measure of acidity. Experiments were performed as follows.

PM (50 mg L^{-1}) solutions were prepared by diluting the respective PM (1000 mg L^{-1}) standard solution with DI water while adjusting their M_{HCl} to 1–5 M using a HCl (37%) aqueous solution. S-PAcH(L) (10 mg) was added to the PM solutions (50 mL) and stirred at 200 rpm for 3 h. The mixture was centrifuged at 10000 rpm for 10 min, and the supernatant (0.5 mL) was diluted 100-fold using DI water. The diluted solution was then filtered through a PSF ultrafiltration membrane, and the supernatant was collected. The PM ion concentrations of the solutions obtained before and after the addition of S-PAcH(L) were measured using an ICP-OES to quantify the R_e .

As shown in **Fig. R14**, S-PAcH(L) maintained its very high R_e ($>99\%$) for all PMs at 1M HCl (corresponding to ~ 0 pH). However, its R_e value for all PMs slightly decreased at 3M HCl and drastically decreased at 5M HCl. We attributed the reduced R_e of S-PAcH(L) under extremely acidic conditions ($>1\text{M}$ HCl) to the hydrolysis of the hydrazide groups of S-PAcH [Zhang et al., *J. Mater. Chem. A* 6 (2018) 10217]. Nevertheless, **we can guarantee the very high R_e ($>99\%$) of S-PAcH(L) under strong acidic conditions (up to pH 0) corresponding to the pH levels of real waste solutions containing PMs.**

Fig. R14. Recovery efficiency (R_e) of S-PAcH(L) for (a) Au, (b) Pd, and (c) Pt as a function of solution HCl concentrations (S-PAcH(L) concentration = 0.2 g L^{-1} , $C_i = 50 \text{ mg L}^{-1}$, contact time = 3 h).

<Coexisting ion effect>

In addition, as we responded to the above 2-3 and 2-8 comments, we confirmed that S-PAcH(L) completely recovered PMs (i.e., $\sim 100\%$ R_e) from all the real-word CPU and spent catalyst leachate solutions without absorbing coexisting metal ions (response to the comment 2-8) and

organic pollutants (response to the comment 2-3). This result clearly highlights the excellent selectivity of S-PAcH(L) towards PMs over coexisting metal ions and organic pollutants, confirming its practically feasible, excellent PM recovery performance and selectivity.

In the revised manuscript, we included the comment on the pH effect on the PM recovery performance of PAcH(L) as follows, “S-PAcH(L) also maintained its very high R_e (>99%) for all PMs even at 1M hydrochloric acid (HCl) (corresponding to pH ~0) (Supplementary Fig. 22 and Supplementary Text).” (Lines 192-193) and “Likewise, the lower R_e of S-PAcH(L) under basic conditions (pH >8) can be attributed to reinforced electrostatic repulsion between S-PAcH(L) and anionic PM species⁹. The Pd adsorption tests were valid only at pH <5 because Pd ions are precipitated at pH ≥ 5 ¹⁰. S-PAcH(L) maintained its very high R_e (>99%) for all PMs even at 1M HCl (corresponding to ~0 pH) (Supplementary Fig. S22). However, its R_e value for all PMs slightly decreased at 3M HCl and drastically decreased at 5M HCl. We attributed the reduced R_e of S-PAcH(L) under extremely acidic conditions (>1M HCl) presumably to the hydrolysis of the hydrazide groups of S-PAcH¹¹.” (Supplementary Text), together with providing the above Fig. R14 as a new Supplementary Fig. 22.

Reviewer #3

General comment: In their work, the Choi and Lee research groups introduce two star-shaped polymers designed for the efficient recovery of precious metals, including gold, palladium, and platinum. These polymers are synthesized via ATRP polymerization using functionalized beta-cyclodextrin molecular skeletons. The incorporation of hydrazide groups onto the polymer chains grants them excellent reduction capabilities, a crucial factor in precipitating precious metals through reduction reactions. The authors evaluated the performance of their star-shaped polymers against other amino-based polymers and observed a significant improvement in adsorption capacity, selectivity, and kinetics for precious metal precipitation. While the reported findings are promising, the reviewer recommends against publication of this manuscript in Nature Communications due to the following concerns:

Response to General Comment: We greatly appreciate the reviewer for many valuable and insightful comments, which helped us to significantly improve the scientific and practical impacts of our manuscript. We carefully addressed all the issues raised by the reviewer. Please refer to our responses to the reviewer's comments as follows.

Comment 3-1: Mischaracterization of Polymer Function: The manuscript describes the star-shaped polymers as adsorbents, but they appear to function primarily as reducing agents. Comparing their efficacy with that of amino-based polymers, which lack reduction capabilities, is an inappropriate comparison. It would be more appropriate to compare these polymers with established reducing agents like hydrazine, particularly in terms of cost-effectiveness and efficiency.

Response to Comment 3-1: Thank you for the insightful comment. We fully agree with the reviewer that our star-shaped polymer (S-PAcH) needs to be compared with conventional reducing agents as well as commercial amine polymers for the following reason.

As mentioned in the original manuscript (Lines 206-209), our S-PAcH adsorbs PMs via both “electrostatic and chelation interactions (like adsorbents)” and “reduction (like reducing agents)”. Both the adsorbent and reductant characters of S-PAcH were clearly demonstrated by XPS analysis showing that the PM species adsorbed in S-PAcH exist as both ionic and reduced metal states (original Fig. 2e and Supplementary Fig. 11). Hence, it would be appropriate to compare our S-PAcH with both polymer adsorbents and reducing agents. Because we already extensively compared S-PAcH with other polymer adsorbents in the original manuscript, we additionally compared S-PAcH with widely used reducing agents such as hydrazine and sodium borohydride (NaBH_4) in terms of PM recovery efficiency (R_e), selectivity, and cost-effectiveness as follows.

1. Comparison of PM recovery efficiency/selectivity

We first characterized and compared the PM reduction capabilities of our S-PAcH(L) and reducing agents (hydrazine and NaBH_4) by examining PM-ion (200 mg L^{-1})-containing aqueous solutions, to which the same dose (0.2 g L^{-1}) of S-PAcH(L), hydrazine, and NaBH_4 was added, as shown in **Fig. R1**. Like S-PAcH(L), both reducing agents reduced all PM ions to PM NPs, which were precipitated, except for hydrazine in the Pt-ion-containing solution. This result indicates that hydrazine has a significantly lower Pt reduction capability than S-PAcH(L) and NaBH_4 , which was also supported by the previous finding that highly concentrated hydrazine is required for reducing Pt ions [Wu et al., *Angew. Chem. Int. Ed.* 60 (2021) 17587]. Consistently, we observed that Pt ions were reduced to Pt NPs at a very high

hydrazine concentration (150 g L^{-1}), as shown in **Fig. R2**.

Fig. R1. Photographs of the PM (200 mg L^{-1}) aqueous solutions after the addition of S-PAcH(L) and reducing agents (hydrazine and NaBH_4) (S-PAcH(L) and reducing agent concentration = 0.2 g L^{-1} , solution pH = 2, contact time = 3 h).

Fig. R2. Photograph of the Pt (200 mg L^{-1}) aqueous solution after the addition of 150 g L^{-1} hydrazine (solution pH = 2, contact time = 3 h).

Based on the experiments, the R_e of S-PAcH(L) and reducing agents (hydrazine and NaBH_4) for all PMs was estimated and compared, as shown in **Fig. R3**. Both reducing agents exhibited remarkably lower R_e for all PMs compared with S-PAcH(L). The significantly low Pt reduction capability of hydrazine [Wu et al., *Angew. Chem. Int. Ed.* 60 (2021) 17587] can account for its marginal R_e for Pt ($\sim 0\%$) under the given conditions. **The higher PM recovery efficiency of**

S-PAcH(L) than those of the reducing agents can be attributed to its high reduction capability combined with its adsorption mechanism *via* electrostatic and chelation interactions. This result highlights the beneficial feature of S-PAcH(L) with both adsorbent and reductant functions, which synergistically improve PM recovery performance above that achievable by reducing agents with a reduction function only.

Fig. R3. Recovery efficiency (R_e) of S-PAcH(L) and reducing agents (hydrazine and NaBH₄) for Au, Pd, and Pt (S-PAcH(L) and reducing agent concentration = 0.2 g L⁻¹, initial PM ion concentration = 200 mg L⁻¹, solution pH = 2, contact time = 3 h).

The PM selectivity of the reducing agents was also characterized and compared with that of S-PAcH(L) using the same simulated feed solutions (CPU and spent catalyst leachates and groundwater) and protocols described in the original manuscript, as shown in **Fig. R4**. As demonstrated in our original manuscript, for all the simulated feed solutions, S-PAcH(L) exhibited ~100% R_e for all PMs but negligible R_e for other coexisting metal cations, indicating its excellent selectivity toward all PMs. In contrast, both the reducing agents displayed significantly lower selectivity toward PMs than S-PAcH(L) as follows.

(1) Simulated CPU leachate: Compared with S-PAcH(L), both reducing agents exhibited lower R_e for Au and remarkably higher R_e for Cu (31.4 % for hydrazine and 26.4% for NaBH₄) (Fig. R4a), indicating their lower selectivity toward Au. Together with Au ions, Cu ions with high reduction potential (0.34 V) were likely to be reduced to Cu NPs and precipitated by reducing agents [Lisiecki and Pileni, *J. Am. Chem. Soc.* 115 (1993) 3887]. In fact, it has been reported that reducing agents such as hydrazine and NaBH₄ can reduce co-existing metal cations (*e.g.*, Cu²⁺, Fe³⁺, and Pb²⁺) as well as PM ions in e-waste leachates [Awadalla and Ritcey, *Sep. Sci. Technol.* 26 (1991) 1207; Chen and Lim, *Chemosphere* 49 (2002) 363; Medding and Lander, *Precious Metals 1981*, Pergamon (1982) 3], resulting in low selectivity toward PMs.

(2) Simulated spent catalyst leachates: Both reducing agents exhibited significantly lower R_e for Pd and Pt (even negligible R_e for Pt in the case of hydrazine) than S-PAcH(L) (Figs. R4b and c). Interestingly, despite the low reduction potential of Al (-1.66 V), NaBH₄ exhibited significantly higher R_e for Al (41.5% and 19.8% for the simulated Pd and Pt spent catalyst leachates, respectively) than S-PAcH(L). Because borohydride ions (BH₄⁻) can act as a stabilizer for metal NPs [Arvizo et al., *Chem. Soc. Rev.* 41 (2012) 2943; Deraedt et al., *Chem. Commun.* 50 (2014) 14194], cationic Al ions were likely to be adsorbed onto anionic BH₄⁻ ion-bound Pd or Pt NPs *via* electrostatic attraction, and thus, recovered with PM NPs. Therefore,

for the simulated spent catalyst leachates, both reducing agents exhibited significantly lower selectivity toward Pd/Pt compared with S-PAcH(L).

(3) Simulated groundwater: Both reducing agents selectively recovered PMs by preferentially reducing PMs with significantly higher reduction potentials than those of co-existing metal ions, except for Pt recovery by hydrazine whose Pt reduction capability is low. As a result, for the simulated groundwater, both reducing agents exhibited high selectivity toward PMs, comparable to that of S-PAcH(L), except for hydrazine whose selectivity toward Pt was marginal owing to its low Pt recovery efficiency.

These experimental results clearly demonstrate **the superior PM selectivity of S-PAcH(L), which can also be attributed to its strong reduction capability combined with electrostatic repulsion between positively charged S-PAcH(L) and co-existing metal cations**, as mentioned in the original manuscript (Lines 226-230). This result further emphasizes the beneficial feature of S-PAcH, whose combined adsorbent and reductant functions enables selective PM recovery. The superior PM recovery performance and selectivity of S-PAcH(L) compared with reducing agents were further confirmed using real-world CPU and spent catalyst leachate feed solutions, as responded to the below comment 3-5. (Please also refer to our response to the comment 3-5).

Fig. R4. Recovery efficiency (R_e) of S-PAcH(L) and reducing agents (hydrazine and NaBH₄) with simulated leachate and groundwater feed solutions: (a) CPU leachate (Au), (b) spent Pd catalyst leachate, (c) spent Pt catalyst leachate, and (d-f) groundwater containing (d) Au, (e) Pd, and (f) Pt.

2. Comparison of cost effectiveness

It is very difficult to accurately estimate the price of S-PAcH(L), which is a lab-scale sample, because the accurate information of additional production costs (e.g., facilities, plant planning,

labor, etc.) and the industrial-grade prices of raw materials is not available for us, who are academic field researchers. Nevertheless, we made our best efforts to evaluate the final price of S-PAcH(L) based on the reagent-grade prices of the raw materials required for its lab-scale synthesis.

Table R1 summarizes the amounts and unit prices of the raw materials used for synthesizing S-PAcH(L) of 1 kg. The pure price of S-PAcH(L) calculated based on its raw material prices was ~\$6961.3 kg⁻¹. If we assume that raw material costs occupy approximately 60–90% of the total production cost [Meneses et al., *Heliyon* 8 (2022) e09028], the final price of S-PAcH(L) can be reasonably estimated to be \$7734.8–11602.2 kg⁻¹. However, this calculation is a very rough estimation, and the accurate total price should be estimated by manufacturers.

Table R1. Unit prices, used amounts, and respective costs of the raw materials used for synthesizing S-PAcH(L) of 1 kg.

Raw material	Manufacturer	Unit price	Used amount	Cost
CDx	Sigma-Aldrich	\$997.4 kg ⁻¹	3.6 g	\$3.6
BiBr	Sigma-Aldrich	\$409.4 kg ⁻¹	24.5 g	\$10.0
CuBr	Sigma-Aldrich	\$3401 kg ⁻¹	8.3 g	\$28.2
MAc	Sigma-Aldrich	\$52.8 L ⁻¹	4.6 L	\$242.8
Alumina	Sigma-Aldrich	\$156.5 kg ⁻¹	1.3 kg	\$195.6
TBABr	Sigma-Aldrich	\$1706.8 kg ⁻¹	1.0 kg	\$1706.8
Hydrazine hydrate	Sigma-Aldrich	\$379.8 kg ⁻¹	7.6 kg	\$2886.5
Na ₂ CO ₃	Daejung Chemical	\$141.4 kg ⁻¹	46.1 g	\$6.5
DCM	Daejung Chemical	\$6.8 L ⁻¹	214.3 mL	\$1.5
THF	Daejung Chemical	\$21.2 L ⁻¹	30.0 L	\$634.6
Methanol	Daejung Chemical	\$7.2 L ⁻¹	162.5 L	\$1175
NMP	Daejung Chemical	\$89.7 L ⁻¹	29.1 mL	\$2.6
Silica	Alfa Aesar	\$53.8 kg ⁻¹	1.3 kg	\$67.3
DI water	Millipore	\$1.5 L ⁻¹	214.3 mL	\$0.3
Total cost:				\$6961.3 kg ⁻¹
Expected final price ^a :				\$7734.8 –11602.2 kg ⁻¹

^aEstimated with the assumption that raw material costs occupy approximately 60–90% of the total production cost [Meneses et al., *Heliyon* 8 (2022) e09028].

To fairly assess the cost effectiveness of S-PAcH(L), hydrazine, and NaBH₄, their costs required for recovering the same amount of PM were compared. Hence, we first identified the doses of the reducing agents that could achieve the same R_e as that achieved by the dose (0.2 g L⁻¹) of S-PAcH(L) at a given PM ion concentration (200 mg L⁻¹), as shown in **Fig. R5**.

Although both reducing agents exhibited lower R_e for all PMs compared with S-PAcH(L) at the same dose (0.2 g L⁻¹), their R_e values increased with their doses, except for hydrazine whose R_e for Pt remained marginal (~0%) even at the highest dose investigated (5.0 g L⁻¹) owing to its very low Pt reduction capability as demonstrated above.

Fig. R5. Recovery efficiency (R_e) of S-PAcH(L) (0.2 g L^{-1}) and reducing agents (hydrazine and NaBH_4) for PMs as a function of reducing agent doses (initial PM ion concentration = 200 mg L^{-1} , solution pH = 2, contact time = 3 h).

Based on the prices of materials and their doses required for achieving the same R_e at a given PM ion concentration of 200 mg L^{-1} (determined from **Fig. R5**), we calculated their costs for PM recovery, as summarized in **Table R2**. The material price of S-PAcH(L) was notably higher than those of hydrazine and NaBH_4 . Nevertheless, the cost of S-PAcH(L) for PM recovery was similar or even lower compared with hydrazine and slightly higher compared with NaBH_4 owing to the superior PM adsorption capacity of S-PAcH(L).

We also need to consider the PM selectivity and recoverability of adsorbents and reducing agents to assess their cost effectiveness for practical PM recovery. The significantly lower PM selectivity of reducing agents than that of S-PAcH(L) would incur additional costs for PM separation. Furthermore, the small molecular size of reducing agents (unreacted and/or reacted forms) renders their recovery/collection difficult [Chen et al., *Chemosphere* 49 (2002) 363], which requires additional costs for purification/recovery processes. Particularly, in a PM reduction process using NaBH_4 , toxic impurities, such as boric acid (H_3BO_3) and sodium metaborate (NaBO_2), are generated, which should be removed by employing additional separation processes [Chen et al., *Chemosphere* 49 (2002) 363].

Although detailed and more extensive economic analysis is needed, we believe that **our S-PAcH polymers with higher PM adsorption capacity/selectivity and recoverability would be more cost-effective compared with reducing agents**, based on the above considerations.

Table R2. Prices, required doses, and respective costs of the materials used for PM recovery.

Materials	Manufacturer	Material price (\$ kg ⁻¹)	Dose (g L ⁻¹)			Cost (\$ L ⁻¹)		
			Au	Pd	Pt	Au	Pd	Pt
S-PAcH(L)	Lab-made	7734.8–11602.2	0.2	0.2	0.2	1.55–2.32	1.55–2.32	1.55–2.32
Hydrazine	Sigma-Aldrich	379.8	5.0	>5.0	>>5.0	1.90	>1.90	>>1.90
NaBH_4	Daejung Chemical	157.7	>5.0	>5.0	0.36	>0.79	>0.79	0.06

In the revised manuscript, we emphasized the advantages of our S-PAcH over conventional reducing agents as follows.

(1) We included the comments on the superior PM recovery efficiency of S-PAcH(L) and its underlying mechanism as follows, “S-PAcH(L) was also more effective at recovering PMs than conventional reducing agents such as hydrazine and sodium borohydride (NaBH₄) (Supplementary Figs. 24–26 and Supplementary Text). This result highlights the beneficial feature of S-PAcH(L) with both adsorbent and reductant functions, which synergistically improves PM recovery performance above that achievable by amine polymers with an adsorption function only or reducing agents with a reduction function only; the unprecedentedly high-capacity and rapid PM adsorption of S-PAcH(L) can be attributed to its high reduction capability combined with its effective adsorption mechanism *via* strong electrostatic and chelation interactions, endowed by its numerous hydrazide groups that are effectively packed in a star-shaped configuration.” (Lines 231-240), together with providing the above Figs. R1–R3 and related explanations as new Supplementary Figs. 24–26 and Supplementary text.

(2) We included the comments on the superior PM selectivity of S-PAcH(L) and its underlying mechanism as follows, “Compared with reducing agents that can reduce coexisting cations as well as PM ions⁵¹, PAcH(L) also exhibited significantly higher selectivity toward PMs (Supplementary Fig. 28 and Supplementary Text), further highlighting the benefit of its both adsorbent and reductant functions in selective PM recovery.” (Lines 261-265), together with providing the above Fig. R4 and related explanations as new Supplementary Fig. 28 and Supplementary text.

(3) We included the comment on the cost effectiveness of our S-PAcH(L) compared with reducing agents as follows, “Moreover, considering the difficulty in recovering small molecular-sized reducing agents⁵¹, we believe that our PAcH with higher PM adsorption performance and recoverability would be more cost-effective at recovering PMs compared with reducing agents (Supplementary Fig. 29, Supplementary Tables 7 and 8, and Supplementary Text).” (Lines 265-269), together with providing the above Fig. R5, Tables R1 and R2, and related explanations as new Supplementary Fig. 29, Supplementary Tables 7 and 8, and Supplementary text.

Comment 3-2: Lack of Evidence for Polymer Recovery and Reusability: They claim that these polymers are recoverable adsorbents for precious metals is not substantiated by experimental data. The manuscript does not provide evidence to support the polymers' recovery or reusability, which is a crucial aspect of their proposed application.

Response to Comment 3-2: We appreciate the reviewer’s insightful comment. We agree with the reviewer that the reusability of adsorbents is an important aspect. Hence, we assessed the regeneration ability (reusability) of our S-PAcH polymer adsorbent as follows.

<Reusability>

As mentioned in the manuscript, our proposed S-PAcH polymer adsorbs PMs *via* reduction combined with electrostatic and chelation interactions. Careful reanalysis of its PM adsorption mechanism allowed us to reveal that the primary amine (–NH₂) groups of S-PAcH are oxidized to –NO₂ groups when they reduce PM ions to PM NPs, as appended at the end of our response to this comment.

To regenerate PM-adsorbed adsorbents by desorbing PMs, agents with high affinity for PMs, such as thiourea, cyanide, and thiocyanate, are commonly used [Ruan et al., *J. Phys. Chem. C* 121 (2017) 25882]. In our study, we used thiourea and Fe³⁺ to regenerate S-PAcH through PM desorption. Thiourea with high affinity for PMs facilitates the desorption of PM species

(especially PM ions) from PM-adsorbed S-PAC_H (PM/S-PAC_H), while Fe³⁺ with high oxidation ability oxidizes adsorbed PM NPs to PM ions, thus assisting PM desorption [Zhou et al., *Environ. Sci. Technol.* 57 (2023) 3334]. Meanwhile, the reduced form of Fe³⁺ (Fe²⁺) can reduce the –NO₂ groups (oxidized amine groups) of S-PAC_H to primary amine (–NH₂) groups, thus regenerating S-PAC_H [Hofstetter et al., *Environ. Sci. Technol.* 40 (2006) 235].

The reusability of our S-PAC_H was assessed with real feed solutions, which are the leachates of real-world CPU and spent Pd and Pt catalyst samples. Experiments were performed as follows. CPU (Au, Intel) was obtained from an end-of-life computer, and spent catalysts (Pd and Pt, Sigma Aldrich) were obtained after their use in hydrogenation reactions. Real-world leachate feed solutions were prepared by following a previously reported protocol [Zhang et al., *Sep. Purif. Technol.* 292 (2022) 121021]. Each real-world sample (20 g) was immersed in aqua regia (500 mL) for 3 d. The mixture was filtered through a cellulose filter paper (JIS P 3801, pore size = 1 μm, Advantec) to remove undissolved solids and further diluted to 1 L with DI water while adjusting pH to 2 using 1 N NaOH aqueous solution. The PM ion concentrations of the obtained leachate solutions were measured using ICP-OES. PM ion concentrations in the real-world leachates were 9.6 (Au), 10.8 (Pd), and 11.1 (Pt) mg L⁻¹, respectively. Next, S-PAC_H(L) (10 mg) was added to each leachate solution (50 mL) and stirred at 200 rpm for 3 h. The mixture was then filtered through a PSF ultrafiltration membrane, and the supernatant was collected. The PM ion concentration of the permeate solution was measured using an ICP-OES to calculate the *R_e*. The filtrated PM/S-PAC_H(L) was put into the aqueous solution containing thiourea (1M), FeCl₃ (1M), and HCl (1M) and sonicated for 30 min to desorb PM species from PM/S-PAC_H(L). The mixture was filtered through a PSF ultrafiltration membrane, and the PM concentration of the permeate solution (*C_D*) was measured using an ICP-OES to calculate the PM desorption efficiency (*D_e*), as given by:

$$D_e = \frac{C_D}{C_L \times R_e} \times 100 \quad (\text{Eq. R1})$$

where *C_L* is the PM ion concentration of the real-world leachate. The filtrated S-PAC_H(L) was freeze-dried and then used to repeat the above adsorption–desorption process seven times.

Fig. R6 shows the *R_e* and *D_e* values as a function of the number of adsorption–desorption cycles. The *R_e* value of S-PAC_H(L) very slightly decreased with increasing the number of adsorption–desorption cycles; S-PAC_H(L) underwent ~5% (for Au and Pd) and ~14% (for Pt) reductions in its *R_e* after seven adsorption–desorption cycles, corresponding to ~0.7% (for Au and Pd) and ~2% (for Pt) reductions in *R_e* per adsorption–desorption cycle. Compared with other reported PM adsorbents, S-PAC_H(L) exhibited a lower reduction in *R_e* per adsorption–desorption cycle (Table R3), confirming its excellent reusability (sustainable use).

Fig R6. Recovery (*R_e*) and desorption (*D_e*) efficiency of S-PAC_H(L) with real-world leachate

solutions as a function of the number of adsorption–desorption cycles: (a) CPU leachate (Au), (b) spent Pd catalyst leachate, and (c) spent Pt catalyst leachate (S-PAcH(L) concentration = 0.2 g L⁻¹, solution pH = 2, and adsorption time = 3 h).

Table R3. Reductions in R_e per adsorption–desorption cycle of reported PM adsorbents.

Adsorbent	PM	Reduction in R_e per adsorption–desorption cycle	Ref.
ADH@BC hybrid membrane ^a	Au	0.4	[1]
Poly-Cys-g-PDA@GPUF ^b	Au	2.5	[2]
2-Mercaptobenzothiazole-impregnated amine-functionalized resin	Pd	4.8	[3]
AHPP-MOF ^c	Pd	2.1	[4]
MNP-G3 ^d	Pd	1.7	[5]
Poly(allylamine hydrochloride)-modified E. coli	Pt	4.4	[6]
S-PAcH(L)	Au	0.7	This study
	Pd	0.7	
	Pt	2	

^aAdipic dihydrazide-grafted bacterial cellulose hybrid membrane. ^bCysteine polymer brush-grafted polydopamine-modified graphene-based polyurethane foam. ^c4-amino-3-hydroxybenzoic acid-modified Zr-based metal-organic framework. ^dMagnetic nanoparticle modified by third-generation dendrimer. ([1] Zhang et al., *Sep. Purif. Technol.* 292 (2022) 121021; [2] Xue et al., *React. Funct. Polym.* 136 (2019) 138; [3] Sharma and Rajesh, *Chem. Eng. J.* 283 (2016) 999; [4] Tang et al., *Chem. Eng. J.* 407 (2021) 127223; [5] Yen et al., *J. Hazard. Mater.* 322 (2017) 215; [6] Mao et al., *Water Res.* 44 (2010) 5919)

<Reanalyzed PM adsorption mechanism of S-PAcH>

To identify the reliable PM adsorption mechanism of S-PAcH, we carefully considered any factors to cause artifacts. We suspected that water in the atmosphere could be readily adsorbed on highly hydrophilic S-PAcH, possibly leading to misinterpretation of XPS data. Specifically, we speculated that the original broad O1s XPS spectrum of PM/S-PAcH at ~533 eV (Fig. 2g and Supplementary Fig. 15) could be interfered with the adsorbed water whose O1s peak strongly appears at 533.1 eV [Heine et al., *J. Am. Chem. Soc.* 138 (2016) 13246].

To avoid the possible interference of adsorbed water, we performed the XPS analysis of PM/S-PAcH immediately after the sample was vacuum-dried at 50 °C for 48 h. As shown in **Fig. R7**, the reanalyzed O1s XPS spectrum of PM/S-PAcH exhibited the narrower peak at a lower binding energy without displaying the peak at ~533 eV compared with the original counterpart. This result clearly confirms that our previous XPS spectra were contaminated with adsorbed water. Hence, the XPS spectra of PM/S-PAcH were reanalyzed with the sample immediately after vacuum drying.

Fig. R7. Original (top) and reanalyzed (bottom) high-resolution O1s XPS spectra of the PM/S-PAcH(L) precipitates: (a) Au, (b) Pd and (c) Pt.

Fig. R8 shows the reanalyzed N1s, O1s, and C1s XPS spectra of PM/S-PAcH. The N1s XPS peak of PM/S-PAcH was deconvoluted into three peaks at 399.7 ($-\text{NO}_2$), 400.5 (N-metal-N), and 401.9 (**protonated amine**) eV (**Figs. R8a–c**), which were absent for pristine S-PAcH (original Supplementary Fig. 2a) [Zhang et al., *Sep. Purif. Technol.* 292 (2022) 121021]. Compared with the original spectra, the reanalyzed N1s spectra of PM/S-PAcH showed a new peak at 399.7 eV ($-\text{NO}_2$) without displaying the peak at 399.5 eV (N-metal-O). This result indicates that (1) protonated amine and $-\text{NO}_2$ groups and N-metal-N chelation bonding are formed in PM/S-PAcH after PM adsorption.

The deconvolution of the O1s spectra was not informative because both peaks corresponding to $-\text{NO}_2$ and C-O-C were overlapped at 532.3 eV (**Figs. R8d–f**) [Luo et al., *Adv. Mater.* 30 (2018) 1706498]. Hence, we focused on the C1s spectrum, which was deconvoluted to two peaks at 284.8 (C-C) and 288.0 (O=C-N) eV for both pristine S-PAcH and PM/S-PAcH (**Figs. R8g–i and R9**) [Zhang et al., *Sep. Purif. Technol.* 292 (2022) 121021]. This result suggests that (2) the oxygen atom in the carbonyl (C=O) group (carbonyl oxygen) of S-PAcH is not protonated in PM/S-PAcH after PM adsorption, unlike our original interpretation from the O1s spectra that protonation on the carbonyl oxygen (C=OH^+) occurs.

Fig. R8. Deconvolution of the high-resolution (a–c) N1s, (d–f) O1s, and (g–i) C1s XPS spectra of the PM/S-PACH(L) precipitates: (a, d, g) Au, (b, e, h) Pd and (c, f, i) Pt.

Fig. R9. Deconvolution of the high-resolution C1s XPS peaks of S-PACH.

Based on the reanalyzed XPS data, we carefully identified the PM adsorption mechanism of S-

PAC_H at low pH, as illustrated in **Fig. R10**. The primary amine ($-\text{NH}_2$) groups of S-PAC_H are protonated preferentially over its secondary amine ($-\text{NH}-$) groups under acidic conditions owing to their stronger basicity (*i.e.*, higher electron-donating ability) [Smith, *Organic chemistry 4th edition*, McGraw-Hill (2014)]. S-PAC_H with protonated amine ($-\text{NH}_3^+$) groups adsorbs anionic PM species (*i.e.*, AuCl_4^- , PdCl_4^{2-} , and PtCl_6^{2-}) via long-range electrostatic interaction [Lin et al., *J. Mater. Chem. A* 5 (2017) 13557], followed by ion-exchange and chelation mainly with the unshared electron-bearing nitrogen atoms of its unprotonated ($-\text{NH}-$) groups (**Fig. R10a**) [Yang et al., *J. Mater. Chem. A* 8 (2020) 3438; Ain et al., *Spectrochim. Acta, Part A* 115 (2013) 683]. Meanwhile, S-PAC_H molecules coagulate owing to their screened electrostatic charges. Subsequently, the $-\text{NH}_3^+$ groups of S-PAC_H are deprotonated to $-\text{NH}_2$ while protonating the adsorbed PM ions via the acid–base reaction between the $-\text{NH}_3^+$ groups (acid) and PM ions (base) (**Fig. R10b**) [Zhang et al., *Sep. Purif. Technol.* 292 (2022) 121021]. The hydrazide groups of S-PAC_H then reduce adsorbed PM ions to NPs while their $-\text{NH}_2$ groups being converted into $-\text{NO}_2$ (**Figs. R10b and c**) [Zhang et al., *Sep. Purif. Technol.* 292 (2022) 121021]. Continuous PM reduction leads to NP growth and induces intra/intermolecular chain fusion through chelation (N–metal–N) between the NPs and unshared electron-bearing nitrogen atoms of $-\text{NH}-$ groups in neighboring PAC_H chains, leading to the rapid formation of large and robust precipitates (**Fig. 10c**). This mechanism is consistent with the presence of XPS spectra corresponding to $-\text{NO}_2$ and N–metal–N for PM/S-PAC_H. Meanwhile, a small fraction of the adsorbed PM species exists as an ionic state in PM/S-PAC_H via electrostatic and chelation interaction (**Fig. R10c**), as evidenced by the presence of the PM ionic XPS peak (original Fig. 2e and Supplementary Fig. 11) and N1s XPS peak corresponding to the protonated amine (**Figs. R8a–c**) for PM/S-PAC_H.

Fig. R10. Revised PM adsorption mechanism of S-PAC_H

In the revised manuscript, we made the following corrections.

(1) We included the protocol of regeneration experiments in the “Methods” section (Lines 467–480).

(2) We included the results and discussion regarding the reusability of our S-PAC_H adsorbent as follows, “Furthermore, S-PAC_H(L) was reusable by desorbing PM from PM/S-PAC_H(L) following a well-established regeneration protocol⁵⁷. The R_e value of S-PAC_H(L) very slightly decreased with increasing the number of adsorption–desorption cycles (Figs. 5g–i); S-PAC_H(L) underwent ~5% (for Au and Pd) and ~14% (for Pt)

reductions in its R_e after seven adsorption–desorption cycles, corresponding to ~0.7% (for Au and Pd) and ~2% (for Pt) reductions in R_e per adsorption–desorption cycle. Compared with other reported PM adsorbents, S-PAcH(L) exhibited a lower reduction in R_e per adsorption–desorption cycle (Supplementary Table 11), confirming its excellent reusability.” (Lines 333-340), together with providing the above Fig. R6 and Table R3 as new Figs. 5g–i and Supplementary Table 11.

(3) We revised the PM adsorption mechanism of S-PAcH as follows, “The N1s XPS peak of PM/S-PAcH was broader than that of the pristine S-PAcH and deconvoluted into three peaks at 399.7 (–NO₂), 400.5 (N–metal–N), and 401.9 (protonated amine) eV²⁰, which were absent for S-PAcH (Fig. 2f and Supplementary Figs. 2 and 16). Deconvolution of the C1s peak revealed two peaks at 284.8 (C–C) and 288.0 (O=C–N) eV for both S-PAcH and PM/S-PAcH (Fig. 2g and Supplementary Fig. 17)²⁸. These results suggest that protonated amines, –NO₂ groups, and N–metal–N chelation bonding are formed while carbonyl oxygen atoms remaining unprotonated in PM/S-PAcH after PM adsorption. Given the results above, the PM adsorption mechanism of S-PAcH at low pH can be depicted as illustrated in Figs. 2h–j. The primary amines (–NH₂) of S-PAcH are protonated preferentially over its secondary amines (–NH–) under acidic conditions owing to their higher basicity (*i.e.*, electron donating nature)³³. S-PAcH with protonated primary amines (–NH₃⁺) adsorbs anionic PM species (*i.e.*, AuCl₄[–], PdCl₄^{2–}, and PtCl₆^{2–}) via long-range electrostatic interactions^{30,40}, followed by ion-exchange and chelation mainly with the unshared electron-bearing nitrogen atoms of its unprotonated –NH– (Fig. 2h)^{14,41}. Meanwhile, S-PAcH molecules coagulate owing to their screened electrostatic charges. Subsequently, –NH₃⁺ of S-PAcH are deprotonated to –NH₂ while protonating the adsorbed PM ions via the acid (–NH₃⁺)–base (PM ion) reaction (Fig. 2i)²⁸. The hydrazide groups of S-PAcH then reduce the adsorbed PM ions to NPs while their –NH₂ being converted into –NO₂ (Figs. 2i and j)²⁸. Continuous PM reduction leads to NP growth and induces intra/intermolecular chain fusion through chelation (N–metal–N) between the NPs and unshared electron-bearing nitrogen atoms of –NH– in neighboring PAcH chains, leading to the rapid formation of large and robust precipitates (Fig. 2j). A small fraction of the adsorbed PM species exists as an ionic state in PM/S-PAcH via ion electrostatic and chelation interaction (Fig. 2j), as evidenced by the ionic PM and N1s (corresponding to the protonated amine) XPS peaks detected for PM/S-PAcH.” (Lines 156-180).

(4) We replaced original Figs. 2f–j with the above Figs. R8a and g and R10 with providing the above Fig. R9 as a new Supplementary Fig. 2c. We also replaced original Supplementary Figs. S14 and S15 with the above Figs. R8b, c, h, and i.

Comment 3-3: Unclear Advantages Over Traditional Reductants: The manuscript does not convincingly demonstrate the advantages of using these star polymers over simpler, more conventional molecular reductants like hydrazine or NaBH₄. This comparison is vital for establishing the novelty and practical utility of the polymers.

Response to Comment 3-3: Thank you for the valuable suggestion, which helped us improve the novelty and practical utility of our S-PAcH polymer for PM recovery. This comment is essentially similar to the above comment 3-1. As responded to the above comment 3-1, we clearly demonstrated that compared with transitional reducing agents such as hydrazine and NaBH₄, our S-PAcH(L) exhibited significantly higher recovery efficiency and selectivity towards PMs. This result highlights the beneficial feature of S-PAcH(L) with both adsorbent

and reductant functions, which synergistically improves PM recovery performance and selectivity above that achievable by reducing agents with a reduction function only. Specifically, **the superior PM recovery efficiency of S-PAcH(L) can be attributed to its high reduction capability combined with its adsorption mechanism *via* electrostatic and chelation interactions.** Furthermore, **the superior PM selectivity of S-PAcH(L) can be attributed to its strong reduction capability combined with electrostatic repulsion between positively charged S-PAcH(L) and co-existing metal cations.** In fact, it has been reported that reducing agents (*e.g.*, hydrazine and NaBH₄) are not appropriate for practical selective PM recovery because they can reduce co-existing metal cations with relatively high reduction potentials (*e.g.*, Cu²⁺, Fe³⁺, and Pb²⁺) as well as PM ions in e-waste leachates [Awadalla and Ritcey, *Sep. Sci. Technol.* 26 (1991) 1207; Chen et al., *Chemosphere* 49 (2002) 363; Medding and Lander, *Precious Metals 1981*, Pergamon (1982) 3]. Furthermore, as we demonstrated in our response to the above comment 3-1, Pt recovery using hydrazine is inefficient because highly concentrated hydrazine is required for reducing Pt ions [Wu et al., *Angew. Chem. Int. Ed.* 60 (2021) 17587].

Moreover, whereas the large size of S-PAcH enables its easy collection by membrane filtration, the small molecular size of reducing agents (unreacted and/or reacted forms) renders their recovery/collection difficult [Chen et al., *Chemosphere* 49 (2002) 363], which requires additional costs for purification/recovery processes. Particularly, in a PM reduction process using NaBH₄, toxic impurities, such as H₃BO₃ and NaBO₂, are generated, which should be removed by employing additional separation processes [Chen et al., *Chemosphere* 49 (2002) 363]. It is also known that hydrazine and NaBH₄ undergo self-oxidation in acidic solutions, limiting their long-time availability [Moliner and Street, *J. Environ. Qual.* 18 (1989) 483; Chen et al., *Chemosphere* 49 (2002) 363].

Therefore, as demonstrated in our response to the above comment 3-1, we believe that **our S-PAcH polymers with higher PM adsorption capacity/selectivity and recoverability would be more cost-effective compared with reducing agents.**

In the revised manuscript, we clearly emphasized the advantages and novel points of our S-PAcH over conventional reducing agents as follows.

(1) We highlighted the advantages of our S-PAcH over conventional reducing agents in the “Abstract” as follows, **“Compared with previously reported PM adsorbents, commercial amine polymers, and reducing agents, S-PAcH exhibited significantly higher adsorption capacity, selectivity, and kinetics toward three PMs (gold, palladium, and platinum) with model, simulated, and real-world feed solutions. The superior PM recovery performance of S-PAcH was attributed to its strong reduction capability combined with its chemisorption mechanism.”** (Lines 24-29).

(2) We included the comments on the superior PM recovery efficiency of S-PAcH(L) and its underlying mechanism as follows, **“S-PAcH(L) was also more effective at recovering PMs than conventional reducing agents such as hydrazine and sodium borohydride (NaBH₄) (Supplementary Figs. 24–26 and Supplementary Text). This result highlights the beneficial feature of S-PAcH(L) with both adsorbent and reductant functions, which synergistically improves PM recovery performance above that achievable by amine polymers with an adsorption function only or reducing agents with a reduction function only; the unprecedentedly high-capacity and rapid PM adsorption of S-PAcH(L) can be attributed to its high reduction capability combined with its effective adsorption mechanism *via* strong electrostatic and chelation interactions, endowed by its numerous hydrazide groups that are effectively packed in a star-shaped configuration.”** (Lines 231-240), together with providing the above Figs. R1–R3 and related explanations

as new Supplementary Figs. 24–26 and Supplementary text.

(3) We included the comments on the superior PM selectivity of S-PaCH(L) and its underlying mechanism as follows, “Compared with reducing agents that can reduce coexisting cations as well as PM ions⁵¹, PaCH(L) also exhibited significantly higher selectivity toward PMs (Supplementary Fig. 28 and Supplementary Text), further highlighting the benefit of its both adsorbent and reductant functions in selective PM recovery.” (Lines 261-265), together with providing the above Fig. R4 and related explanations as new Supplementary Fig. 28 and Supplementary text.

(4) We included the comment on the recoverability and cost effectiveness of our S-PaCH(L) compared with reducing agents as follows, “Moreover, considering the difficulty in recovering small molecular-sized reducing agents⁵¹, we believe that our PaCH with higher PM adsorption performance and recoverability would be more cost-effective at recovering PMs compared with reducing agents (Supplementary Fig. 29, Supplementary Tables 7 and 8, and Supplementary Text).” (Lines 265-269), together with providing the above Fig. R5, Tables R1 and R2, and related explanations as new Supplementary Fig. 29, Supplementary Tables 7 and 8, and Supplementary text.

Comment 3-4: Comparison with Other Precipitation Methods: The selective precipitation of anionic precious metal ions using small hydrophobic anionic species has been documented (Nat. Commu. 2021, 12, 6258). The manuscript fails to adequately compare the proposed polymers with these established methods, particularly in terms of efficiency and reusability.

Response to Comment 3-4: We appreciate the valuable and insightful comment. The suggested reference [Kinsman et al., *Nat. Commun.* 12 (2021) 6258] presents a very innovative precipitation process to selectively recover Au from various acidic feed streams. Specifically, in the reference study, a simple tertiary diamide (denoted by L) was used as a highly selective and recyclable solid precipitant for the precipitation of Au from various aqueous acidic solutions. Addition of solid L to an acidic aqueous solution induces the protonation of L, forming a proton-chelated structure that assembles with anionic Au ions (AuCl_4^-) into insoluble, extended supramolecular structures/clusters ($[\text{HL}][\text{AuCl}_4]$) as precipitates. This precipitation strategy achieved highly selective Au recovery from various feed streams and guaranteed high recyclability of the precipitant L. Despite many beneficial features of the reference study, we can still appeal differentiating and advantageous features of our strategy in terms of working mechanism, methodology, efficiency, and reusability.

1. Working mechanism and methodology

(1) The precipitation process proposed by the reference study relies on the dissolution-precipitation mechanism, and thus, **dissolution can significantly affect the kinetics and efficiency** of metal uptake (adsorption) owing to the limited solubility of L in water. Because of the dissolution-controlled mechanism, the conditions/processes to facilitate the dissolution of L need to be used to enhance its adsorption rate and capacity. **By contrast, in our proposed precipitation strategy, the dissolution of our S-PaCH adsorbent in water is unimpeded owing to its high water solubility, and thus, its adsorption kinetics and capacity are marginally affected by dissolution.**

(2) The reference study induces precipitation by forming the supramolecular cluster between the proton-chelated structure and Au ions (AuCl_4^-) *via* chemical interactions (**supramolecular assembly-mediated precipitation**), whereas our study induces precipitation by forming Au NPs *via* reduction (**reduction-mediated precipitation**).

In conclusion, although both the reference and our studies rely on a precipitation process for recovering Au, the underlying mechanism is significantly different from each other.

2. Efficiency

Unfortunately, it was difficult to quantitatively compare the adsorption efficiency (capacity and kinetics) of our S-PAcH with that of the reference material (L) because the maximum adsorption capacity and equilibrium adsorption time data of L are not reported in the reference paper. Thus, based on the data given in the reference, we roughly calculated the best Au adsorption capacity of L to be $\sim 858 \text{ mg g}^{-1}$. Compared with L, S-PAcH(L) exhibited ~ 3.3 times higher Au adsorption capacity ($\sim 2847 \text{ mg g}^{-1}$), highlighting its superior Au adsorption capacity. Moreover, although detailed Pt and Pd adsorption data are not reported in the reference, the reference material (L) exhibited remarkably low adsorption capacity for Pt and Pd because precipitation was dependent on the ease of formation of chloridometalates and their structures. By contrast, as mentioned in the original manuscript, our S-PAcH(L) displayed significantly higher adsorption capacity for both Pt and Pd compared with commercial amine polymer and other reported PM adsorbents. This result implies that our S-PAcH material is more effective at recovering three PMs (Au, Pd, Pt) than the reference L precipitant although L is highly selective to recover Au.

3. Reusability

In the reference study, the Au ionic species adsorbed in the precipitate can be readily recovered by washing with DI water, which resulted in transport of HAuCl_4 into the solution and the recycling of L. L was successfully reused with three recycling cycles. The easy reusability of L is its significant benefit.

As responded to the above comment 3-2, our S-PAcH was successfully reused/regenerated with seven recycling cycles, demonstrating its excellent reusability. Although more toxic agents (thiourea and Fe^{3+}) are needed to regenerate S-PAcH compared with the reference study, S-PAcH can be recovered *via* a well-established method. Furthermore, S-PAcH can also be upcycled for value-added usages (*e.g.*, catalysts) because it can adsorb PMs as a reduced metal NP form, which is not feasible for the reference diamine precipitant that adsorbs PMs as an ionic form only.

In the revised manuscript, we included detailed comparison between the reference and our studies as follows, “Recently, an innovative precipitation method for Au recovery using a simple tertial diamide compound as a highly Au-selective and recyclable precipitant has been proposed by other researchers¹². This unique strategy induces precipitation by forming a supramolecule between the proton-chelated structure and Au ions *via* chemical interactions, while our approach induces precipitation by forming Au NPs *via* reduction. Although the diamide precipitant selectively recovers Au from acidic solutions, its recovery performance could be significantly affected by its dissolution process owing to its limited solubility in water, unlike our highly water-soluble S-PAcH adsorbent. Moreover, compared with the diamide precipitant displaying low Pt and Pd uptake, our S-PAcH exhibits excellent adsorption capacity and selectivity toward Pt and Pd, indicating its versatile use for PM recovery. Although more toxic agents (thiourea and Fe^{3+}) are needed for regenerating S-PAcH compared with that (deionized (DI) water) needed for recovering the diamide precipitant, S-PAcH can be sustainably reused *via* a well-established regeneration method, as will be demonstrated below. S-PAcH can also be upcycled for value-added usages (*e.g.*, catalysts) because it can adsorb PMs as a reduced metal NP form, which is not feasible for the diamine precipitant that adsorbs PMs as an

ionic form only.” (Lines 308-323).

Comment 3-5: Ineffectiveness at Realistic Metal Concentrations: The manuscript uses unrealistically high concentrations of metals (200 ppm) for the recovery simulations, which do not reflect real-world conditions found in CPU leachates (~20 ppm). Figure S5 suggests that these polymers are ineffective at lower, more realistic concentrations, raising significant concerns about their practical application.

Response to Comment 3-5: We appreciate the reviewer for pointing out the critical aspect. Let us explain why our S-PAcH polymer is still effective at recovering PMs at lower, more realistic concentrations as follows. As shown in original Fig. S5, our S-PAcH polymers led to the less extent of precipitation at lower PM concentrations (<50 ppm) compared with the case at higher PM concentrations. Nevertheless, as displayed in original Figs. 3a–c, S-PAcH(L) achieved ~100% R_e for all PMs even at low, realistic PM concentrations (<50 ppm), indicating its effectiveness for practical and realistic PM recovery.

To further confirm this, we closely examined the PM adsorption behavior of S-PAcH(L) at a very low PM concentration (1 mg L^{-1}), as shown in **Fig. R11**. Although microscale precipitates were not formed after the addition of S-PAcH(L) to the PM solutions at the low PM concentration (**Figs. R11a–c**), a distinct increase in the hydrodynamic diameter (H_R) of S-PAcH(L) was observed (**Fig. R11d**). Moreover, the S-PAcH(L)-added Au solution exhibited the characteristic red color of Au NPs, indicating PM reduction by S-PAcH(L). Despite their small sizes, PM/S-PAcH(L) particles were completely screened *via* a PSF membrane (MWCO = 20 kg mol^{-1}), as evidenced by PM/S-PAcH(L) particles collected by the membrane (**Figs. R11e–g**) and clear permeate solutions (**Figs. R11h–j**). This result clearly demonstrates that even without microscale precipitation, a strong reduction mechanism enabled by S-PAcH(L) can achieve high R_e for all PMs at low PM ion concentrations (<50 mg L⁻¹).

Fig. R11. a–c, Photographs of the (a) Au, (b) Pd, and (c) Pt aqueous solutions after the addition of S-PAcH(L) (S-PAcH(L) concentration = 0.2 g L^{-1} , initial PM ion concentration = 1 mg L^{-1} , solution pH = 2, contact time = 3 h). d, Corresponding DLS curves of pristine S-PAcH(L) and

PM/S-PAcH(L). **e–j**, Photographs of the (**e–g**) PM/S-PAcH(L) particles screened by a PSF membrane and (**h–j**) permeate solutions: (**e, h**) Au, (**f, i**) Pd, and (**g, j**) Pt.

To further confirm the effectiveness of our S-PAcH(L), we evaluated its PM recovery efficiency (R_e) with real-world feed solutions, which are the leachates of real-world CPU and spent Pd and Pt catalyst samples. Experiments were performed as follows. CPU (Au, Intel) was obtained from an end-of-life computer, and spent catalysts (Pd and Pt, Sigma Aldrich) were obtained after their use in hydrogenation reactions. Real-world leachate feed solutions were prepared by following a previously reported protocol [Zhang et al., *Sep. Purif. Technol.* 292 (2022) 121021]. Each real-world sample (20 g) was immersed in aqua regia (500 mL) for 3 d. The mixture was filtered through a cellulose filter paper (JIS P 3801, pore size = 1 μm , Advantec) to remove undissolved solids and further diluted to 1 L with DI water while adjusting pH to 2 using 1 N NaOH aqueous solution. Polymer adsorbents (S-PAcH(L), bPEI and PAAm) or reducing agents (hydrazine and NaBH_4) (10 mg) were added to each leachate solution (50 mL) and stirred at 200 rpm for 3 h. The mixture was then filtered through a PSF ultrafiltration membrane, and the supernatant was collected. The metal ion concentrations of the leachate solutions before and after the addition of the adsorbents and reducing agents were measured using an ICP-OES.

The real-world CPU and spent Pd/Pt catalyst leachate solutions contained various coexisting ions; the CPU leachate mainly contained Au (9.6 mg L^{-1}), Ni (84.5 mg L^{-1}), and Cu (102.9 mg L^{-1}) ions while spent Pd and Pt catalysts contained Al (2.9 and 3.7 mg L^{-1} , respectively) as a main coexisting metal ion together with Pd (10.8 mg L^{-1}) and Pt (11.1 mg L^{-1}), respectively (**Fig. R12**).

Fig. R12. a–c, Photographs, SEM, and SEM-EDX mapping images of the (a) CPU and spent (b) Pd and (c) Pt catalysts. d, Metal composition of the CPU and spent Pd and Pt catalyst leachates.

Fig. R13 showed that for all the real-world leachates, S-PAC_H(L) exhibited ~100% R_e for all PMs but negligible R_e for other coexisting metal cations, similar to the case for simulated feed solutions, as demonstrated in the original manuscript. This result highlights the excellent selectivity of S-PAC_H(L) toward all PMs. Compared with S-PAC_H(L), commercial amine polymers (bPEI and PAAm) and reducing agents (hydrazine and NaBH₄) exhibited lower R_e for PMs and higher R_e for coexisting metal ions, and thus, significantly lower selectivity toward PMs (Fig. R13). The superior PM adsorption capability and selectivity of S-PAC_H(L) can be attributed to its high reduction capability combined with its adsorption mechanism *via* electrostatic and chelation interactions. This result highlights the beneficial feature of S-PAC_H(L) with both adsorbent and reductant functions, which synergistically improve PM recovery performance above that achievable by amine polymers with an adsorption function

only or reducing agents with a reduction function only. Particularly, the excellent PM selectivity of S-PAC_H(L) can be attributed to its strong reduction capability combined with electrostatic repulsion between positively charged S-PAC_H(L) and coexisting metal cations, as mentioned in the original manuscript (Lines 226-230).

As a result, we can reasonably claim that our S-PAC_H(L) is effective at recovering PMs at realistic, low PM concentrations.

Fig. R13. Recovery efficiency (R_e) of the polymers (S-PAC_H(L), bPEI, and PAAM) and reducing agents (hydrazine and NaBH₄) with the real-world CPU and spent Pd and Pt catalyst leachate solutions: (a) CPU leachate, (b) spent Pd catalyst leachate, and (c) spent Pt catalyst leachate. (polymer and reducing agent concentration = 0.2 g L⁻¹, solution pH = 2, contact time = 3 h).

In the revised manuscript, we emphasized the effectiveness of our S-PAC_H at low, realistic PM concentrations as follows.

(1) We emphasized the high PM recovery performance of our S-PAC_H(L) even at low initial PM concentrations as follows, “By contrast, S-PAC_H, in particular S-PAC_H(L), exhibited very high R_e (~100%) for all PMs even at low C_i (<50 mg L⁻¹), in which microscale precipitation was not induced (Figs. 3a–c and Supplementary Figs. 6 and 21), demonstrating its superior PM recovery performance.” (Lines 189-191), together with providing the above Fig. R11 and related discussion as a new Supplementary Fig. S21.

(2) We included the protocol of PM recovery experiments with real-world samples in the “Methods” section (Lines 451-466).

(3) We included the results and discussion on the PM recovery performance of S-PAC_H(L) at the realistic conditions with real-world samples as follows, “We further assessed the feasibility of S-PAC_H for its use in PM recovery from the leachates of real-world CPU and spent Pd/Pt catalysts (Figs. 5a–c and Supplementary Fig. 33). S-PAC_H(L) completely recovered PMs (*i.e.*, ~100% R_e) from all the real-world leachate solutions without absorbing coexisting metal ions (Figs. 5d–f) and organic pollutants (Supplementary Fig. 34). Compared with S-PAC_H(L), commercial amine polymers and reducing agents exhibited lower R_e for PMs and higher R_e for coexisting metal ions, and thus, displaying significantly lower selectivity toward PMs (Figs. 5d–f). This result highlights the practically feasible, excellent PM recovery performance and selectivity of S-PAC_H(L), which are attributable to its high reduction capability combined with its electrostatic and chelation interaction-mediated adsorption mechanism.” (Lines 324-333), together with providing the above Figs. R12 and R13 as new Figs. 5a–f and Supplementary Fig. 33.

REVIEWER COMMENTS

Reviewer #1 (Remarks to the Author):

The authors have undertaken substantial efforts in improvising the manuscript based on the inputs provided by all the reviewers. Though the manuscript can now be accepted for publication in the prestigious Nature Communications, there are couple of minor things the authors need to consider/mention in the final version.

1. The recovery of the PMs has now been demonstrated by using a mixture of thiourea and $\text{Fe}^{3+}/\text{HCl}$. This means now the desorbed PM ions would coexist with Fe^{2+} ions and an additional step would be required to obtain the PMs in pure form. Is this correct? If so, the authors need to mention it clearly in the discussion part.

2. The authors have mentioned the revised mechanism of PM ion reduction through the deconvolution of the N 1s narrow scan spectra. Since the deconvolutions also depends on the inputs to a significant extent, the authors need to provide the FWHM of the N 1s scan before and after PM adsorption to see a broadening as a function of reductive adsorption.

Reviewer #2 (Remarks to the Author):

The overall revision of this article demonstrates significant improvement compared to the initial draft. To be frank, the author's literature summary on precious metal recycling is incomplete, particularly given the limited relevant literature in the past three years. Consequently, the performance comparison in this section, as illustrated in Figure 3f, appears relatively weak. It is recommended that the author further enhance this aspect.

Reviewer #3 (Remarks to the Author):

The authors have made commendable efforts in revising the manuscript, resulting in significant improvements over the initial submission. Nevertheless, there are crucial aspects that require further clarification and evidence before the manuscript can be considered for publication.

1. The innovative approach for recycling the starting polymer through regeneration, employing thiourea, Fe^{3+} , and HCl, demonstrates notable performance across seven cycles with minimal loss in recovery efficiency. However, the mechanism outlined in Figures 2h-2j appears overly simplistic and unsupported by the presented data. Specifically, the postulation of a nitro group's involvement lacks direct evidence, relying solely on deconvolution from XPS experiments. If the proposed mechanism holds, an IR absorption band characteristic of the $-\text{NO}_2$ group should be observable. Additionally, the source of oxygen within the $-\text{NO}_2$ group is ambiguous. It is unclear whether it originates from solvent water or atmospheric oxygen. The manuscript should also address why PM ions would not be directly protonated at low pH levels, bypassing the proposed proton transfer via an ammonium intermediate. Given that both PM ions and hydrazine are likely to be protonated under acidic conditions, the rationale behind the suggested reaction pathway warrants further elucidation.

2. Regarding the recycling process, the continuous addition of thiourea and FeCl_3 without subsequent separation raises questions about their impact on the PM ion reduction process. If these reagents are not removed before the next capture cycle, they could potentially desorb captured PM nanoparticles, undermining the recycling efficiency. The manuscript should clarify the process, particularly emphasizing the treatment or removal of desorption agents before polymer reuse, to ensure the described procedure is viable and logical.

In summary, while the manuscript presents a promising approach to PM recycling using start polymers, addressing the aforementioned concerns with additional experimental evidence and clarification will be critical for advancing the manuscript toward publication.

Responses to the Reviewers' Comments

Reviewer #1

General comment: The authors have undertaken substantial efforts in improvising the manuscript based on the inputs provided by all the reviewers. Though the manuscript can now be accepted for publication in the prestigious Nature Communications, there are couple of minor things the authors need to consider/mention in the final version.

Response to General Comment: We thank the reviewer for many positive and constructive comments. We further improved the manuscript by addressing the reviewer's additional comments as follows.

Comment 1-1: The recovery of the PMs has now been demonstrated by using a mixture of thiourea and Fe³⁺/HCl. This means now the desorbed PM ions would coexist with Fe²⁺ ions and an additional step would be required to obtain the PMs in pure form. Is this correct? If so, the authors need to mention it clearly in the discussion part.

Response to Comment 1-1: Thank you for the reviewer's valuable comment. First, let us explain the state of Fe ions during the regeneration of S-PAC_H using thiourea, Fe³⁺ (FeCl₃), and HCl as PM desorption agents. Thiourea with high affinity for PMs facilitates the desorption of PM species (especially PM ions) from PM/S-PAC_H, while Fe³⁺ with high oxidation ability oxidizes adsorbed PM NPs to PM ions, thus assisting PM desorption [Zhou et al., *Environ. Sci. Technol.* 57 (2023) 3334]. Meanwhile, the reduced form of Fe³⁺ (Fe²⁺) can reduce the –NO₂ groups (oxidized amine groups) of S-PAC_H to primary amine (–NH₂) groups, thus regenerating S-PAC_H [Hofstetter et al., *Environ. Sci. Technol.* 40 (2006) 235] while being oxidized to Fe³⁺. Hence, after PM desorption, desorbed PM ions can coexist with desorption agents (Fe³⁺, thiourea, and HCl), which need to be removed *via* additional separation processes to obtain high-purity PMs.

Fe³⁺ ions can be removed *via* precipitation through simple pH adjustment for the following reason. Fe ions are precipitated at pH ≥3 [Stefánsson, *Environ. Sci. Technol.* 41 (2007) 6117], while Pd ions are precipitated at pH ≥5 [Park et al., *J. Hazard. Mater.* 181 (2010) 794], and Au and Pt ions are not precipitated at the pH range of 1–10 [Wang et al., *J. Phys. Chem. C* 113 (2009) 6505; Romero-Freire et al., *Ecotoxicol. Environ. Saf.* 227 (2021) 112924]. The precipitation behavior of PM ions was described in our previous Supplementary Text and Fig. 18. Thus, by adjusting the solution pH to 3–4, we can preferentially precipitate out Fe ions over PM ions. We verified the feasibility of the aforementioned method by adjusting the pH of the desorption solution containing thiourea (1M), FeCl₃ (1M), and HCl (1M). The pristine desorption solution is brown-colored (**Fig. R1a**). When the solution pH was adjusted to 3 using 30% ammonia solution (Daejung Chemical), brown-colored insoluble Fe precipitates and colorless supernatant solution were obtained (**Fig. R1b**). The precipitates were readily collected with a cellulose filter paper (pore size = 1 μm), and the Fe concentration in the permeate solution was measured to be ~0 ppm using ICP-OES. This result suggests that Fe ions can be readily removed by simple pH adjustment.

Fig. R1. Photographs of the PM desorption aqueous solution containing thiourea (1M), iron chloride (FeCl_3 , 1M), and HCl (1M) (a) before and (b) after adjusting the solution pH to 3.

Then, thiourea, HCl, and ammonia remaining in the solution can be removed by thermal treatment because they are completely vaporized (HCl and ammonia) and decomposed (thiourea) at 300 °C [Park et al., *RSC Adv.* 4 (2014) 9118].

In the revised manuscript, we included the comment on the removal of desorption agents including Fe^{3+} ions as follows, “After PM desorption, desorbed PM ions can coexist with desorption agents, which need to be removed *via* additional separation processes to obtain high-purity PMs. Fe^{3+} ions can be readily removed by adjusting the solution pH to 3–4, where Fe ions can be preferentially precipitated over PM ions (Supplementary Figs. 19 and 36 and Supplementary Text)⁶¹. Thiourea and HCl can also be removed by thermal treatment because they are completely vaporized (thiourea) and decomposed (HCl) at 300 °C⁶²” (Page 15, Lines 354-360), together with providing the above Fig. R1 and related discussion as a new Supplementary Fig. 36.

Comment 1-2: The authors have mentioned the revised mechanism of PM ion reduction through the deconvolution of the N 1s narrow scan spectra. Since the deconvolutions also depends on the inputs to a significant extent, the authors need to provide the FWHM of the N 1s scan before and after PM adsorption to see a broadening as a function of reductive adsorption.

Response to Comment 1-2: Thank you for the reviewer’s insightful comment. We agree with the reviewer that a quantitative measure of peak broadness needs to be provided. First above all, let us explain the reliability of full width at half maximum (FWHM) in determining the degree of peak broadness. FWHM, which is defined as the width of a line at half of the maximum amplitude of a spectrum curve, is used to characterize the broadness of “**the peak with Gaussian distribution**” [Wang et al., *Sens. Actuators, B* 360 (2022) 131680]. In the case of our XPS N1s peak, which is formed with the overlap of several sub-component peaks and thus deviates from Gaussian distribution, the FWHM value can lead to misinterpretation of peak broadness [Robeson et al., *IEEE Trans. Nucl. Sci.* 37 (1990) 1506]. This is because the shape of the peak significantly depends on the width, height, and position of its sub-component peaks.

To demonstrate this challenge, we compared the FWHM values of the N1s peaks of S-PACH(L) and PM-adsorbed S-PACH(L) (PM/S-PACH(L)) as shown in **Fig. R2**. The FWHM of PM/S-PACH(L) was similar to or even smaller than that of S-PACH(L). This result indicates the aforementioned limitation of FWHM because FWHM cannot depict the fact that the deconvoluted peaks of PM/S-PACH(L) (399.7, 400.5, and 401.9 eV) distributed in a wider range of binding energy than those of S-PACH(L) (399.8 and 400.9 eV).

Hence, we speculated that **the width at zero point (denoted as peak width) would be a**

more reasonable measure of the broadness of the N1s peak because it can directly reflect the distributed range of deconvoluted peaks. In fact, all PM/S-PAcH(L)s exhibited larger peak width values than that of S-PAcH(L), and each PM/S-PAcH(L) showed nearly the same peak width value in accordance with the results of deconvoluted N1s peaks. This result indicates that a peak width value can reasonably depict the broadness (distribution of deconvoluted peaks) of the N1s peak.

Fig. R2. FWHM and peak width of N1s XPS peak of S-PAcH(L) and PM/S-PAcH(L)s

In the revised manuscript, to avoid the possible misleading interpretation of FWHM, we provided the width at zero point values of S-PAcH(L) and PM/S-PAcH(L) to quantitatively compare their peak broadness as follows, “The N1s XPS peak of PM/S-PAcH was broader (the width at zero point of ~5.8) than that of the pristine S-PAcH (the width at zero point of ~4.5) and deconvoluted into three peaks at 399.7 (–NO₂), 400.5 (N–metal–N), and 401.9 (protonated amine) eV²⁰” (Pages 6-7, Lines 156-159).

Reviewer #2

Comment 2: The overall revision of this article demonstrates significant improvement compared to the initial draft. To be frank, the author's literature summary on precious metal recycling is incomplete, particularly given the limited relevant literature in the past three years. Consequently, the performance comparison in this section, as illustrated in Figure 3f, appears relatively weak. It is recommended that the author further enhance this aspect.

Response to Comment 2: Thank you for the reviewer's thoughtful comment. We agreed with the reviewer that references on PM recovery need to be updated. We added 10 additional references published in the past three years [Wang et al., *Chem. Eng. J.* 468 (2023) 143453; Rasoulzadeh et al., *J. Environ. Chem. Eng.* 9 (2021) 105954; Zeng et al., *ACS Sustainable Chem. Eng.* 10 (2022) 1103; Liu et al., *Sep. Purif. Technol.* 328 (2024) 124925; Maponya et al., *Environ. Nanotechnol. Monit. Manage.* 20 (2023) 100805; Nazri et al., *Sep. Purif. Technol.* 259 (2021) 118197; Maponya et al., *Sep. Purif. Technol.* 307 (2023) 122767; Li et al., *Environ. Res.* 227 (2023) 115814; Guo et al., *J. Hazard. Mater.* 415 (2021) 125617; Ianăși et al., *Materials* 16 (2023) 2837] to Supplementary Tables 6 and 11 and Fig. 3f. **The revised Fig. 3f shown below still demonstrated that compared with other reported PM adsorbents, our S-PACH(L) exhibited significantly higher PM adsorption capacity and shorter adsorption equilibrium times for all three PMs.** As also summarized in revised Supplementary Table 11, compared with other reported PM adsorbents, our S-PACH(L) exhibited a relatively lower reduction in R_e per adsorption–desorption cycle, confirming its excellent reusability.

Revised Fig. 3f. Comparison of the PM adsorption performance of S-PACH(L) with those of other reported PM adsorbents.

In the revised manuscript, we added 10 additional references published in the past three years to Supplementary Tables 6 and 11 and Fig. 3f, together with replacing the original Fig. 3f with the above revised Fig. 3f.

Reviewer #3

General comment: The authors have made commendable efforts in revising the manuscript, resulting in significant improvements over the initial submission. Nevertheless, there are crucial aspects that require further clarification and evidence before the manuscript can be considered for publication.

Response to General Comment: We thank the reviewer for additional constructive and thoughtful comments. We carefully addressed all the concerns raised by the reviewer as follows.

Comment 3-1: The innovative approach for recycling the starting polymer through regeneration, employing thiourea, Fe³⁺, and HCl, demonstrates notable performance across seven cycles with minimal loss in recovery efficiency. However, the mechanism outlined in Figures 2h-2j appears overly simplistic and unsupported by the presented data. Specifically, the postulation of a nitro group's involvement lacks direct evidence, relying solely on deconvolution from XPS experiments. If the proposed mechanism holds, an IR absorption band characteristic of the -NO₂ group should be observable. Additionally, the source of oxygen within the -NO₂ group is ambiguous. It is unclear whether it originates from solvent water or atmospheric oxygen. The manuscript should also address why PM ions would not be directly protonated at low pH levels, bypassing the proposed proton transfer via an ammonium intermediate. Given that both PM ions and hydrazine are likely to be protonated under acidic conditions, the rationale behind the suggested reaction pathway warrants further elucidation.

Response to Comment 3-1: Thank you for the reviewer's critical and insightful comments, which helped us significantly improve our proposed PM adsorption mechanism as follows.

(1) We have proposed the step-by-step PM adsorption mechanism of our S-PAC_H adsorbent based on previously reported PM adsorption mechanisms of amine-functionalized adsorbents; our TEM, XRD, and XPS results clearly support that electrostatic and chelation interactions combined with reduction are involved in PM adsorption by S-PAC_H. It is well known that electrostatic and chelation interactions are one of the key interactions in PM adsorption by amine groups [Gurung et al., *Chem. Eng. J.* 228 (2013) 405]. It is also well documented that amine (-NH₂) groups are oxidized to nitro (-NO₂) groups by reducing PM ions to metal PM [Zhang et al., *Sep. Purif. Technol.* 292 (2022) 121021; Qin et al., *Chem. Eng. J.* 428 (2022) 132493; Xiang et al., *Hydrometallurgy* 215 (2023) 105964], as confirmed by our XPS analysis (Fig. 2f and Supplementary Fig. 16).

To further confirm the formation of -NO₂ groups, we performed FT-IR analysis for pristine S-PAC_H(L) and PM-adsorbed S-PAC_H (PM/S-PAC_H(L)) as shown in **Fig. R1**. While pristine S-PAC_H(L) exhibited the FT-IR peak at 1610 cm⁻¹ (C=O stretching, amide) [Jeon et al., *J. Membr. Sci.* 611 (2020) 118415], all PM/S-PAC_H(L)s displayed peaks at 1655 (C=O stretching, amide) and 1596 cm⁻¹ (N=O stretching, -NO₂) [Xavier and Peridandy, *Spectrochim Acta, Part A* 149 (2015) 216]. It should be noted that the C=O stretching peak of PM/S-PAC_H(L) shifted toward a longer wavenumber compared to that of S-PAC_H(L) because the electron-withdrawing -NO₂ group of PM/S-PAC_H(L) rendered the carbon atom of the C=O group electron-deficient, thus shortening the C=O bond [Singh et al., *Spectrochim. Acta A* 87 (2012) 106]. **The FT-IR result further confirms the conversion of -NH₂ groups to -NO₂ groups during the PM adsorption process.**

Fig. R1. FT-IR spectra of the pristine S-PAC(H)L and PM/S-PAC(H)L precipitates. The precipitates were formed by allowing contact between S-PAC(H)L (0.2 g L^{-1}) and PM (200 mg L^{-1}) aqueous solutions ($\text{pH} = 2$) for 3 h.

(2) It has also been reported that water is an oxygen source of $-\text{NO}_2$ during the oxidation of $-\text{NH}_2$ to $-\text{NO}_2$, as given by $-\text{NH}_2 + 2\text{H}_2\text{O} \rightarrow -\text{NO}_2 + 6\text{H}^+ + 6\text{e}^-$ [Zhang et al., *Sep. Purif. Technol.* 292 (2022) 121021]. Furthermore, water has been utilized as a source of oxygen in various oxidation reactions in aqueous media [Zhang et al., *J. Catal.* 332 (2015) 95; Fukuzumi et al., *Coord. Chem. Rev.* 333 (2017) 44; Cai et al., *New J. Chem.* 41 (2017) 3882; Hirai et al., *Angew. Chem.* 120 (2008) 5856]. **Hence, we can reasonably postulate that the main oxygen source of $-\text{NO}_2$ would be solvent water.**

(3) Regarding the protonation of PM ions, we are deeply grateful for the reviewer's critical comment, which allowed us to carefully reconstruct the PM adsorption mechanism of S-PAC(H)L. First above all, it is well documented that the precursors of PM ions (HAuCl_4 , H_2PdCl_4 , and H_2PtCl_6) are strong acids (*i.e.*, high tendency to donate protons) [Shelimov et al., *J. Am. Chem. Soc.* 121 (1999) 545; Drelinkiewicz et al., *React. Funct. Polym.* 69 (2009) 630; Aghaei et al., *Metals* 7 (2017) 529]. Hence, **it is reasonable to postulate that PM ions exist as deprotonated anionic species (*i.e.*, AuCl_4^- , PdCl_4^{2-} , and PtCl_6^{2-}) in both the bulk aqueous phase and S-PAC(H)L at $\text{pH} 2$** [Wojnicki et al., *Hydrometallurgy* 127–128 (2012) 45; Thanh and Liu, *Colloids Surf., A* 616 (2021) 126326; Shelimov et al., *J. Am. Chem. Soc.* 121 (1999) 545], where PM adsorption tests were performed.

Meanwhile, the hydrazide primary amine ($-\text{NH}_2$) of S-PAC(H)L would presumably be protonated to a less extent than conventional primary amines owing to its lower basicity (*i.e.*, higher $\text{p}K_b$ value) [Yaremenko et al., *Org. Lett.* 24 (2022) 6582]. This was evidenced by the fact that the positive zeta potential of S-PAC(H)L noticeably increased while those of conventional amine polymers were nearly unchanged when pH decreased from 2 to 1 (**Fig. R2**). Hence, **it can be postulated that S-PAC(H)L at low pH (*i.e.*, $\text{pH} 2$) contains both protonated ($-\text{NH}_3^+$) and unprotonated ($-\text{NH}_2$) primary amines (*i.e.*, protonated and unprotonated hydrazides).**

Fig. R2. Zeta potentials of commercial amine (bPEI and PAAm) and PAcH-series polymers as a function of solution pH.

We also need to consider the oxidation–reduction (redox) reaction between S-PAcH and PM ions. Based on the reference [Zhang et al., *Sep. Purif. Technol.* 292 (2022) 121021], the redox reaction, oxidation half-reaction, and reduction half-reaction between the unprotonated hydrazide $-\text{NH}_2$ of S-PAcH and Au ions (AuCl_4^-) can be described as follows.

Based on the above redox reaction mechanism, the reaction between $-\text{NH}_2$ of S-PAcH and Pd ions (PdCl_4^{2-}) can be depicted as follows.

The reaction between $-\text{NH}_2$ of S-PAcH and Pt ions (PtCl_6^{2-}) can also be depicted as follows.

Based on the above reaction mechanisms, it is evident that **one unprotonated $-\text{NH}_2$ group of S-PAcH provides six electrons during its oxidation to $-\text{NO}_2$, thus reducing multiple PM ions (*i.e.*, 2 for AuCl_4^- , 3 for PdCl_4^{2-} , and 1.5 for PtCl_6^{2-}) to PM metals.**

Now, we can reasonably postulate the PM reduction mechanism of S-PAcH as follows. First, the protonated primary amines ($-\text{NH}_3^+$) of S-PAcH would adsorb anionic PM species *via* long-range electrostatic interactions, followed by ion-exchange and chelation mainly with the unshared electron-bearing nitrogen atoms of its unprotonated $-\text{NH}-$ (**Fig. R3a**). Subsequently, the unprotonated primary amines ($-\text{NH}_2$) of S-PAcH would then reduce multiple adsorbed PM ions to NPs while being oxidized to $-\text{NO}_2$ (**Figs. R3b and c**). Continuous PM reduction leads

to NP growth and induces intra/intermolecular chain fusion through chelation (N–metal–N) between the NPs and unshared electron-bearing nitrogen atoms of –NH– in neighboring PAcH chains, leading to the rapid formation of large and robust precipitates (Fig. R3c).

Fig. R3. Revised PM adsorption mechanism of S-PAcH.

In the revised manuscript, we made following revisions.

(1) We included the comment regarding the FT-IR evidence of the formation of –NO₂ as follows, “The formation of –NO₂ groups in S-PAcH after PM adsorption was further confirmed by FT-IR analysis where PM/S-PAcH exhibited the peak at 1596 cm⁻¹ (N=O stretching, –NO₂)⁴⁰, which was absent for S-PAcH (Supplementary Fig. 18)” (page 7, Lines 164-166), together with providing the above Fig. R1 and related discussion as a new Supplementary Fig. 18.

(2) We included the comment on the source of oxygen in –NO₂ as follows, “Subsequently, the unprotonated –NH₂ of S-PAcH then reduce the adsorbed PM ions to NPs while being converted into –NO₂, as given by $-\text{NH}_2 + 2\text{H}_2\text{O} (\text{solvent water}) \rightarrow -\text{NO}_2 + 6\text{H}^+ + 6\text{e}^-$ (Figs. 2i and j)²⁸²⁸” (Page 8, Lines 182-184), together with citing the relevant reference [Zhang et al., *Sep. Purif. Technol.* 292 (2022) 121021].

(3) We deleted our claim on the protonation of PM ions *via* an ammonium intermediate “–NH₃⁺ of S-PAcH are deprotonated to –NH₂ while protonating the adsorbed PM ions *via* the acid (–NH₃⁺)–base (PM ion) reaction”.

(4) We included comments on the corrected PM adsorption mechanism as follows, “PM ions would exist as deprotonated anionic species (*i.e.*, AuCl₄⁻, PdCl₄²⁻, and PtCl₆²⁻) at low pH owing to the strong acidity of their precursors⁴¹” (Page 7, Lines 168-170), “Furthermore, hydrazide –NH₂ of S-PAcH would presumably be protonated to a less extent than conventional –NH₂ owing to its lower basicity (*i.e.*, higher pK_b)⁴². This was evidenced by the fact that the positive zeta potential of S-PAcH noticeably increased while those of conventional amine polymers were nearly unchanged when pH decreased from 2 to 1 (Fig. 1c). Hence, it can be reasonably postulated that S-PAcH at low pH (*i.e.*, pH 2) contains both protonated (–NH₃⁺) and unprotonated (–NH₂) primary amines (*i.e.*, protonated and unprotonated hydrazides)” (Page 7, Lines 172-178), and “Because one –NH₂ group of S-PAcH provides six electrons during its oxidation to –NO₂, it can reduce

multiple PM ions (*i.e.*, 2 for AuCl_4^- , 3 for PdCl_4^{2-} , and 1.5 for PtCl_6^{2-}) to PM NPs²⁸ (Fig. 2i and Supplementary Text)” (Page 8, Lines 184-187), together with replacing Fig. 1c and Figs. 2h–j with the above Figs. R2 and R3, respectively, and providing the above Eqs. R1–R9 and related discussion as a new Supplementary Text.

Comment 3-2: Regarding the recycling process, the continuous addition of thiourea and FeCl_3 without subsequent separation raises questions about their impact on the PM ion reduction process. If these reagents are not removed before the next capture cycle, they could potentially desorb captured PM nanoparticles, undermining the recycling efficiency. The manuscript should clarify the process, particularly emphasizing the treatment or removal of desorption agents before polymer reuse, to ensure the described procedure is viable and logical. In summary, while the manuscript presents a promising approach to PM recycling using smart polymers, addressing the aforementioned concerns with additional experimental evidence and clarification will be critical for advancing the manuscript toward publication.

Response to Comment 3-2: Thank you for the reviewer’s valuable comment. First above all, **the regeneration process (using thiourea, FeCl_3 , and HCl) we employed in this study is a well-established protocol, which has been widely used in many other studies, and has proven effective at regenerating adsorbents without impairing their adsorption capabilities** [Zhang et al., *Sep. Purif. Technol.* 292 (2022) 121021; Jung et al., *Chem. Eng. J.* 438 (2022) 135618; Mao et al., *Water Res.* 44 (2010) 5919; Tang et al., *Chem. Eng. J.* 407 (2021) 127223; Chang et al., *J. Hazard. Mater.* 391 (2020) 122175; Bratskaya et al., *Ind. Eng. Chem. Res.* 55 (2016) 10377]. Nevertheless, we further verified the viability of our regeneration process as follows.

As the reviewer mentioned, the complete removal of desorption agents (thiourea, Fe^{3+} , and HCl) from S-PACH(L) adsorbent after PM desorption is critical to maintain the high PM adsorption capability of S-PACH(L) for the next adsorption process. As mentioned in the “Methods” section of the original manuscript, the mixture solution after PM desorption was filtered through a PSF ultrafiltration membrane (molecular weight cut-off = 20 kg mol⁻¹). During this filtration step, very small thiourea (molecular weight = 76.12 g mol⁻¹) and ionic species (*i.e.*, FeCl_3 , HCl, and PM ions) permeated through the membrane, whereas large S-PACH(L) (molecular weight = 318 kg mol⁻¹) was screened, enabling complete collection of S-PACH(L). The filtrated S-PACH(L) was further washed with methanol to completely remove loosely bound thiourea and ionic species as follows. Because S-PACH(L) is marginally soluble in methanol to form clusters (easy collection) while thiourea and ionic species are highly soluble in methanol, the filtrated S-PACH(L) was immersed in a methanol bath with stirring for 1 h followed by filtration through a cellulose filter paper (pore size = 1 μm). The collected S-PACH(L) was freeze-dried and used for the next adsorption step. **The complete removal of desorption agents and PM ions from S-PACH(L) was confirmed by XPS spectra of S-PACH(L) before and after the regeneration process as shown in Fig. R4; no S (corresponding to thiourea), Fe (corresponding to FeCl_3), Cl (corresponding to FeCl_3 and HCl), and PM (corresponding to PM ions) peaks were detected for S-PACH(L) before and after regeneration.**

Fig. R4. XPS survey spectra of S-PACH(L) before and after the regeneration step.

The regenerated S-PACH(L) also maintained its high PM recovery efficiency during seven adsorption–desorption cycles, as demonstrated in Figs. 5g–i. This result implies that desorption agents and PM ions were completely removed to maintain the PM adsorption capability of S-PACH(L).

Finally, after PM desorption, desorbed PM ions can coexist with desorption agents, which need to be removed *via* additional separation processes to obtain high-purity PMs. Fe³⁺ ions can be removed *via* precipitation through simple pH adjustment for the following reason. Fe ions are precipitated at pH ≥ 3 [Stefánsson, *Environ. Sci. Technol.* 41 (2007) 6117], while Pd ions are precipitated at pH ≥ 5 [Park et al., *J. Hazard. Mater.* 181 (2010) 794], and Au and Pt ions are not precipitated at the pH range of 1–10 [Wang et al., *J. Phys. Chem. C* 113 (2009) 6505; Romero-Freire et al., *Ecotoxicol. Environ. Saf.* 227 (2021) 112924]. The precipitation behavior of PM ions was described in our previous Supplementary Text and Fig. 18. Thus, by adjusting the solution pH to 3–4, we can preferentially precipitate out Fe ions over PM ions. We verified the feasibility of the aforementioned method by adjusting the pH of the desorption solution containing thiourea (1M), FeCl₃ (1M), and HCl (1M). The pristine desorption solution is brown-colored (Fig. R5a). When the solution pH was adjusted to 3 using 30% ammonia solution (Daejung Chemical), brown-colored insoluble Fe precipitates and colorless supernatant solution were obtained (Fig. R5b). The precipitates were readily collected with a cellulose filter paper (pore size = 1 μm), and the Fe concentration in the permeate solution was measured to be ~ 0 ppm using ICP-OES. This result suggests that Fe ions can be readily removed by simple pH adjustment.

Fig. R5. Photographs of the PM desorption aqueous solution containing thiourea (1M), iron chloride (FeCl_3 , 1M), and HCl (1M) (a) before and (b) after adjusting the solution pH to 3.

Then, thiourea, HCl, and ammonia remaining in the solution can be removed by thermal treatment because they are completely vaporized (HCl and ammonia) and decomposed (thiourea) at $300\text{ }^\circ\text{C}$ [Park et al., *RSC Adv.* 4 (2014) 9118].

In the revised manuscript, we made following revision.

(1) We included comments to justify our regeneration protocol as follows, “Small thiourea (molecular weight = 76.12 g mol^{-1}) and ionic species (*i.e.*, FeCl_3 , HCl, and PM ions) permeated through the membrane, whereas large S-PACH(L) (molecular weight = 318 kg mol^{-1}) was screened, enabling complete collection of S-PACH(L)” (Page 21, Lines 494-497) and “The filtrated S-PACH(L) was further washed with methanol to completely remove loosely bound thiourea and ionic species. Because S-PACH(L) is marginally soluble in methanol to form clusters while thiourea and ionic species are highly soluble in methanol, the filtrated S-PACH(L) was immersed in a methanol bath with stirring for 1 h followed by filtration through a cellulose filter paper. The complete removal of desorption agents and PM ions from S-PACH(L) was confirmed by the XPS spectra of S-PACH(L) before and after the regeneration process (Supplementary Fig. 37)” (Pages 21-22, Lines 500-506), together with providing the above Fig. R4 and related discussion as a new Supplementary Fig. 37.

(2) We also included the comment on the removal of desorption agents as follows, “After PM desorption, desorbed PM ions can coexist with desorption agents, which need to be removed *via* additional separation processes to obtain high-purity PMs. Fe^{3+} ions can be readily removed by adjusting the solution pH to 3–4, where Fe ions can be preferentially precipitated over PM ions (Supplementary Figs. 19 and 36 and Supplementary Text)⁶¹. Thiourea and HCl can also be removed by thermal treatment because they are completely vaporized (thiourea) and decomposed (HCl) at $300\text{ }^\circ\text{C}$ ⁶²” (Page 15, Lines 354-360), together with providing the above Fig. R5 and related discussion as a new Supplementary Fig. 36.

REVIEWERS' COMMENTS

Reviewer #1 (Remarks to the Author):

The latest revised version of the manuscript satisfactorily addresses the concerns raised. I have no further reservations to recommend the article for publication in 'Nature Communications'.

Reviewer #3 (Remarks to the Author):

The authors have properly addressed all the reviewers' comments. I recommend acceptance of the current manuscript.

Responses to the Reviewers' Comments

Reviewer #1

General comment: The latest revised version of the manuscript satisfactorily addresses the concerns raised. I have no further reservations to recommend the article for publication in 'Nature Communications'.

Response to General Comment: We thank the reviewer for many positive and constructive comments during the revision process.

Reviewer #3

General comment: The authors have properly addressed all the reviewers' comments. I recommend acceptance of the current manuscript.

Response to General Comment: We thank the reviewer for many positive and constructive comments during the revision process.